# MULTI-LINGUAL EVALUATION OF CODE GENERATION MODELS

**Ben Athiwaratkun[†] , Sanjay Krishna Gouda[†], Zijian Wang[†], Xiaopeng Li[†], Yuchen Tian[†],
Ming Tan, Wasi Uddin Ahmad, Shiqi Wang, Qing Sun, Mingyue Shang, Sujan Kumar
Gonugondla, Hantian Ding, Varun Kumar, Nathan Fulton, Arash Farahani, Siddhartha Jain,
Robert Giaquinto, Haifeng Qian, Murali Krishna Ramanathan, Ramesh Nallapati,
Baishakhi Ray, Parminder Bhatia, Sudipta Sengupta, Dan Roth, Bing Xiang[†]**

AWS AI Labs

## ABSTRACT

We present two new benchmarks, MBXP and Multilingual HumanEval, designed
to evaluate code generation models in over 10 programming languages. These
datasets are generated using a conversion framework that transpiles prompts and
test cases from the original MBPP and HumanEval datasets into the corresponding
data in the target language. By using these benchmarks, we are able to assess the
performance of code generation models in a multi-lingual fashion, and discovered
generalization ability of language models on out-of-domain languages, advantages
of multi-lingual models over mono-lingual, the ability of few-shot prompting to
teach the model new languages, and zero-shot translation abilities. In addition, we
use our code generation model to perform large-scale bootstrapping to obtain syn-
thetic canonical solutions in several languages, which can be used for other code-
related evaluations such as code insertion, robustness, or summarization tasks. [*]

## 1 INTRODUCTION

Code completion by machine-learning models has great potential to improve developer productivity
(Barke et al., 2022). This line of research has seen tremendous progress with several models recently
proposed such as Codex (Chen et al., 2021), CodeGen (Nijkamp et al., 2022), PaLM (Chowdhery
et al., 2022), BLOOM (Mitchell et al., 2022), and InCoder (Fried et al., 2022).

One key component for code generation research is how to evaluate such program synthesis abil-
ities. In the literature, two primary evaluation approaches emerged, namely, the match-based and
the execution-based evaluations. For both approaches, each problem contains a *prompt* which a
model uses as input to generate a candidate body of code. The match-based evaluation compares
the candidate code *against reference source code* using n-gram metrics such as BLEU, whereas the
execution-based evaluation executes the candidate code *against test cases* and calculates success
rate. The execution-based evaluation has benefits over the n-gram evaluation in that it permits solu-
tions that are functionally correct but might not be equivalent to the reference solution in terms of
the exact implementation. Since the release of datasets such as HumanEval (Chen et al., 2021) or
MBPP (Austin et al., 2021), the community has been widely adopting the execution-based approach
as a primary tool to evaluate program generation capabilities. However, creating execution-based
evaluation datasets is time-consuming since it requires careful construction of test cases to check
the correctness of the code's functionality. Such difficulty leads to limited available of execution-
based evaluation data. For instance, to date, many execution-based datasets contain only problems
in Python.

[†]Corresponding authors {benathi,skgouda,zijwan,xiaopel,tiayuche,bxiang}@amazon.com
[*]We release the data and evaluation code at https://github.com/amazon-research/
mbxp-exec-eval.

Figure 1: Benchmark Construction.

In this work, we propose a scalable framework for dataset convertion from Python to many different languages. While translating code from one language to another is typically a non-trivial task, it is possible to convert existing execution-based datasets to another language by transforming only *prompts* and *test statements* (see Figure 1 part A and Figure 2). That is, the purpose of evaluating function completion ability, we do not need the canonical solution since it is not used during evaluation. The function signature prompts and test cases of basic programming problems involve sufficiently simple data structures that can be analyzed to synthesize dataset in new languages. Without having to translate the generic function body of code to another language, the conversion process becomes possible via a rule-based transpiler.

The result of such conversion are two benchmarks, MBXP[‡] and Multilingual HumanEval, which are derived from the original Python dataset MBPP (Austin et al., 2021) and HumanEval (Chen et al., 2021). We provide the evaluation data in many languages besides the original Python, namely, Java, JavaScript, TypeScript, Go, Ruby, Kotlin, PHP, C#, Scala, C++, Swift, and Perl, with plans for more language expansion in the future. Along with these datasets, we also release a code package to perform execution in all supported languages. In addition, our conversion framework is easily extensible and allows us to obtain the multi-lingual version of other existing datasets such as MathQA (Schubotz et al., 2019). In the main paper, we provide results and analyses mostly on MBXP where the results on Multilingual HumanEval and MathQA can also be found in Appendix D.

Our benchmarks also support other code completion tasks such as code insertion or translation in many languages. This extension is made possible by performing large-scale bootstrapping to synthetize solutions (Section O.1.11). The result of our dataset conversion framework and the solution synthesis process is, to date, the first multi-lingual execution-based evaluation benchmark equipped with canonical solutions, which can be adapted for many code-related evaluations. In this paper, we process MBXP for multiple use cases, namely, for zero-shot translation t-MBXP, prompt robustness r-MBXP, code insertion i-MBXP, and the summarization s-MBXP.

Overall, the constructed datasets provides us new opportunities to explore many facets of code generation abilities. In this work, we conduct a large scale evaluation where we train models of various sizes spanning three orders of magnitude (from $\sim$ 100M to $\sim$ 10B parameters) in both multi-lingual and mono-lingual settings. We analyze results from hundreds of thousands of code generation samples to investigate the models' code generation abilities with respect to in-domain versus out-of-domain languages, the effectiveness of few-shot prompting, zero-shot translation abilities, robustness to prompt perturbation, code summarization, and code insertion.

## 2 FINDING HIGHLIGHTS

We provide the highlights of out findings below.

1. Given the same model size, a multi-lingual model often outperforms the best of mono-lingual models trained with equivalent training resources, especially when the models are sufficiently large. This observation indicates that it is beneficial to train a single model on all programming languages, and provided that the model size has enough capacity, the performance will be better than the best of monolingual models.

2. Language models are able to generate code with correct syntax and pass unit tests in programming languages they are not intentionally trained on. We hypothesize that the data "spillover" effect, where code in one language is present in other languages through code

---

[‡]MBXP stands for **M**ost **B**asic **X**(Python/Java/Go/Ruby, etc.) Programming **P**roblems

Function signature and docstring

```python
def binomial_coeff(n,k):
    """ Write a Python function to
find binomial coefficient.
    >>> binomial_coeff(5,2)
    10
    """
```

```java
class BinomialCoeff {
    /** Write a Java function to find binomial co-
efficient.
     * > BinomialCoeff.binomialCoeff(5, 2)
     * 10
     */
    public static int binomialCoeff(int n, int k){
```

Function body

```python
    if k > n :
        return 0
    if k==0 or k ==n :
        return 1
    return binomial_coeff(n-1,k-1)
+ binomial_coeff(n-1,k)
```

```java
        if (n < k) return 0;
        int res = 1;
        for (int i = 1; i <= k; i++) {
            res *= (n + 1 - i);
            res /= i;
        }
        return res;
```

Test cases

```python
assert binomial_coeff(5,2) == 10
assert binomial_coeff(4,3) == 4
assert binomial_coeff(3,2) == 3
```

```java
class Main {
    public static void main(String[] args) throws
Exception {
        int x0 = BinomialCoeff.binomialCoeff(5, 2);
        if (!x0.equals(10)) {
            throw new java.lang.Exception("Exception
-- test case 0 did not pass. x0 = " + x0);
```

Model

Figure 2: Conversion of formatted MBPP (Python) to MBJP (Java).

comments or co-occurrences. Such amount of spillover data are enough for large language models to learn different languages that are embedded within the main language.

3. The occurrences of multi-lingual data in natural data also explains the superior performance of multi-lingual over mono-lingual models. That is, the multi-lingual model can perform better on language $A$ since it can pick up and combine all knowledge of language $A$ from the training data in languages $A$, $B$, $C$, etc. in the multi-lingual setting.

4. Few-shot prompting can effectively help teach provide knowledge on a new language the model has not seen, significantly improving out-of-domain code generation abilities. Through error analysis, few-shot prompting helps reduce compilation or parsing errors that are the major sources of errors when it comes to a programming language the model is not familiar with.

5. Language models have zero-shot code translation abilities; that is, even though they are not specifically trained to perform translation, they are able to use reference code in one language to improve code generation in another language. Problems that are difficult can become much easier with access to another language's solution. This observation holds for mono-lingual as well as multi-lingual models.

6. Multi-lingual models are also more robust to prompt perturbation and better at summarizing code.

## 3 CONVERSION OF EXECUTION-BASED EVALUATION DATASETS

In this section, we provide high-level details on the data conversion process. Figure 2 illustrates the mapping of the original Python prompt, consisting of a function signature and a docstring, to an equivalent prompt in Java (which we call a target prompt). The target prompt is a valid code including a function signature from which the model can use to complete the function body. In the case of Java or typed languages, constructing the target prompt requires inferring input and output types. We perform such type inference by parsing the original test cases, taking into account heterogeneous data types. For instance, if the first argument includes values of types `int` and `float`, we deduce it to have the most general type of all types encountered. The converted prompt also needs to work in harmony with the converted test cases. For instance, the Java test case in Figure 2 refers to the defined class `BinomialCoeff` and the defined method `binomialCoeff` in the converted prompt with appropriate function call based on the defined argument list. For more details including data validation and generated solutions via bootstrapping, see Appendix O.

## 4 Multi-lingual Evaluation of Code Generation Models

From the previous section, we have established a framework to perform dataset conversion, from which we obtain a collection of execution-based evaluation datasets in 10+ programming languages. These evaluation datasets contain rich information in the prompts including natural language description as well as appropriate function signatures that help steer a model to generate code in a particular language. Most importantly, they also contain test cases in the respective language that can be used to check code correctness, which is applicable for most types of evaluation in MBXP+. This section describes the training, evaluation setup, and findings from each evaluation task.

### 4.1 Data and Models

For the purpose of this work, we collected training data in three primary programming languages, namely, Python, Java, and JavaScript, containing permissively licensed code data from GitHub. Following Chen et al. (2021); Nijkamp et al. (2022), we perform filtering, deduplication, and remove data that contains a significant amount of non-English text or is not parsable with respect to that language's syntax parser. We also ensure the original MBPP and HumanEval datasets are not included in data. After all the post processing steps, our dataset contains 101 GB Python, 177 GB Java, and 216 GB JavaScript data.

We use a decoder-only transformers as the model architecture and train the models via next-token prediction loss (Vaswani et al., 2017; Brown et al., 2020). We design our training to compare multi-lingual versus mono-lingual settings by using the same compute budget for each language in both cases. In particular, we train mono-lingual models on 210 billion tokens with their respective languages (Python, Java, and JavaScript) and train multi-lingual models on 210 billion tokens from each language, with 630 billion tokens in total. To study effects of model sizes, we train models of various number of parameters, namely, 125M, 672M, 2.7B and 13B. For the synthetic canonical solution process, we use a separate 13B multi-lingual model which we refer to as the 13B* model.

### 4.2 Execution-Based Function Completion

We use pass@k scores (Kulal et al., 2019) with the unbiased estimate presented in (Chen et al., 2021) as the metrics for our evaluation, where each task is considered successful if any of the $k$ samples are correct. We generate up until the end of the function, such as end of indented function block for Python or until the closing curly brace for PHP or Go, for example (see Appendix C.2 for end of scope details). We refer to an evaluation language that the model is not specifically trained on as *out-of-domain* with respect to that model. Otherwise, the language is considered *in-domain*. For instance, Java is out-of-domain for a Python-only model and PHP is out-of-domain for our multi-lingual model trained on Python, Java, and JavaScript.

#### 4.2.1 Accuracy vs. Sampling Budget

Overall, we observe sigmoid-like relationships between pass@k and sampling budget $k$ across all datasets in MBXP where the performance increases smoothly as $k$ increases (Figure 3, and Appendix F.2). This trend is consistent with the original MBPP and HumanEval which are manually-annotated. This sigmoid-like performance with respect to sampling budget indicates that problems vary in terms of difficulty, where certain problems require many more attempts to get them right. We do not find a degenerate case in any evaluation language where all problems are either trivial to solve (pass@k saturated near 100%), or impossible (pass@k all zeros). The consistency of the observed performance trend across all programming languages in the MBXP benchmark provides reassurance regarding the benchmark's applicability as a multi-lingual evaluation tool for assessing a model's capabilities at different levels.

#### 4.2.2 Generalization to out-of-domain languages

As demonstrated in Figure 3, our model can achieve non-zero pass@k scores for out-of-domain languages. We emphasize that our models are not specifically trained on out-of-domain languages since we filter languages based on file extensions and verify that the data have correct syntax with respect to each language (refer to Section 4.1). However, we hypothesize that cross-language data

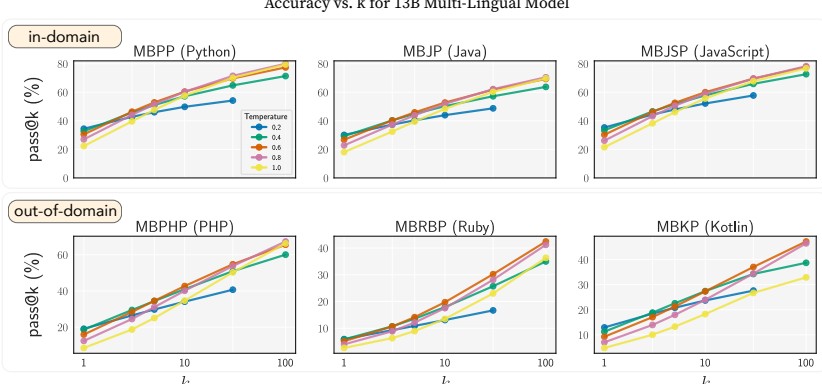

Figure 3: pass@k versus sampling budget $k$ for various datasets across MBXP. We observe generalization behavior where the model can write valid code on languages not trained on, as indicated by the non-zero execution scores on out-of-domain evaluation. Model performance also tends to be sigmoid-like; that is, when the performance is on the lower end such as in the out-of-domain case, the curve breaks out upward, similar to the earlier part of the sigmoid function. The behavior also applies for models of other sizes as well as mono-lingual models (not shown in this figure).

spillover are quite typical, since there can be data related to other languages mentioned in code comments, natural texts, or intentionally embedded in cross-lingual code projects. Examples of such projects are Django or Flask, where JavaScript pieces of code can be embedded in Python files for web development, or mixed use of Java and Python code in projects such as Jython. We provide further discussion of types and examples of cross-lingual data occurrences in Appendix E.

In Figure 4a, we also observe that the out-of-domain scores are not symmetric for a given language pair; i.e., Python models perform well on Java but Java models have negligible performance on Python. The data spillover hypothesis supports this observation where it is likely that there are many languages embedded in, e.g. Python files, whereas not as many languages are embedded in Java files. We provide further analyses related to data spillover hypothesis in Section 4.2.3.

### 4.2.3 MULTI-LINGUAL VERSUS MONO-LINGUAL MODELS

Figure 4a shows a plot of pass@k scores versus model sizes for multi- and mono-lingual models, where we observe approximate log-linear relationships similar to those found in the literature (Chowdhery et al., 2022; Chen et al., 2021; Nijkamp et al., 2022; Austin et al., 2021; Li et al., 2022a). For small model sizes, we see that multi-lingual models can perform slightly sub-par or on-par to mono-lingual models. For instance, at size 125M and 672M, mono-lingual models outperform multi-lingual models in some evaluation languages such as Python and Ruby. However, once we reach a certain size such as 2.7B or 13B parameters, a large multi-lingual model begins to outperform the best of mono-lingual models in all evaluation languages. The performance gains of multi-lingual over mono-lingual models are particularly significant for out-of-domain languages such as MBPHP and also noticeable for in-domain ones such as MBJSP and MBJP.

We observe that for MBPP, the mono-lingual Java and JavaScript models obtain close to zero pass@k, suggesting that the amount of spillover Python code in Java or JavaScript training data is likely low. This finding coincides with the Python and multi-lingual models achieving near identical MBPP scores in Figure 4a, suggesting that both Python and multi-lingual models observed similar amount of Python code during training. This evidence is consistent with the previous observation that there is little Python code in Java or JavaScript training data.

In contrast, for the JavaScript evaluation (MBJSP) shown in Figure 4a, each of the mono-lingual models obtain reasonable pass@k scores, suggesting that the spillover of JavaScript code is prevalent (at least in Python and Java data). This finding also explains why the multi-lingual model performs significantly better to the JS model on JS evaluation (MBJSP), as the multi-lingual model learn JS knowledge from other sources, while the mono-lingual JS model's source of knowledge is more limited.

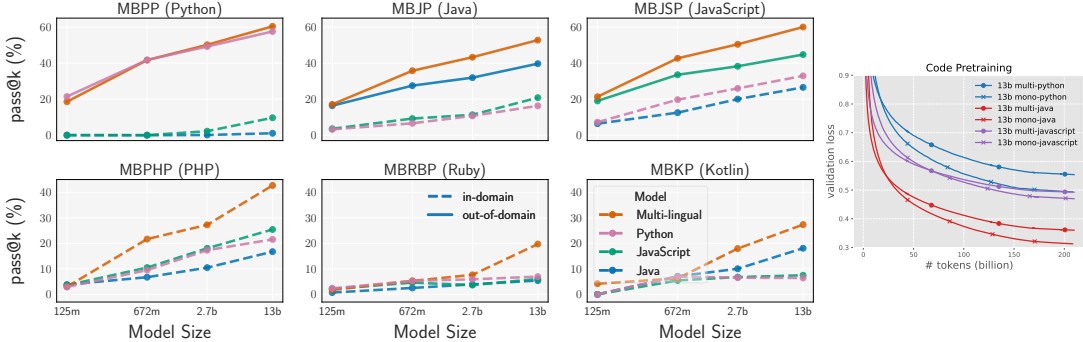

(a) pass@10 vs. model size for multi- and mono-lingual models

(b) Validation losses per language

Figure 4: (a) We observe log-linear relationships between model sizes and scores, with multi-lingual models outperforming mono-lingual ones. This trend persists across all evaluation datasets in MBXP, including out-of-domain languages such as PHP, Ruby, and Kotlin. Interestingly, the performance of MBRBP (Ruby) breaks out of this log-linear trend, as the multi-lingual 13B model performs significantly better than the extrapolated performance would suggest. (b) Despite having higher validation losses for each in-domain language compared to their mono-lingual counterparts, multi-lingual models consistently outperform mono-lingual models in all evaluation datasets in MBXP.

On languages such as PHP, Ruby, Kotlin which are outside of the core training data (Python, Java, JS), multi-lingual models are also more capable of learning such languages, as demonstrated in Figure 4a. Overall, the performance in the multi-lingual setting tends to improve more rapidly as they are able to draw knowledge from many sources at once, as observed by higher slopes in the plots (Figure 4a).

Interestingly, we note that even though the multi-lingual models perform better during evaluation, the validation losses per language for multi-lingual models are higher than those of mono-lingual models (See Figure 4b). We provide further discussion on validation losses in Appendix N.2.

### 4.3    ZERO-SHOT CODE TRANSLATION

Our dataset conversion framework yields parallel data in many different languages. These parallel datasets provide a valuable resource for studying the translation abilities of code generation models, as we can evaluate how well the models generate code in any other supported language using the canonical solutions in our source language. For this study, we prepend the function in a source language to the beginning of the function-completion prompt of the target language (Figure 5). We can also think of this setup as a function completion with augmented information, where we provide a reference solution in another language. Therefore, we also refer to the usual function completion setup as the non-translation setting.

**Zero-shot translation abilities**    Figure 6a showcases the ability of language models to perform translation by using reference solutions in a different language to aid in function completion. Examples in Figures 5 and 8 illustrate how the models are able to produce code that retains the same underlying logic as the reference solution in Python, while conforming to the syntax of the target language, such as PHP (e.g., using $ before variable names) Specifically, the generated code in the translation mode mimics the content of the reference solution, including the same loop and control flow structures, as shown in the upper part of Figure 8. Additionally, the generated code cab exhibit similar semantics, such as sorting by the summation, as illustrated in the lower part of Figure 8.

Interestingly, our analysis in Figure 7c suggests that the translation setting can enable the models to solve problems that are otherwise difficult without the aid of reference solutions. For instance, on the MathQA dataset, which requires complex reasoning but has solutions with simple arithmetic syntax, our models are able to translate to a different language with near-perfect accuracy, achieving almost $100\%$ pass@100 scores (see Appendix D.1).

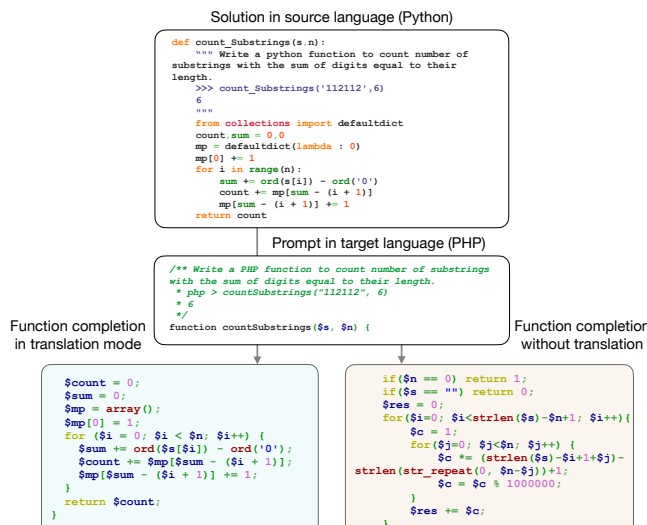

Figure 5: Demonstration of prompt construction in the translation setting where we prepend the source language's solution. In this translation, the generated code retains similar logic as the reference solution, but has the correct syntax of the target language.

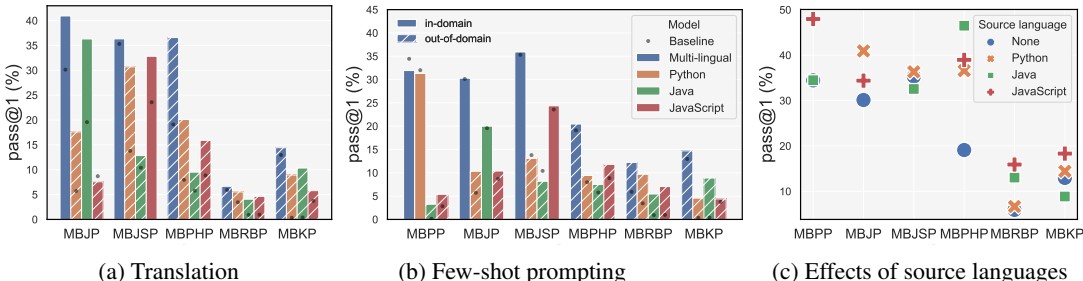

(a) Translation      (b) Few-shot prompting      (c) Effects of source languages

Figure 6: **(a)** Translation setting: Significant gain in function completion ability when using reference solutions in Python. **(b)** Few-shot prompting: Improvement on out-of-domain evaluation due to few-shot prompting, where the examples help guide the model to generate more correct code in the given language. **(c)** Translation with different source languages: Reference solutions in different source languages have unequal effects on translation performance. Note: The dots in (a) and (b) indicate the baseline without few-shot or translation. All results are from the 13B multi-lingual model.

**Unequal effects of different source languages** We find that different source languages can interact quite differently with each target language. For instance, JavaScript yields better translation performance as a source language compared to Python, when evaluated on datasets such as Kotlin (MBKP) or Ruby (MBRBP). Languages that are too close in terms of syntax can confuse the model when used together in the translation setting. For instance, Python as a source language for translation to Ruby can sometimes lead the model to generate code in Python, which is undesirable. For each evaluation language, the best source language is fairly consistent, relatively consistent across models. We discuss the language compatibility with respect to translation further in Appendix H.1.

### 4.4 FEW-SHOT PROMPTING

Few-shot prompting is a technique that can provide additional information to a language model, which can help it perform better on specific tasks (Brown et al., 2020). In our experiment, we construct few-shot prompts consisting of three correct functions from the respective MBXP dataset (see Appendix G.2 for prompt format). We observe consistent improvement in execution accuracies, especially for out-of-domain evaluations, as shown in Figure 6b.

One possible explanation for this improvement is that the few-shot prompt can help disambiguate programming languages, which is most beneficial in out-of-domain evaluations when the models

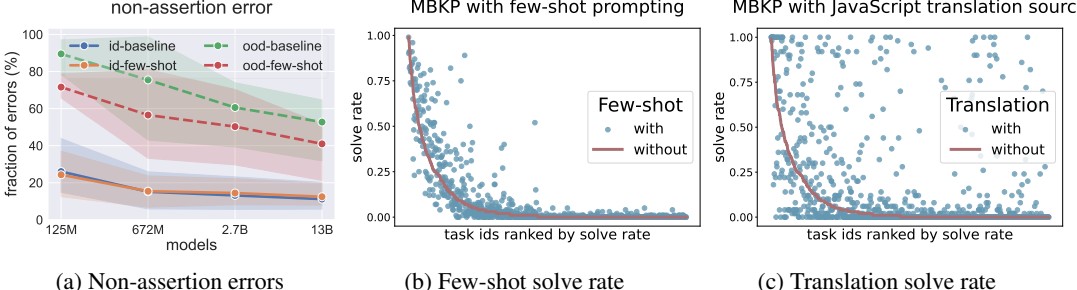

(a) Non-assertion errors     (b) Few-shot solve rate     (c) Translation solve rate

Figure 7: **(a)**: Few-shot prompts results in lower non-assertion (compile, parsing, syntax) errors on out-of-domain (ood) evaluation but has no effect on in-domain (id). **(b)** and **(c)**: For each task, we show a fraction of generations that pass the tests over the total number of samples (solve rate), where the task indices are ranked to show increasing difficulty. In the translation setting, tasks that are previously difficult (low solve rate for the baseline) can become easily solvable, demonstrating that models can leverage reference solutions in the source language to solve hard tasks. In contrast, the solve rates with few-shot prompting do not deviate as much per task from the baseline solve rate.

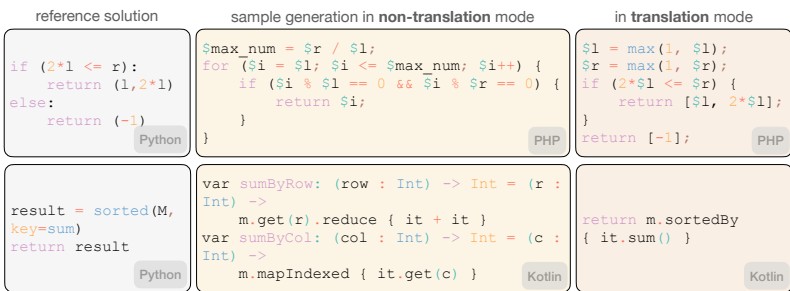

Figure 8: Code generation model can use the style and content of a reference solution in the translation setting to generate a correct solution in a different language.

are not familiar with the target language. For example, in MBRBP evaluation (Ruby), the Ruby function signature can be very similar to that of Python, which can lead to confusion and the model generating Python code without the few-shot prompt. The error analysis in Figure 7a demonstrates that compilation, syntax, or parsing errors (non-assertion errors) drop significantly due to the few-shot prompts.

The improvement due to few-shot prompts also applies to other datasets such as MathQA (Appendix D.1). These observations suggest that soft prompts obtained via prompt tuning or its variants (Lester et al., 2021; Liu et al., 2021b;a; Li & Liang, 2021) could further help condition models to perform better in out-of-domain or scarce programming languages.

## 4.5 ROBUSTNESS EVALUATION: R-MBXP

We evaluate the robustness of models across r-MBXP datasets perturbed by common transformations in NL-Augmenter (Dhole et al., 2021), a standard collection of data augmentations for robustness evaluation on text. Our experiments show that multi-lingual models are more robust on average, with less percentage of performance drops (7.80% vs 9.39% for multi- and mono-lingual models) and higher pass@1 scores across most perturbed datasets compared to mono-lingual models. For more details and other interesting observations on robustness, we refer readers to Appendix J. As the first code-generation robustness benchmark, we encourage researchers to further investigate robustness evaluation metrics, data augmentations, adversarial attacks, and defenses based on our released datasets.

## 4.6 CODE INSERTION: I-MBXP

We introduce i-MBXP, an insertion-based variant of our MBXP benchmark, which is the first execution-based multi-lingual code insertion benchmark. Each data sample consists of left and right

contexts where we split the original function signature and the canonical solution into left context, right context, and ground truth insertion code. Code insertion is evaluated in an execution-based manner by using the same test statements as in MBXP. We benchmark using the publicly available insertion-based model, InCoder (Fried et al., 2022).

Both models show that incorporating right context can significantly boost performance compared to using only the left context. In InCoder, we observed 23.2%, 14.4%, and 37.6% relative improvements on Python, JavaScript, and Java respectively compared to the case without right context. Ablation studies on the performance versus the number of right context lines show a positive correlation, indicating the models' abilities to incorporate partial right context information to improve prediction.

We provide further details on dataset construction and results in Appendix K. This work demonstrates the usefulness of our MBXP benchmark for code insertion and highlights the need for further research in execution-based multi-lingual code insertion evaluation.

### 4.7 Code Summarization: s-MBXP

We evaluate the ability of models to perform code summarization, where we use a function signature along with its solution as the prompt, with the natural language description in the docstring removed. Based on this prompt, we induce the model to generate the description of the code's functionality. Our results show that, in both zero-shot and few-shot settings, multi-lingual models generally outperform mono-lingual models, consistent with the performance trends observed in other evaluation tasks discussed in Section 4.2.3. In the few-shot case, we observe noticeable improvements compared to the zero-shot setting, with more significant improvement on larger models. We provide examples and detailed results in Appendix L.

## 5 Related Work

Many other evaluation datasets can be considered for the conversion to multi-lingual counterparts such as APPS (Hendrycks et al., 2021) and CodeContest (Li et al., 2022a). These datasets in its original forms are execution-based datasets containing challenging algorithmic competition problems and tests that are language-agnostic, but can be converted to Python and many other languages. Existing benchmarks for code generation are primarily either match-based or focused mostly on Python, if not language-agnostic. Our work fills a gap in the literature by providing a multi-lingual code evaluation framework that includes synthetic solutions, handles datasets beyond HumanEval (e.g., MBPP and MathQA), and investigates various types of code generation abilities. Concurrent work by Cassano et al. (2022) converts prompts and test cases of HumanEval into multiple languages. Recent work by Orlanski et al. (2023) presents BabelCode, a framework for execution-based evaluation, and investigates the effectiveness of balancing the distribution of languages in a training dataset.Together, these works provide a valuable resource for researchers to evaluate multi-lingual code generation. We provide further discussion of related work in Appendix B.

## 6 Discussion

Our release of these datasets is a significant contribution to the field of code generation research, providing researchers with a valuable resource to evaluate various aspects of code generation abilities. The findings from our evaluations have shed light on interesting areas such as multi- vs mono-lingual models, out-of-domain performance, zero-shot translation abilities, and multi-lingual code insertion, all of which hold potential for advancing the state-of-the-art in code generation.

Our observations suggest that large multi-lingual models are more effective than multiple monolingual models in code generation tasks, benefiting from the data spillover across languages. The success of our multi-lingual models in out-of-domain evaluations and robustness testing demonstrates their potential to generalize to new languages and tasks. However, to comprehensively evaluate the complexities of real-world software development tasks, it may be necessary to include additional language-specific evaluations where appropriate. Overall, our datasets provide a solid foundation for future research to explore and enhance various aspects of code generation, with the potential to lead to significant advancements in the field.

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

# Appendix

## Table of Contents

# A  EXTENDED DISCUSSION

## A.1  IMPLICATION OF FINDINGS

From our findings, it is clear that a large multi-lingual model compared to multiple mono-lingual is a better choice if we are to consider deploying code generation models. This is due to the data spillover from each language source which reinforces the knowledge of the model in the multi-lingual training. However, such model needs to be of sufficient size to capture all the available knowledge. For our controlled setting, model sizes $2.7B$ and above begin to clearly outperform all mono-lingual models. It is possible that as the number of languages in the training set increase, the required size for the multi-lingual model to be superior to individual mono-lingual models can increase.

## A.2  IMPLICATION OF EVALUATION DATA AT SCALE

Our parallel datasets provide a valuable resource for studying the translation abilities of code generation models. By leveraging the canonical solutions in our source language, we can evaluate how well the models generate code in any other supported language. This opens up a range of research questions, such as how well the models generalize across languages, what factors contribute to successful or unsuccessful translations, and how different modeling strategies affect translation performance.

## A.3  POSSIBILITIES OF TRUE GENERALIZATION

Out-of-domain evaluations from our controlled experiments reveal interesting behavior of how code in multiple languages present themselves in natural data. We hypothesize that the out-of-domain code generation abilities are mainly due to the data spillover. However, we believe it is also possible that a true generalization plays a role where the model is able to complete code in a new language that is not in the training data at all. To test this, we can design a new language which avoids the complication of data spillover in the any training dataset. We can use our framework to construct the evaluation set in such language and use it to evaluate the existing models. However, we note that such new language likely are similar to existing languages in the training set in some aspects. For instance, the control flows (if clause), loops, variable declaration, or objects such as lists or dictionaries potentially might not differ much from each component of existing languages. Even with the new language constructed, the boundary between evaluating a true generalization versus generalization between data spillover can be somewhat unclear.

## A.4  POTENTIAL PROXY FOR GENERAL CODING CAPABILITIES

MBXP and other code completion benchmarks such as HumanEval measure the general understanding of basic tasks from natural language description with function signature and the model's ability to complete such tasks. Given the description of these problems in natural language and function signature where a competent human coder can complete, this benchmark helps measure if a code generation model can perform such tasks. The scores of these evaluations can be a useful proxy for overall code capabilities if they correlate with the performance on all coding-related tasks. We believe that such correlation is possible or likely the case if the models are not trained to adapt to a specific distribution of evaluation datasets. By using these evaluations as proxies of general coding abilities, we implicitly accept the premise that zero-shot evaluation on a slice of all possible problems (the slice being MBXP, for instance) is an unbiased proxy to measure overall model's capabilities in each language. Hence, in this paper, we particularly avoid finetuning even though results in the literature demonstrate increased performance so that the results established can be less biased towards specific kinds of coding problems and can better reflect models' true coding capabilities.

## A.5  LIMITATIONS

The proposed conversion framework is well suited for basic programming problems that are applicable to a wide set of programming languages. While the original MBPP dataset is meant for basic programming problems, some tasks can be more appropriate for certain languages than others. For instance, string manipulation problems can be naturally encountered in languages such as Python

or PHP more than C++. By design, our conversion "naively" assumes that a problem is relevant to the target language which might not be true for all problems in a given dataset. That is, the scores obtained from MBXP benchmark might not align with the distribution of natural usage in different languages equally.

In addition, the programming problems in MBXP do not cover language-specific functionalities; for instance, there are no specific questions related to web development for JavaScript, or memory allocation for C++. Therefore, it can be unclear how the conclusion from MBXP benchmark transfers to coding performance in the wild given the complexity of real-world software development. The test conversion we support are *value-oriented* which do not cover all possible types of testing. The value-oriented test performs assertion by checking if the values match. If the assertion process is more complex such as in deep integration tests with specific code packages or APIs, the conversion process is not applicable. In fact, we explicitly define the types of Python objects that we support converting from in Appendix O. We suggest that it can be beneficial to complement MBXP evaluation with other language-specific evaluation, if available.

### A.6 GENERATION TENDENCY VERSUS GENERATION ABILITY

The prompts in our benchmark heavily guide the model to generate in a particular language. For example, when a prompt contains `function method_name(`, the model is highly encouraged to generate code that has such syntax, in this case PHP, but not Python where the function signature would have started with `def method_name(`. In that sense, this benchmark measures the ability of a model to conform to the guided prompt and the completion ability based on the function signature that has already started, and not necessarily the tendency to generate in particular languages. Note that without explicit guidance or special tags, models can generate code in any languages, especially multi-lingual models, which makes fair evaluation of code completion harder since we might not penalize correct code that is correct but in a different language. Our prompt format helps isolate evaluation of the generation ability in a desired language from the generation tendency. This is contrast to free-form prompt style in datasets like the original MBPP, APPs, or CodeContests where the model generates its own function signature. However, in our case, if the evaluation is out-of-domain, it is still possible that with explicit guidance of function signature, the model can still generate in a similar yet different language, as in the case of confusion between Ruby and Python with similar function signature syntax.

We also observe that while this benchmark is about generic understanding of basic programming tasks and does not particular attempt to measure the knowledge of a model in terms of specific syntax in the desired language , we observe that language-specific syntax usage typically emerge, for instance, the usage of `list.select` in Ruby, or `nums.filter` in Kotlin to select elements of a list, instead of a generic for loop. We provide sample generations for all converted languages in Section R.1.

## B OTHER RELATED WORK

**Code Generation Models**    Language models for code generation is a rising domain in recent years. CodeBERT (Feng et al., 2020) is the first BERT-based language model trained on code. GraphCode-BERT Guo et al. (2021) improves upon CodeBERT by leveraging AST and data flow. CodeT5 Wang et al. (2021) and PLBART Ahmad et al. (2021a) pretrained encoder-decoder based generative language models for code. More recently, various work have been proposed to use large language models for code generation. Codex (Chen et al., 2021) was pretrained on Python on top of GPT-3, resulting in up to 175B parameters code generation models. CodeGen (Nijkamp et al., 2022) was pretrained on multiple programming languages and optimized for conversational code generation with up to 16B parameters. InCoder (Fried et al., 2022), along with CoditT5 Zhang et al. (2022a) on a similar line of research, is able to perform program synthesis (via left-to-right generation) as well as editing (via infilling) with up to 6.7B parameters. Further, researchers also found that generic (natural) language models are also able to perform code completion to a certain degree, e.g., PaLM (Chowdhery et al., 2022) and BLOOM (Mitchell et al., 2022).

In addition, researchers proposed various ways of improving code generation models. For example, Poesia et al. (2022) propose Target Similarity Tuning for code retrieval augmentation and Con-

strained Semantic Decoding to improve code generation by constraining the output to a set of valid programs in the target language. Shi et al. (2022) introduce execution result–based minimum Bayes risk decoding that improves choosing a single correct program from among a generated set. Another line of work is to "repair" the generated code by language models, e.g., (Fan et al., 2022; Li et al., 2022b).

Our work is model-agnostic and complimentary to all the above works that serves as a testbed of code generation.

**Code Completion Resources**  Many code completion evaluation benchmarks have been proposed recently, but they differ in style and focus. Lu et al. (2021) composed a token and line completion datasets in Java and Python based on existing benchmarks (Allamanis & Sutton, 2013; Raychev et al., 2016). Clement et al. (2021) presented a method generation dataset in Python based on CodeSearchNet (Husain et al., 2019). All these datasets are primarily collected from open-source projects or GitHub and focus on match-based evaluation (using n-gram metrics). In contrast to these efforts, recent works promote unit tests-based execution evaluation to assess the functional correctness of ML-based code completion techniques.

In this line of work, Austin et al. (2021) introduced the MBPP dataset focusing on basic programming problems where the prompt format consists of a natural language description and assert statements in Python. HumanEval (Chen et al., 2021) focuses on more challenging algorithmic problems with prompts containing function signatures in Python along with docstring descriptions of the problems, including test case descriptions. APPS (Hendrycks et al., 2021) and CodeContest (Li et al., 2022a) contain language-agnostic problems in algorithmic competition style and tend to be very challenging. Both datasets expect solutions (complete programs, unlike functions in other datasets) in any language that uses standard input and output to consume and return values. The output is compared directly with the expected values without test cases written in any particular language to test for correctness. In contrast, HumanEval and MBPP use test statements written directly in Python. We show all the dataset formats for comparison in Section R.1.

We find that the HumanEval format aligns best with how programmers would write in a typical coding environment; therefore, we use this format for our converted MBXP benchmark. We also convert the original Python MBPP dataset to be of this format as well for comparison consistency. Our benchmark, MBXP, is the first execution-based function completion benchmark available in multiple languages for all (or most) tasks in parallel.

**Code Translation Resources**  Several works in the literature have developed parallel corpus to facilitate source code translation. Earlier works (Nguyen et al., 2013; Karaivanov et al., 2014) focused on building semi-automatic tools to find similar functions in Java and C# from open source projects. Subsequent works used libraries and transcompilers to construct parallel corpora in Python 2 and Python 3 (Aggarwal et al., 2015), and CoffeeScript and JavaScript (Chen et al., 2018). Among the recent works, Lachaux et al. (2020a) collected a corpus of parallel functions in Java, Python, and C++ from `GeeksforGeeks` and provided unit tests for execution-based evaluation. Very recently, Szafraniec et al. (2022) extended the dataset in Go and Rust languages. On a similar line, Zhu et al. (2022) introduce a new dataset which is parallel across 7 programming languages on both snippet level and program level based on `GeeksforGeeks` data. Another work (Ahmad et al., 2021b) aggregated a comparatively larger parallel corpus in Java and Python by collecting programming problem solutions from several sources. Different from the prior works, our proposed dataset, MBXP, covers a wide range of languages with unit tests to facilitate the evaluation of functional accuracy of ML-based code translation models.

**Multi-lingual Evaluation of Code Generation Models**  Wang et al. (2022) proposed MCoNaLa, a multi-lingual version of CoNaLa Yin et al. (2018) in various natural languages. This is orthogonal to our work that extends multi-linguality on programming languages. Similar approaches could be applied to MBXP to expand the dataset to multiple natural languages and we leave it as one of the future directions.

## C    EVALUATION SETUP

### C.1    SAMPLE GENERATION

We use nucleus sampling with $p = 0.95$ (Holtzman et al., 2020). For all experiments, limit the input length to be 1792 and generate up to 256 tokens. If the context exceeds 1792 tokens, we perform truncation from the left. Note that the truncation can happen more often in the case of few-shot prompting or translation settings.

### C.2    STOPPING CRITERIA

We categorize languages into groups where each group has the same stopping criteria.

- Curly-brace style with standalone function: JavaScript, TypeScript, Go, Kotlin, PHP, C++, Rust. We truncate right after the closing } character.
- Curly-brace style with function wrapped under class: Java, C#. We truncate right after the closing } and add \n} to close the higher-level class wrapper. This is slightly different from letting the model generate a closing } for the wrapper class. We find that if we do let the model generate a closing } on its own, it can go on to generate another function, which technically should not harm the evaluation, but it can cause the generation to be too long and can hit the maximum token limit. Therefore, we find that it is fair and more efficient to close out the class right away after the current function is generation.
- Other keywords: 'end' for Ruby

Note that it is possible to extend these stopping criteria to include multi-function evaluation, where the first function can refer to other functions that follow. However, it is out of scope for this current paper.

### C.3    CODE EXECUTION

We adapted the `human-eval`[†] repository by OpenAI which provides multi-thread execution-based evaluation framework in Python along with unbiased pass@k calculation. Our adaptation supports execution in all languages in MBXP where we use Python's `subprocess` to execute native command in each respective language. For instance, we execute with `node file.js` for JavaScript. The test statements for each language are such that exceptions are thrown if the test cases fail. Each task can also fail due to improper generated code that does not parse or compile. We capture the failure or success of each execution via exit code as well as standard error message for further analysis.

---

[†]`https://github.com/openai/human-eval`

# D  EVALUATION RESULTS ON ADDITIONAL DATASETS

## D.1  MULTI-LINGUAL MATHQA

We evaluate our 13B multi-lingual and mono-lingual models on MathQA datasets in different programming languages. The original MathQA dataset was formatted to the Python version by Austin et al. (2021), after which we use our conversion framework to obtain the corresponding data in different languages for analyses. We do not leverage the training and validation sets of MathQA to finetune our models. The purpose of this evaluation is to investigate the generalization capability of our models on complex context, which requires mathematical reasoning. Similar to Section 4.4 and 4.3, we compare the models with respect to adding few-shot prompts, conducting canonical solution translation, as well as the normal function completion.

Table 1: Evaluating pass@100 execution scores (%) on multi-lingual MathQA using sampling with temperature=0.8

| Mode | Model | Param. Size | MathQA-Python | MathQA-Java | MathQA-JS |
|---|---|---|---|---|---|
| | Multi | 672M | N/A | 91.66 | 94.21 |
| Translate | Multi | 13B | N/A | 96.33 | 98.08 |
| | Mono | 13B | N/A | 94.31 | 96.49 |
| | Multi | 672M | 15.61 | 13.54 | 13.54 |
| Few-shot | Multi | 13B | 21.50 | 26.21 | 24.96 |
| | Mono | 13B | 22.78 | 15.29 | 19.33 |
| Normal | Multi | 13B | 13.43 | 18.05 | 10.67 |
| | Mono | 13B | 20.23 | 14.86 | 10.78 |

Specifically, for the few-shot setup, we prepend the first 4 examples in the MathQA training data with their canonical solutions. For MathQA-Python, the canonical solutions are given; we manually adapt the Python solutions to other languages for these four examples. For solution translation, we evaluate the models on Java and JavaScript with the Python canonical solutions given in the context. The mono-lingual models are only evaluated on the MathQA dataset of the same language.

Following findings are summarized below based on Table 1.

- Both multi-lingual and monolingual models perform very well for solution translation (>94%), probably because the solutions required for solving MathQA problems are usually simple mathematical calculations. Converting them to different languages are straightforwards, if python solutions are provided.

- Prepending few-shot examples achieves better performances than normal predictions for both multi-lingual and monolingual models. As illustrated in the MathQA example in Section R.3, the context are significantly different from the training corpus. Involving a few examples from MathQA domain in the context does help alleviate the domain divergence.

- The multi-lingual models do not consistently outperform the monolingual counterparts. This may be due to the nature of MathQA problems that emphasizes the models' mathematical reasoning capability.

Below, we present a correct prediction and a failed case for MathQA-Java examples under the translation mode. The case below demonstrates that the model understands that exponentiation in Python ($a**b$) and is able to translate to `Math.pow(a,b)`.

```
1
2 ---------- Problem: MathQA/2 (Correct prediction)
3 ---------- Python prompt+canonical Solution ----------
4 def problem():
5     """
6     what will be the difference between simple and compound interest at 14 % per annum on a
        sum of rs . 1000 after 4 years ? n0 = 14.0 n1 = 1000.0 n2 = 4.0
7     """
8     n0 = 14.0
9     n1 = 1000.0
10    n2 = 4.0
```

```
11      t0 = n0 / 100.0
12      t1 = t0 + 1.0
13      t2 = n1 * t0
14      t3 = n2 * t2
15      t4 = t1**min(n2, 5)
16      t5 = n1 * t4
17      t6 = t5 - n1
18      answer = t6 - t3
19      return answer
```

```
1  ---------- Java prompt + Translation result ----------
2
3  import java.io.*;
4  import java.lang.*;
5  import java.util.*;
6  import java.math.*;
7
8
9  class Problem {
10     /**
11      * what will be the difference between simple and compound interest at 14 % per annum on a
         sum of rs . 1000 after 4 years ? n0 = 14.0 n1 = 1000.0 n2 = 4.0
12      */
13     public static double problem() {
14         // n0 = 14.0
15         // n1 = 1000.0
16         // n2 = 4.0
17         double n0 = 14.0;
18         double n1 = 1000.0;
19         double n2 = 4.0;
20
21         double t0 = n0 / 100.0;
22         double t1 = t0 + 1.0;
23         double t2 = n1 * t0;
24         double t3 = n2 * t2;
25         double t4 = Math.pow(t1, Math.min(n2, 5));
26         double t5 = n1 * t4;
27         double t6 = t5 - n1;
28         double answer = t6 - t3;
29
30         return answer;
31
32     }
33 }
```

Illustrated by the failed example below, despite the good overall performance, the model some-times fails to translate mathematical built-in functions from Python to Java (eg. **max** in Python vs. **Math.max** in Java). Additionally, **math.log** in Python can take a second argument for logarithmic base, while **Math.log** in Java specifically means natural logarithm, taking only one argument. The translation model ignores this difference.

```
1  ---------- Problem: MathQA/1822 (Wrong prediction)
2  ---------- Python prompt+canonical Solution ----------
3  def problem():
4      """
5      find the least number of complete years in which a sum of money put out at 45 % compound
        interest will be more than double of itself ? n0 = 45.0
6      """
7      import math
8      n0 = 45.0
9      t0 = n0 / 100.0
10     t1 = math.log(max(1e-5, 2.0), 2)
11     t2 = t0 + 1.0
12     t3 = math.log(max(1e-5, t2), 2)
13     t4 = t1 / t3
14     t5 = t4 + 1.0
15     answer = math.floor(t5)
16     return answer
```

```
1  ---------- Java prompt+translation result ----------
2
3  import java.io.*;
4  import java.lang.*;
5  import java.util.*;
6  import java.math.*;
7
8
9  class Problem {
```

```
10      /**
11       * find the least number of complete years in which a sum of money put out at 45 %
         compound interest will be more than double of itself ? n0 = 45.0
12       */
13      public static double problem() {
14          double n0 = 45.0;
15          double t0 = n0 / 100.0;
16          double t1 = Math.log(max(1e-5, 2.0), 2);
17          // Error: cannot find symbol "max". Also, math.log in Python can take a second
         argument for logarithmic base, while Math.log in Java specifically means natural
         logarithm, taking only one argument.
18          double t2 = t0 + 1.0;
19          double t3 = Math.log(max(1e-5, t2), 2); // Error
20          double t4 = t1 / t3;
21          double t5 = t4 + 1.0;
22          int answer = (int) Math.floor(t5);
23          return answer;
24      }
25  }
```

## D.2 Multi-lingual HumanEval

We present the results on multi-lingual HumanEval in Section M using our models as well as publicly available models. We find that the results on few-shot prompting and translation are generally consistent with MBXP. Details on multi-lingual HumanEval dataset preparation can be found in Section R.2

# E  LANGUAGE "SPILLOVER" IN TRAINING DATA

Our evaluation indicates that code generation models typically have out-of-domain generalization performance (see Section 4.2). We hypothesize that it is due to the effect of data spillover that are quite common especially in cross-lingual code projects where each file can have multiple languages present. In this section, we provide discussion and examples of such cross-lingual code occurrences.

## E.1  TYPES OF CROSS-LANGUAGE DATA SPILLOVER

We provide discussion on types of data observed for code in the wild where multiple languages can co-occur. In particular, there are four categories:

1. Source code from two programming languages occurring in the same file via explicit language embedding mechanism other than "putting code in strings". There are actually two categories — "deep" and "shallow" embeddings of the guest language into the host language. A good example of this in Python is `https://nyu-cds.github.io/python-numba/05-cuda/` which uses python syntax but does not have the semantics of the corresponding python program.

2. Source code from two programming languages occurring in the same file, where the "guest language" is included in the "host language" via the host language's string type. Most web code will fit in this category, but also stuff like code generators (e.g. `https://github.com/LS-Lab/KeYmaeraX-release/blob/master/keymaerax-webui/src/main/scala/edu/cmu/cs/ls/keymaerax/codegen/CExpression.scala`)

3. Source code from two programming languages occurring in the same project, but always in separate files. This is another potential source of cross-lingual data, but it does not apply to the models trained in our paper since we filter languages per file, not per project.

4. Combinations of programming languages via a Foreign Function Interface, where the host language does not explicitly use any source code from the source language but does, e.g., refer to identifiers or function names in compiled bytecode.

## E.2  EXAMPLE 1: EMBEDDED JAVASCRIPT IN PYTHON FILES

The example below taken from `https://github.com/brython-dev/brython/blob/master/scripts/make_encoding_js.py#L30` shows JavaScript written in Python strings throughout the code file `make_encoding_js.py`.

```python
1  """Create a Javascript script to encode / decode for a specific encoding
2  described in a file available at
3  http://unicode.org/Public/MAPPINGS/VENDORS/MICSFT/WINDOWS/<ENCODING>.TXT
4  """
5
6  import os
7  import re
8  import json
9  import urllib.request
10
11 line_re = re.compile("^(0x[A-Z0-9]+)\s+(0x[A-Z0-9]+)*", re.M)
12
13 tmpl = "http://unicode.org/Public/MAPPINGS/VENDORS/MICSFT/WINDOWS/{}.TXT"
14 encoding = input("Encoding name: ")
15 req = urllib.request.urlopen(tmpl.format(encoding.upper()))
16 data = req.read().decode("ascii")
17
18 root_dir = os.path.dirname(os.path.dirname(__file__))
19 libs_dir = os.path.join(root_dir, "www", "src", "libs")
20 filename = os.path.join(libs_dir, f"encoding_{encoding.lower()}.js")
21 with open(filename, "w", encoding="utf-8") as out:
22     out.write("var _table = [")
23     for line in data.split("\n"):
24         mo = line_re.match(line)
25         if mo:
26             key, value = mo.groups()
27             out.write(f"{key}, {value or -1},")
28     out.write("]\n")
29     out.write("var decoding_table = [],\n    encoding_table = []\n")
```

```
30     out.write("""for(var i = 0, len = _table.length; i < len; i += 2){
31 var value = _table[i + 1]
32 if(value !== null){
33     encoding_table[value] = _table[i]
34 }
35 decoding_table[_table[i]] = _table[i + 1]
36 }
37 $module = {encoding_table, decoding_table}
38 """)
```

A simple search query[‡] on Github can reveal multiple other examples.

### E.3  EXAMPLE 2: JAVA AND PYTHON INTEGRATION AS JYTHON

This example is taken from `https://jython.readthedocs.io/en/latest/JythonAndJavaIntegration/` which shows a combination of Java and Python code in a cross-lingual project Jython.

```
1 from org.jython.book.interfaces import CostCalculatorType
2
3 class CostCalculator(CostCalculatorType, object):
4     ''' Cost Calculator Utility '''
5
6     def __init__(self):
7         print 'Initializing'
8         pass
9
10     def calculateCost(self, salePrice, tax):
11         return salePrice + (salePrice * tax)
12
13 package org.jython.book.interfaces;
14
15 public interface CostCalculatorType {
16
17     public double calculateCost(double salePrice, double tax);
18
19 }
20
21 import java.io.IOException;
22 import java.util.logging.Level;
23 import java.util.logging.Logger;
24 import org.plyjy.factory.JythonObjectFactory;
25
26 public class Main {
27
28     public static void main(String[] args) {
29
30         JythonObjectFactory factory = JythonObjectFactory.getInstance();
31         CostCalculatorType costCalc = (CostCalculatorType) factory.createObject(
32                 CostCalculatorType.class, "CostCalculator");
33         System.out.println(costCalc.calculateCost(25.96, .07));
34
35     }
36 }
```

---

[‡] `https://github.com/search?q=var+function+extension%3Apy+language%3APython+language%3APython&type=Code&ref=advsearch&l=Python&l=Python`

# F   EXECUTION-BASED FUNCTION COMPLETION RESULTS

## F.1   PERFORMANCE TREND WITH RESPECT TO MODEL SIZE

Figure 9 shows pass@1, pass@10, and pass@100 for many evaluation datasets in MBXP. We can observe that the trends for pass@k for different $k$ are consistent, but simply different in terms of scale for scores. That is, the observation that multi-lingual models begin to clearly outperform mono-lingual models when the model size becomes sufficiently large holds for any $k$.

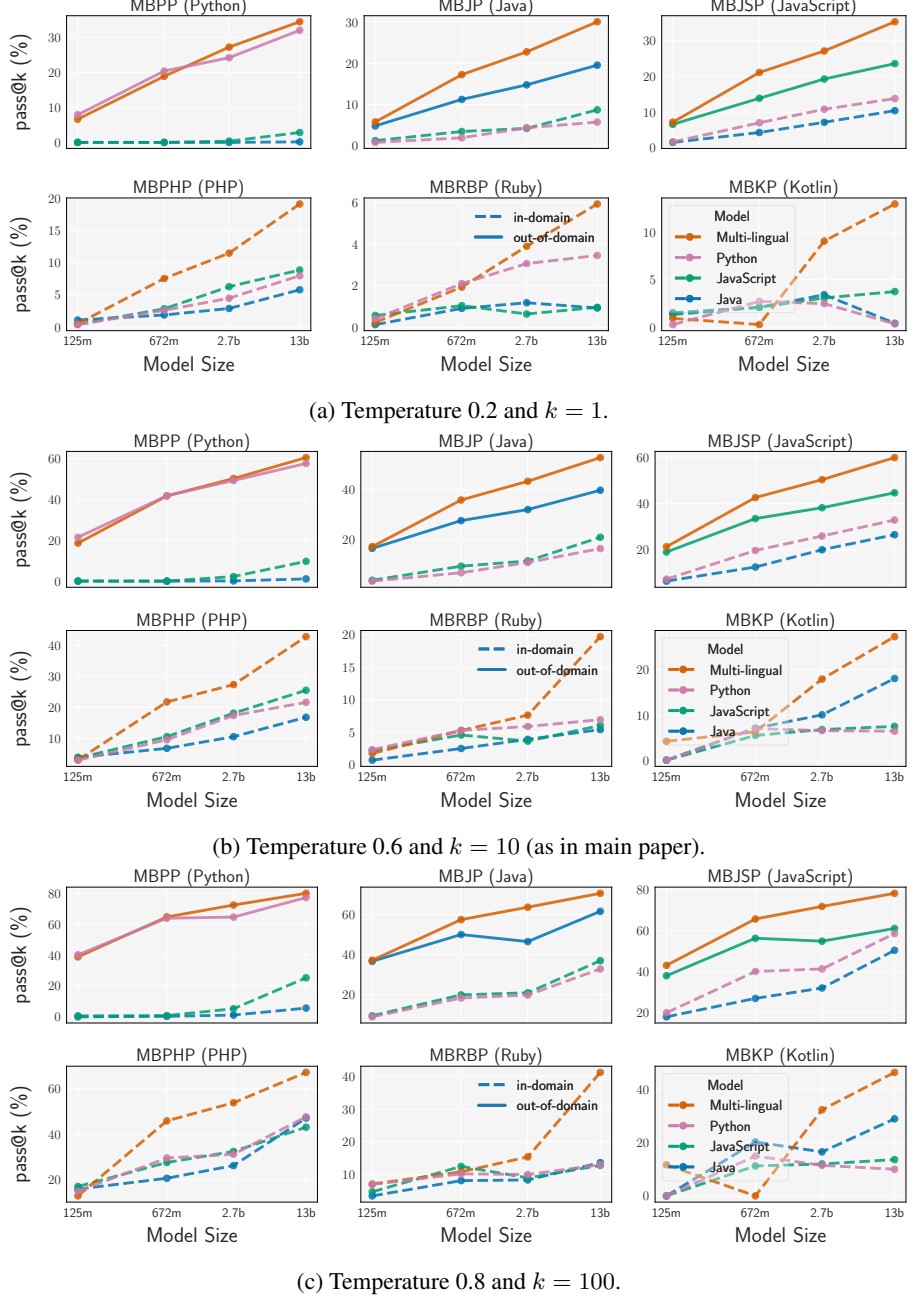

Figure 9: Performance versus model size

## F.2 Comprehensive Sampling Results

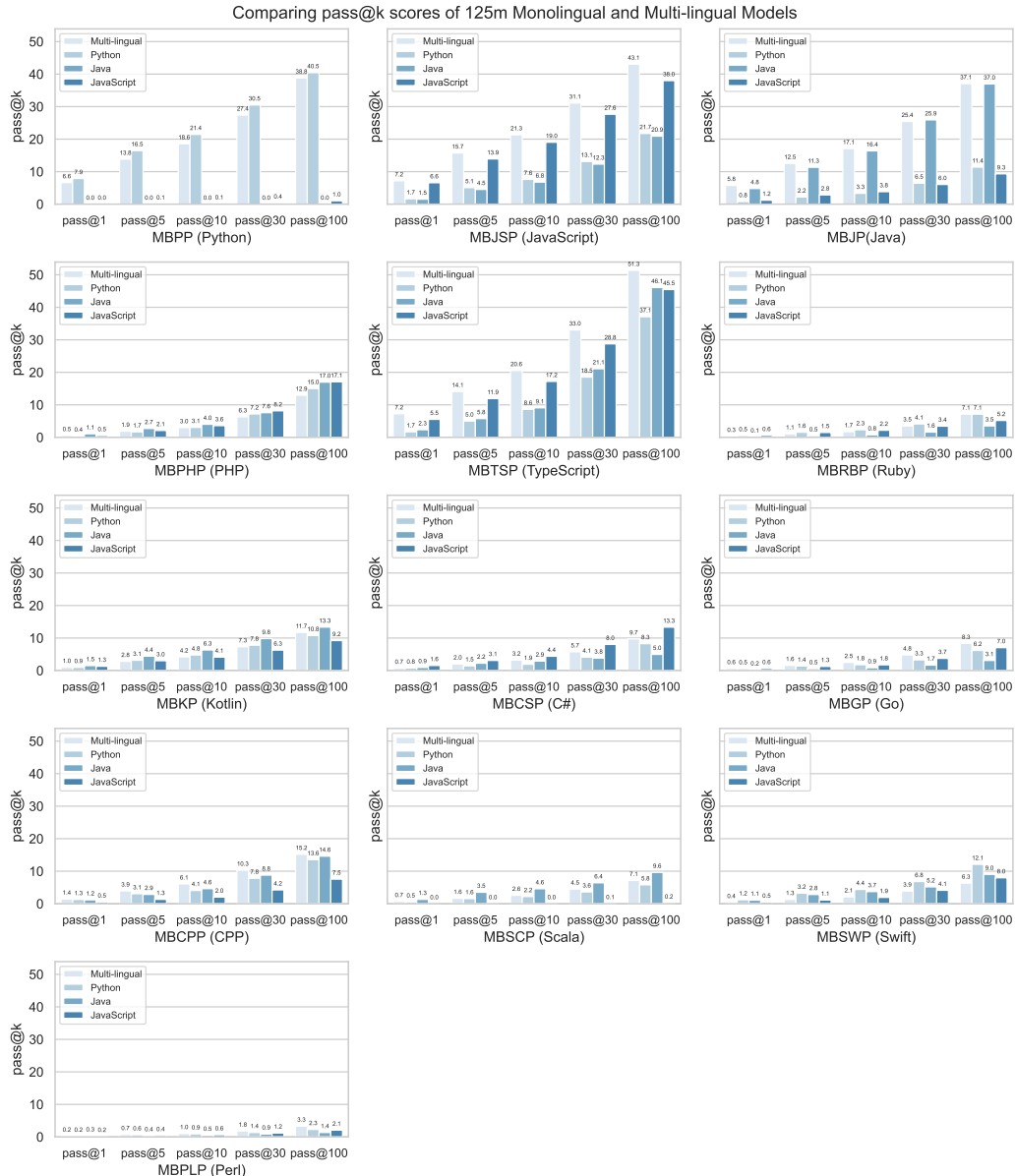

Figure 10: pass@k trends for 125M monlingual and multi-lingual models for in-domain and out-of-domain languages.

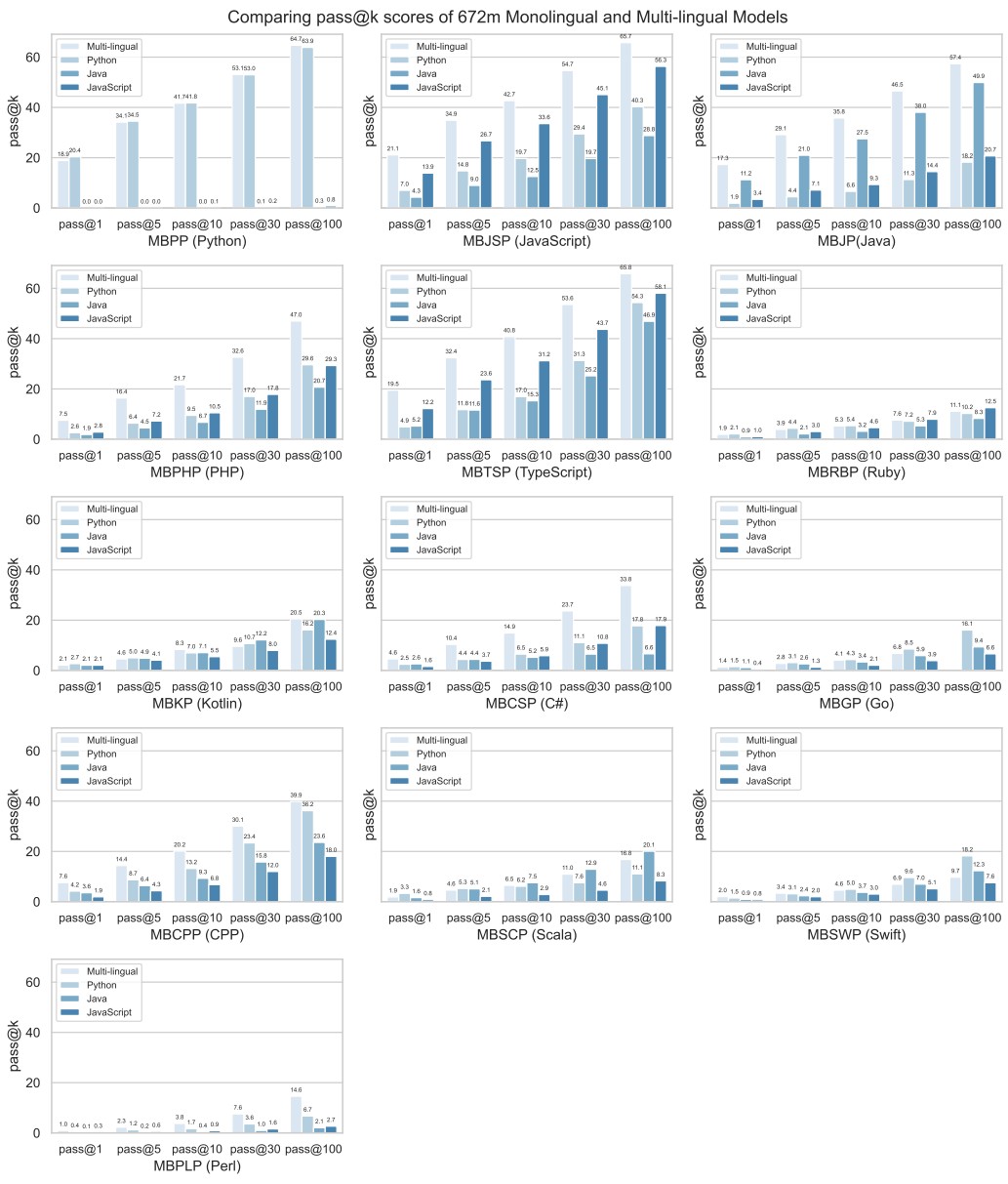

Figure 11: pass@k trends for 672M monlingual and multi-lingual models for in-domain and out-of-domain languages.

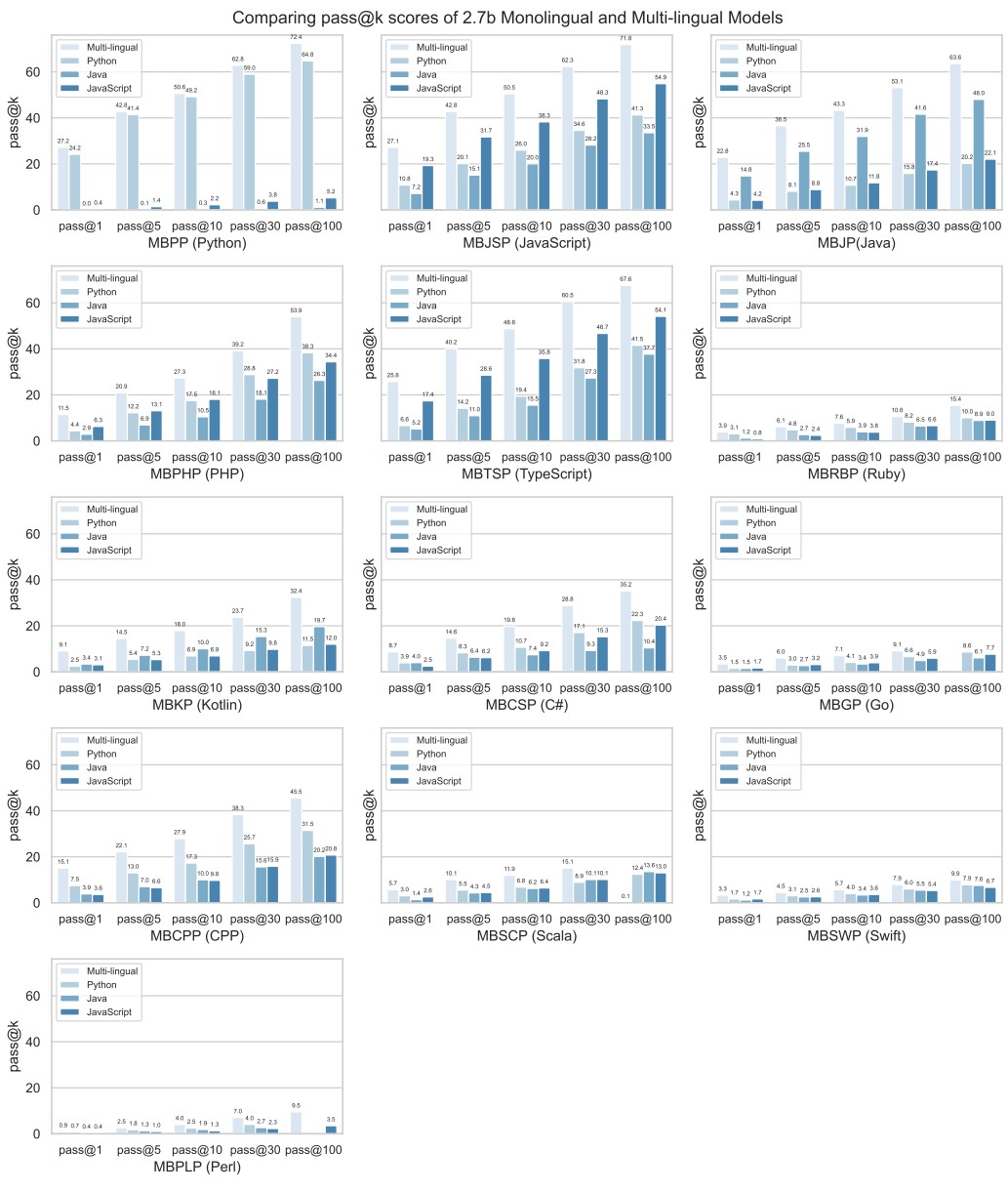

Figure 12: pass@k trends for 2.7B monlingual and multi-lingual models for in-domain and out-of-domain languages.

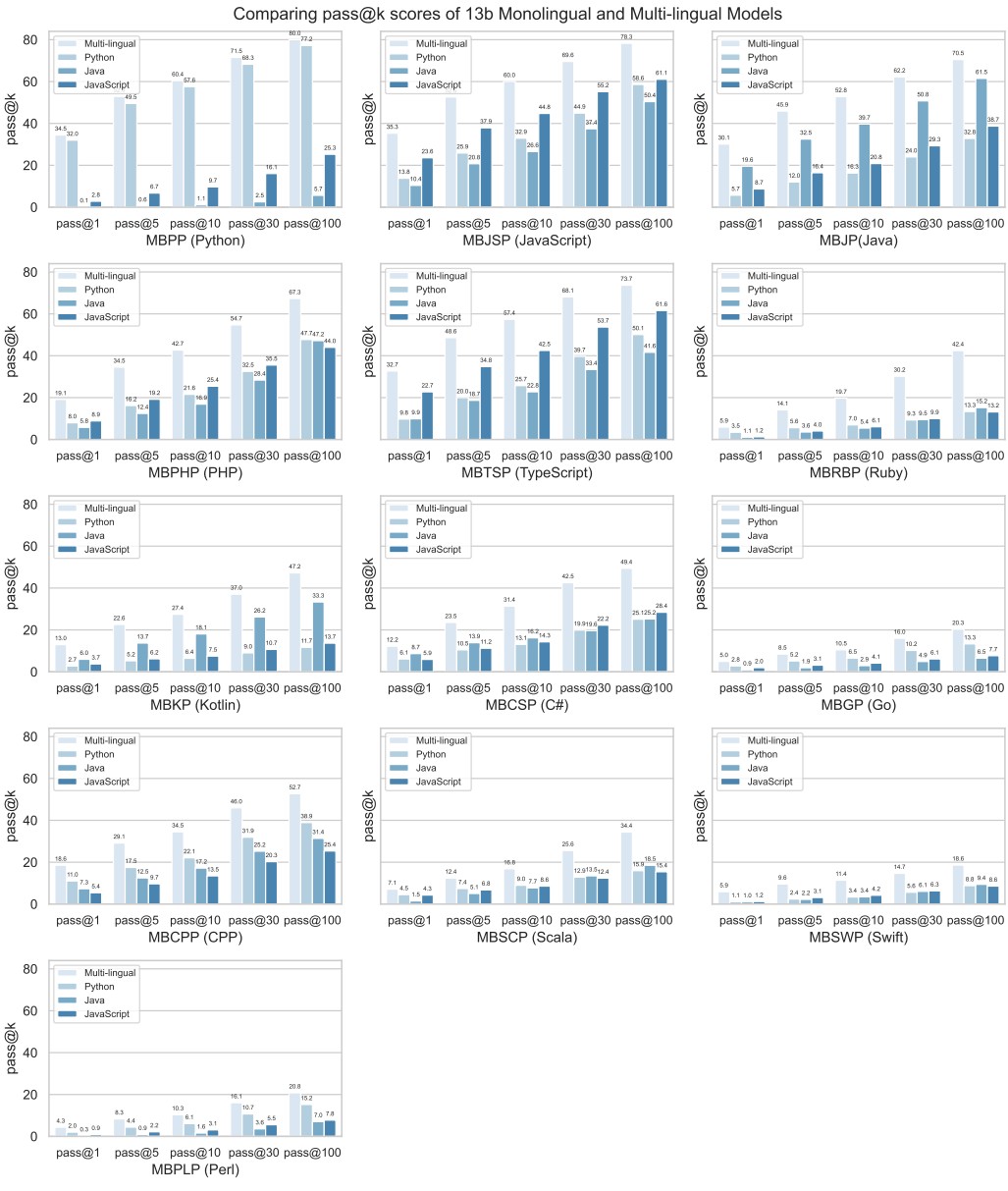

Figure 13: pass@k trends for 13B monlingual and multi-lingual models for in-domain and out-of-domain languages.

## G  FEW-SHOT PROMPTING

In this section, we present more detailed results on few-shot prompting with prompt consisting of three correct functions from the respective MBXP dataset. The few-shot prompts are selected from three correct samples for each language. We note that this gives an automatic performance gain of roughly $0.3\%$ since there are $\approx 1000$ cases for each evaluation language. However, this is quite small compared to the gains observed. We do not tune on the few-shot examples selected; that is, these examples are chosen once and fixed for all usage. It is possible that this can be further tuned, such as the case of prompt engineering in the literature.

### G.1  EVALUATION RESULTS

From Figure 14, we observe that the performance gain is quite clear especially for out-of-domain languages (pass@1 with temperature 0.2). In some cases, there are large performance boosts for mono-lingual models evaluated on out-of-domain languages. For instance, with few-shot prompting, the pass@1 of the 13B Python model evaluated on MBJP increases from $5.7\%$ to $10.3\%$. Similarly, the pass@1 of the 13B multi-lingual model increases from $5.9\%$ to $12.2\%$ with few-shot prompting.

### G.2  QUALITATIVE EXAMPLES

We demonstrate the few-shot prompts for select languages. Each of these prompts precede the function completion prompt for each evaluation.

### G.2.1  PYTHON FEW-SHOT PROMPT

```python
def find_char_long(text):
    """
    Write a function to find all words which are at least 4 characters long in a string by
     using regex.
    >>> find_char_long('Please move back to stream')
    ['Please', 'move', 'back', 'stream']
    >>> find_char_long('Jing Eco and Tech')
    ['Jing', 'Tech']
    >>> find_char_long('Jhingai wulu road Zone 3')
    ['Jhingai', 'wulu', 'road', 'Zone']
    """
    import re

    return re.findall(r"\b\w{4,}\b", text)

def square_nums(nums):
    """
    Write a function to find squares of individual elements in a list using lambda function.
    >>> square_nums([1, 2, 3, 4, 5, 6, 7, 8, 9, 10])
    [1, 4, 9, 16, 25, 36, 49, 64, 81, 100]
    >>> square_nums([10,20,30])
    ([100,400,900])
    >>> square_nums([12,15])
    ([144,225])
    """
    return list(map(lambda x: x**2, nums))

def test_duplicate(arraynums):
    """
    Write a function to find whether a given array of integers contains any duplicate element.
    >>> test_duplicate(([1,2,3,4,5]))
    False
    >>> test_duplicate(([1,2,3,4, 4]))
    True
    >>> test_duplicate([1,1,2,2,3,3,4,4,5])
    True
    """
    if len(arraynums) == len(set(arraynums)):
        return False
    else:
        return True
```

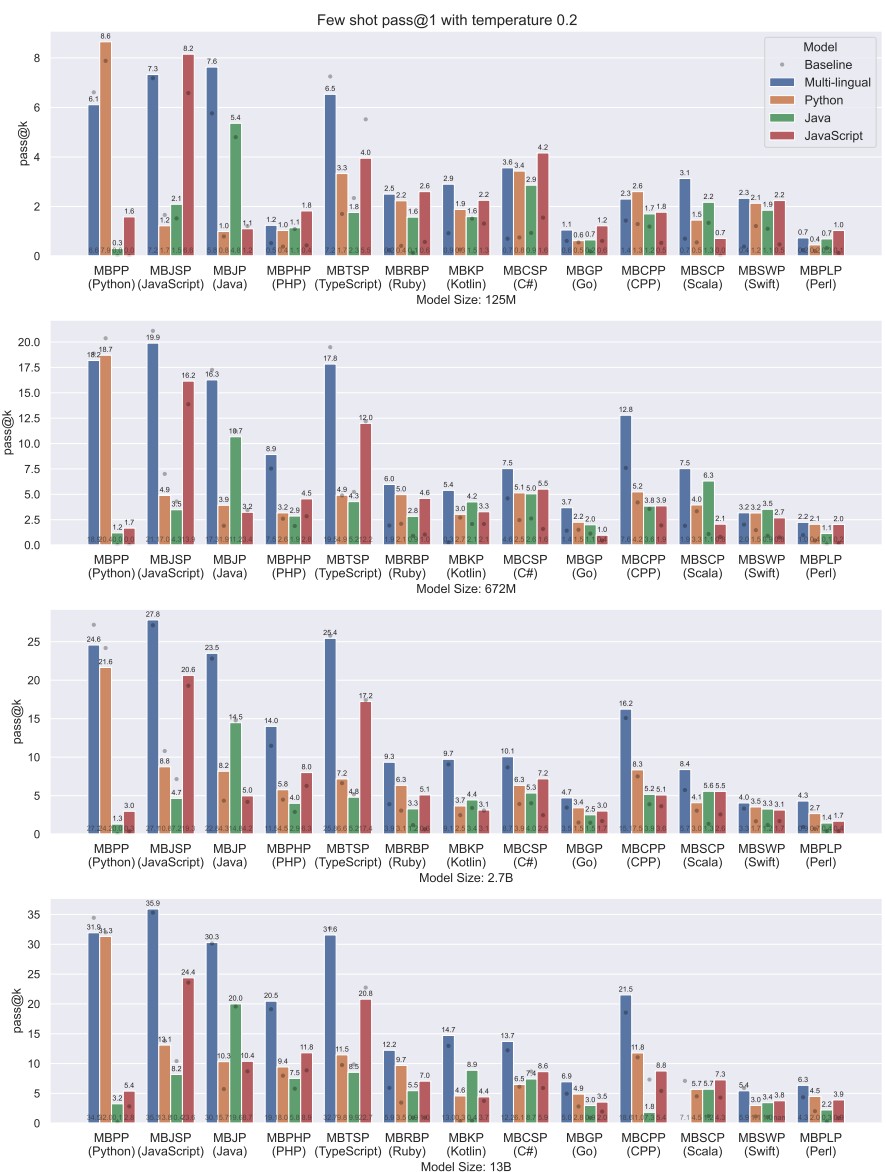

Figure 14: Performance difference due to few-shot prompting (pass@1 with temperature 0.2).

### G.2.2 JAVASCRIPT FEW-SHOT PROMPT

```
1  /**
2   * Write a javascript function to identify non-prime numbers.
3   * > isNotPrime(2)
4   * false
5   * > isNotPrime(10)
6   * true
7   * > isNotPrime(35)
8   * true
9   */
10 function isNotPrime(n) {
11     for (let i = 2; i < n; i++) {
12         if (n % i === 0) {
13             return true;
14         }
15     }
16     return false;
17 }
18
19 /**
20  * Write a javascript function to remove all digits from a list of strings.
21  * > remove(["4words", "3letters", "4digits"])
22  * ["words", "letters", "digits"]
23  * > remove(["28Jan", "12Jan", "11Jan"])
24  * ["Jan", "Jan", "Jan"]
25  * > remove(["wonder1", "wonder2", "wonder3"])
26  * ["wonder", "wonder", "wonder"]
27  */
28 function remove(list) {
29     return list.map(item => item.replace(/\d+/g, ''));
30 }
31
32 /**
33  * Write a javascript function to determine whether all the numbers are different from each
        other are not.
34  * > testDistinct([1, 5, 7, 9])
35  * true
36  * > testDistinct([2, 4, 5, 5, 7, 9])
37  * false
38  * > testDistinct([1, 2, 3])
39  * true
40  */
41 function testDistinct(data) {
42   return data.every((item, index) => {
43     return data.indexOf(item) === index;
44   });
45 }
```

### G.2.3 PHP FEW-SHOT PROMPT

```
1  <?php
2
3  /**
4   * You are an expert PHP programmer, and here is your task.
5   * Write a function of recursion list sum.
6   * php > recursiveListSum([1, 2, [3, 4], [5, 6]])
7   * 21
8   * php > recursiveListSum([7, 10, [15, 14], [19, 41]])
9   * 106
10  * php > recursiveListSum([10, 20, [30, 40], [50, 60]])
11  * 210
12  */
13 function recursiveListSum($dataList) {
14     $sum = 0;
15     $i = 0;
16     foreach ($dataList as $item) {
17         if (is_array($item)) {
18             $sum += recursiveListSum($item);
19         } else {
20             $sum += $item;
21         }
22     }
23     return $sum;
24 }
25
26 /**
27  * You are an expert PHP programmer, and here is your task.
28  * Write function to find the sum of all items in the given dictionary.
```

```
29   * php > returnSum(["a" => 100, "b" => 200, "c" => 300])
30   * 600
31   * php > returnSum(["a" => 25, "b" => 18, "c" => 45])
32   * 88
33   * php > returnSum(["a" => 36, "b" => 39, "c" => 49])
34   * 124
35   */
36  function returnSum($dict) {
37      $sum = 0;
38      foreach ($dict as $key => $value) {
39          $sum += $value;
40      }
41      return $sum;
42  }
43
44  /**
45   * You are an expert PHP programmer, and here is your task.
46   * Write a function to find squares of individual elements in a list using lambda function.
47   * php > squareNums([1, 2, 3, 4, 5, 6, 7, 8, 9, 10])
48   * [1, 4, 9, 16, 25, 36, 49, 64, 81, 100]
49   * php > squareNums([10, 20, 30])
50   * [100, 400, 900]
51   * php > squareNums([12, 15])
52   * [144, 225]
53   */
54  function squareNums($nums) {
55      $squares = [];
56      foreach ($nums as $num) {
57          $squares[] = $num * $num;
58      }
59      return $squares;
60  }
61
62  ?>
63  
```

## G.2.4  RUBY FEW-SHOT PROMPT

```
1   ##
2   # You are an expert Ruby programmer, and here is your task.
3   # Write a Ruby function to remove all digits from a list of strings.
4   # irb> remove(["4words", "3letters", "4digits"])
5   # => ["words", "letters", "digits"]
6   # irb> remove(["28Jan", "12Jan", "11Jan"])
7   # => ["Jan", "Jan", "Jan"]
8   # irb> remove(["wonder1", "wonder2", "wonder3"])
9   # => ["wonder", "wonder", "wonder"]
10  def remove(list)
11    return list.map { |word| word.gsub(/\d+/, '') }
12  end
13
14  ##
15  # You are an expert Ruby programmer, and here is your task.
16  # Write a Ruby function to remove even numbers from a given list.
17  # irb> remove_even([1, 3, 5, 2])
18  # => [1, 3, 5]
19  # irb> remove_even([5, 6, 7])
20  # => [5, 7]
21  # irb> remove_even([1, 2, 3, 4])
22  # => [1, 3]
23  def remove_even(l)
24    return l.reject {|x| x % 2 == 0}
25  end
26
27  ##
28  # You are an expert Ruby programmer, and here is your task.
29  # Write a Ruby function to find the minimum of two numbers.
30  # irb> minimum(1, 2)
31  # => 1
32  # irb> minimum(-5, -4)
33  # => -5
34  # irb> minimum(0, 0)
35  # => 0
36  def minimum(a, b)
37    return a < b ? a : b
38  end
```

# H  TRANSLATION

## H.1  TRANSLATION RESULTS FROM VARIOUS LANGUAGE SOURCES

In this section, we show translation results using (1) multi-lingual and mono-lingual models of various scales and (2) three different languages as source solutions (Python, Java, JavaScript). We note that the canonical solutions from Java and JavaScript are from the data bootstrapping using a separately trained model, as detailed in Section P. For tasks that we do not have solutions for, we do not prepend anything to the usual target-language prompt.

While the training data can potentially consist of translation-like data which allow the model to perform zero-shot translation, we do not know the volume of such translation-related data and suspect such volume to be low. In addition, the model is not trained specifically on certain languages such as Kotlin or PHP for multi-lingual models, or even on Java for Python-only models, for instance.

Figures 15 illustrate the zero-shot translation results for multi-lingual, Python, JavaScript and Java mono-lingual models respectively. In most settings, we observe improvements due to zero-shot translation over the baseline.

**Out-of-domain evaluation languages benefit more from translation**  We can observe consistent performance gains due to translation as opposed to without using reference solution. The performance gain is drastic in certain cases. For example, for Ruby, the 13B multi-lingual model obtains 5.9% pass@1 in the normal mode and 15.9% in the translation mode with JavaScript as a source language, or for PHP, the performance improvement is from $19.1\%$ to $46.5\%$ with Java as a source language.

**Effects of language compatibility or affinity for zero-shot translation**  Based on the trends of performance gains from the translation settings, we observe that different source languages have unequal effects as reference solutions. For instance, based on the multi-lingual 672M and 13B models, Java is the source language that yields the highest performance for MBPHP, whereas JavaScript seems to be the best for MBRBP and MBKP. These compatibility trends can change slightly but are roughly consistent. For instance, for MBJSP, JavaScript is the best source language for the 13B JavaScript and Java monolingual models, whereas Python is the best source language for MBJSP in many other settings. However, for MBKP and MBRBP, JavaScript consistently is the best source language across all model types. We summarize the best model types for each evaluation set below in Table 2. We observe that it is not necessarily the source languages that are closest in syntax that is the best source language, since it has potential to confuse the models during translation and lead the model to generate in an incoorect syntax.

Table 2: Source language that yields the best zero-shot translation scores for each evaluation language

| Evaluation Dataset | Model Type | | | |
|---|---|---|---|---|
| | Multi-lingual | Python | JavaScript | Java |
| MBPP | None or Java | Java | Java | Python |
| MBJP | Python | Python | None or Python | Python |
| MBJSP | Python or Java | Python | Java | Java |
| MBPHP | Java | Python | Java | Java or JavaScript |
| MHRBP | JavaScript | JavaScript | JavaScript | JavaScript |
| MBKP | JavaScript | JavaScript or Python | JavaScript | JavaScript |

We provide some examples in Section H.3.

**Mono-lingual versus multi-lingual models**  For mono-lingual models, we observe large performance boost, partly due to mono-lingual models not performing well for baseline to start with.

**Trends with respect to model sizes**  Larger models typically perform better, as observed in the normal code completion case and also in the translation case as well.

**Model knowledge of source language versus target language** It is likely the case that the knowledge of the target language is more important than the source language for translation performance. We note that Python model obtains high scores with translation on MBJSP with Python as source (13.8% → 30.7%). JavaScript model also obtains high scores with translation on MBJSP with Python as source, with better performance compared to Python model, which is in part due to better baseline performance to start with (23.3% → 32.8%).

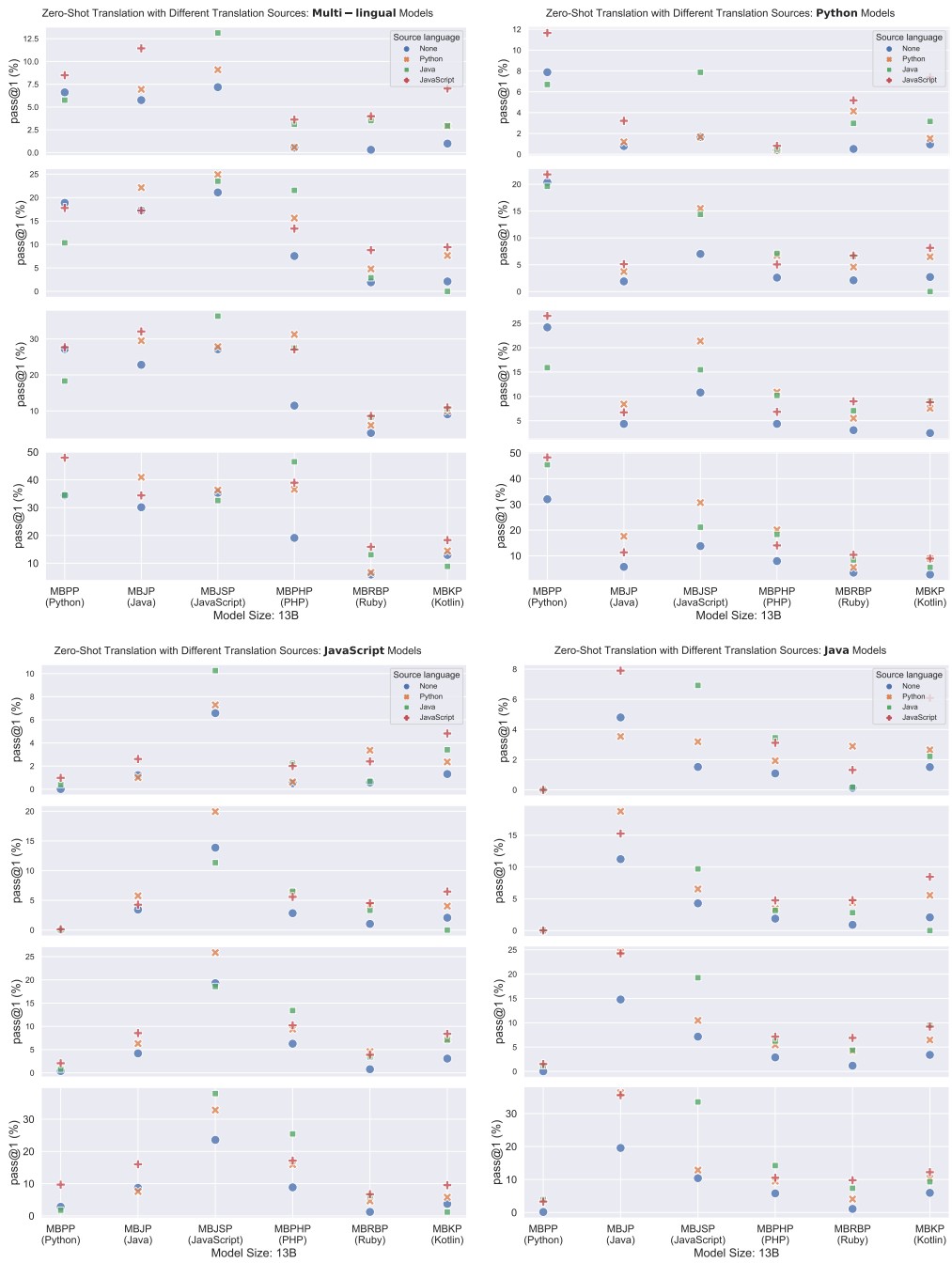

Figure 15: Zero-Shot Translation with Translation Sources from Different Languages: Multi-lingual Models

## H.2 COMPARING TRANSLATION PERFORMANCE OF MULTI-LINGUAL AND MONO-LINGUAL MODELS

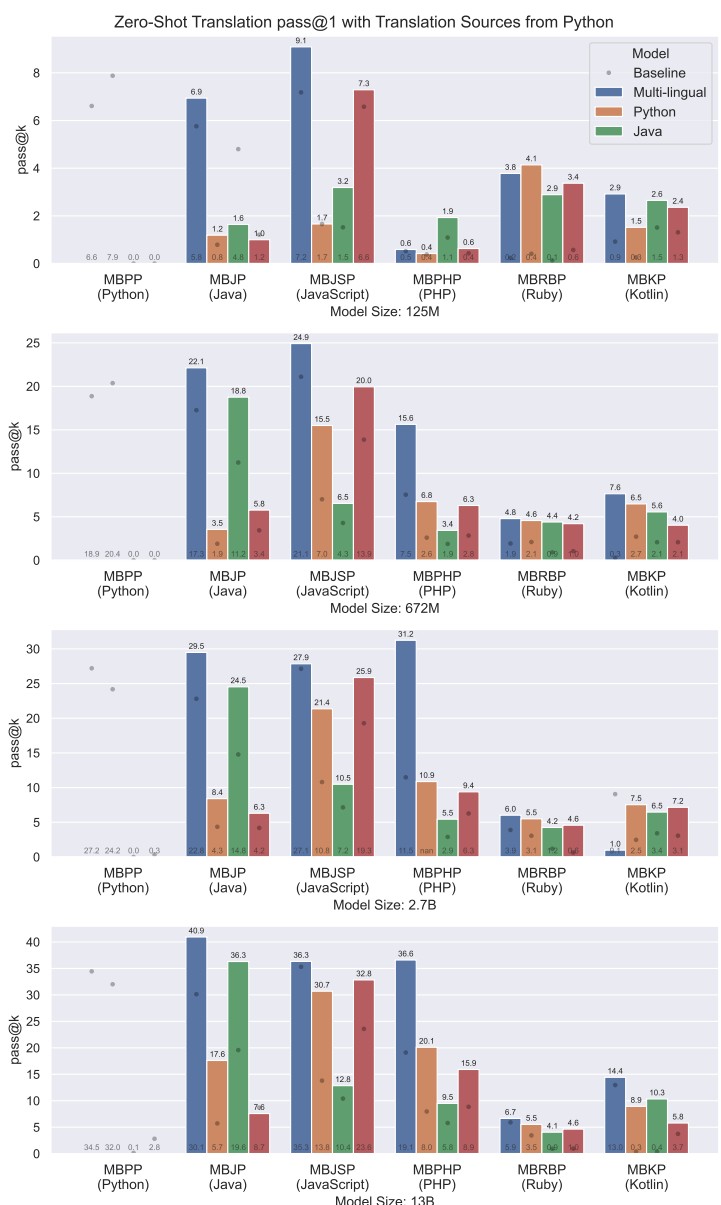

Figure 16: Translation performance compared to baseline (dot) for multi- and mono-lingual models, with Python as a source language.

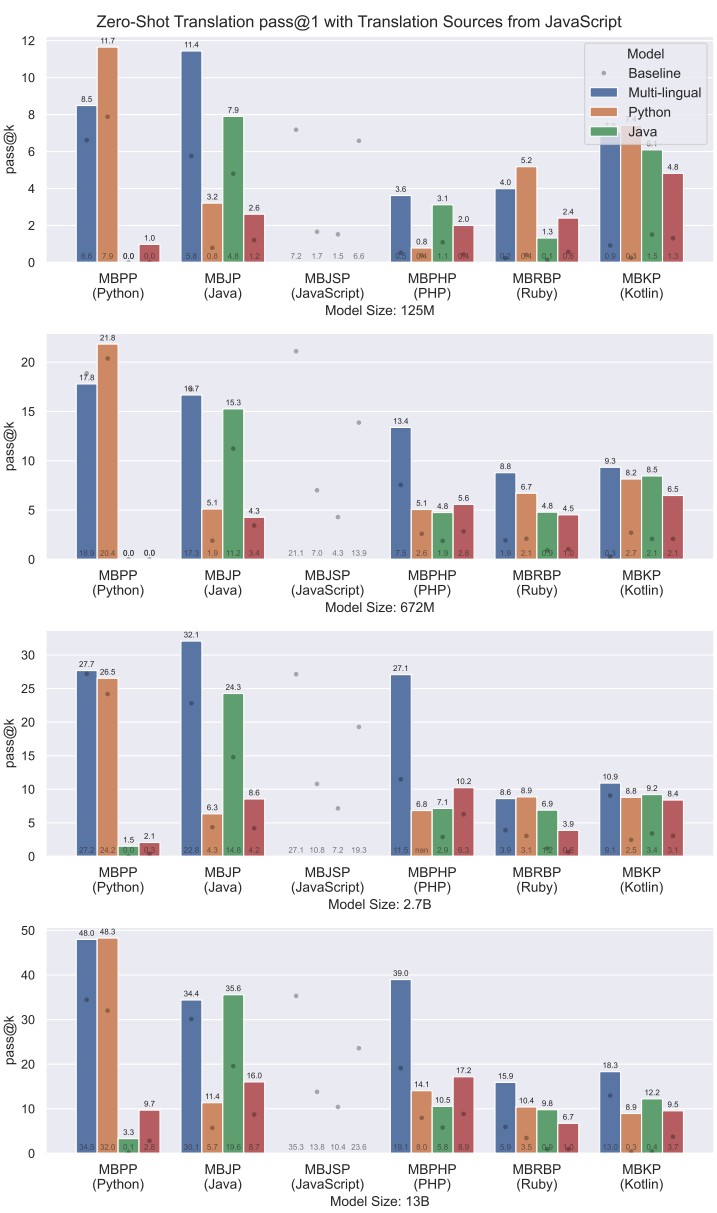

Figure 17: Translation performance compared to baseline (dot) for multi- and mono-lingual models, with JavaScript as a source language.

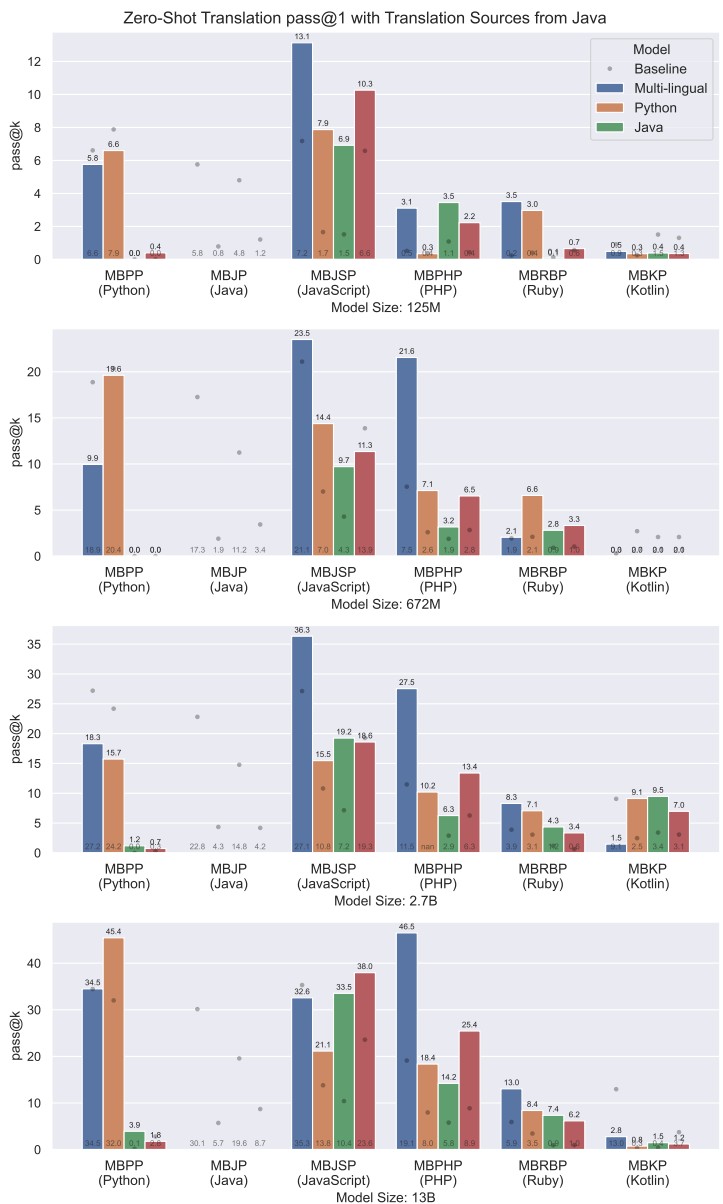

Figure 18: Translation performance compared to baseline (dot) for multi- and mono-lingual models, with Java as a source language.

### H.3 GENERATED TRANSLATION EXAMPLES

#### H.3.1 MBPHP (PHP)

```
1  def find_Sum(arr,n):
2      """
3      Write a python function to find the sum of repeated elements in a given array.
4      >>> find_Sum([1,2,3,1,1,4,5,6],8)
5      3
6      >>> find_Sum([1,2,3,1,1],5)
7      3
8      >>> find_Sum([1,1,2],3)
9      2
10     """
11
12     return sum([x for x in arr if arr.count(x) > 1])
13
14 <?php
15
16 /**
17  * You are an expert PHP programmer, and here is your task.
18  * Write a PHP function to find the sum of repeated elements in a given array.
19  * php > findSum([1, 2, 3, 1, 1, 4, 5, 6], 8)
20  * 3
21  * php > findSum([1, 2, 3, 1, 1], 5)
22  * 3
23  * php > findSum([1, 1, 2], 3)
24  * 2
25  */
26 function findSum($arr, $n) {
27     $count = array_count_values($arr);
28     return array_sum(array_filter($arr, function ($value) use ($count) {
29         return $count[$value] > 1;
30     }));
31 }
```

```
1  def find_Product(arr,n):
2      """
3      Write a python function to find the product of non-repeated elements in a given array.
4      >>> find_Product([1,1,2,3],4)
5      6
6      >>> find_Product([1,2,3,1,1],5)
7      6
8      >>> find_Product([1,1,4,5,6],5)
9      120
10     """
11
12     arr.sort()
13     prod = 1
14     for i in range(0,n,1):
15         if (arr[i - 1] != arr[i]):
16             prod = prod * arr[i]
17     return prod;
18
19 <?php
20
21 /**
22  * You are an expert PHP programmer, and here is your task.
23  * Write a PHP function to find the product of non-repeated elements in a given array.
24  * php > findProduct([1, 1, 2, 3], 4)
25  * 6
26  * php > findProduct([1, 2, 3, 1, 1], 5)
27  * 6
28  * php > findProduct([1, 1, 4, 5, 6], 5)
29  * 120
30  */
31 function findProduct($arr, $n) {
32     $arr = array_unique($arr);
33     $prod = 1;
34     foreach($arr as $key => $value) {
35         if($key == 0 || $value != $arr[$key - 1]) {
36             $prod = $prod * $value;
37         }
38     }
39     return $prod;
40 }
```

#### H.3.2 MBRBP (RUBY)

```
 1  /**
 2   * Write a javascript function to find the element occurring odd number of times.
 3   * > getOddOccurrence([1, 2, 3, 1, 2, 3, 1], 7)
 4   * 1
 5   * > getOddOccurrence([1, 2, 3, 2, 3, 1, 3], 7)
 6   * 3
 7   * > getOddOccurrence([2, 3, 5, 4, 5, 2, 4, 3, 5, 2, 4, 4, 2], 13)
 8   * 5
 9   */
10  function getOddOccurrence(arr, arrSize) {
11
12      for i in range(0, arr_size):
13          count = 0
14          for j in range(0, arr_size):
15              if arr[i] == arr[j]:
16                  count += 1
17          if count % 2 != 0:
18              return arr[i]
19      return -1
20
21
22  ##
23  # You are an expert Ruby programmer, and here is your task.
24  # Write a Ruby function to find the element occurring odd number of times.
25  # irb> get_odd_occurrence([1, 2, 3, 1, 2, 3, 1], 7)
26  # => 1
27  # irb> get_odd_occurrence([1, 2, 3, 2, 3, 1, 3], 7)
28  # => 3
29  # irb> get_odd_occurrence([2, 3, 5, 4, 5, 2, 4, 3, 5, 2, 4, 4, 2], 13)
30  # => 5
31
32  def get_odd_occurrence(arr, arr_size)
33    occurrences = Hash.new
34    for i in 0...arr_size
35      occurrences[arr[i]] ||= 0
36      occurrences[arr[i]] += 1
37    end
38    result = nil
39    occurrences.each_pair do |k, v|
40      if v % 2 != 0 && result == nil
41        result = k
42      end
43    end
44    result
45
46  end
```

### H.3.3  MBKP (KOTLIN)

```
 1  def sort_matrix(M):
 2      """
 3      Write a function to sort a given matrix in ascending order according to the sum of its
        rows.
 4      >>> sort_matrix([[1, 2, 3], [2, 4, 5], [1, 1, 1]])
 5      [[1, 1, 1], [1, 2, 3], [2, 4, 5]]
 6      >>> sort_matrix([[1, 2, 3], [-2, 4, -5], [1, -1, 1]])
 7      [[-2, 4, -5], [1, -1, 1], [1, 2, 3]]
 8      >>> sort_matrix([[5,8,9],[6,4,3],[2,1,4]])
 9      [[2, 1, 4], [6, 4, 3], [5, 8, 9]]
10      """
11      result = sorted(M, key=sum)
12      return result
13
14  /**
15   * You are an expert Kotlin programmer, and here is your task.
16   * Write a function to sort a given matrix in ascending order according to the sum of its rows
        .
17   * >>> sortMatrix([[1, 2, 3], [2, 4, 5], [1, 1, 1]])
18   * [[1, 1, 1], [1, 2, 3], [2, 4, 5]]
19   * >>> sortMatrix([[1, 2, 3], [-2, 4, -5], [1, -1, 1]])
20   * [[-2, 4, -5], [1, -1, 1], [1, 2, 3]]
21   * >>> sortMatrix([[5, 8, 9], [6, 4, 3], [2, 1, 4]])
22   * [[2, 1, 4], [6, 4, 3], [5, 8, 9]]
23   */
24  fun sortMatrix(m : List<List<Int>>) : List<List<Int>> {
25      return m.sortedBy { it.sum() }
26  }
```

# I   ANALYSIS: EFFECTS OF FEW-SHOT AND TRANSLATION PROMPTS

Below, we show an extended version of the analysis in main text (Figure 7) where we demonstrate the differences in qualitative behaviors of few-shot prompting versus translation settings and their effects on helping the usual function completion task. The differences in these two modes are simply the prompts the precede the function completion prompt. That is, the few-shot setting uses 3 examples in the corresponding language and the translation mode uses the solution from the same problem in a different language. The translation setting helps the model solve difficult tasks that are very difficult to solve without a reference solution (Figure 21) whereas the the few-shot prompting helps condition the model to generate code properly in that respective syntax (See Section I.1).

## I.1   TEST CASE ERROR VERSUS NON-ASSERTION ERROR

We categorize the failure of each generation code sample into two main categories: assertion or test-based errors versus non-assertion errors, which consist of all other errors such as compile, parsing, or runtime error not related to test cases. We use the results from temperature 0.2 with 30 samples for each problem and calculate the fraction of non-assertion errors over the number of all samples.

The results in Figure 19 show that the few-shot prompting results in lower non-assertion errors for out-of-domain languages, indicating that few-shot prompts help models generate code with more precise syntax in each language. In contrast, there is little effect even evaluated on the in-domain languages, since the models already are fluent in these languages where the additional signals from the few-shot prompts do not help further. For the translation case, interesting we observe higher non-assertion errors on in-domain evaluation.

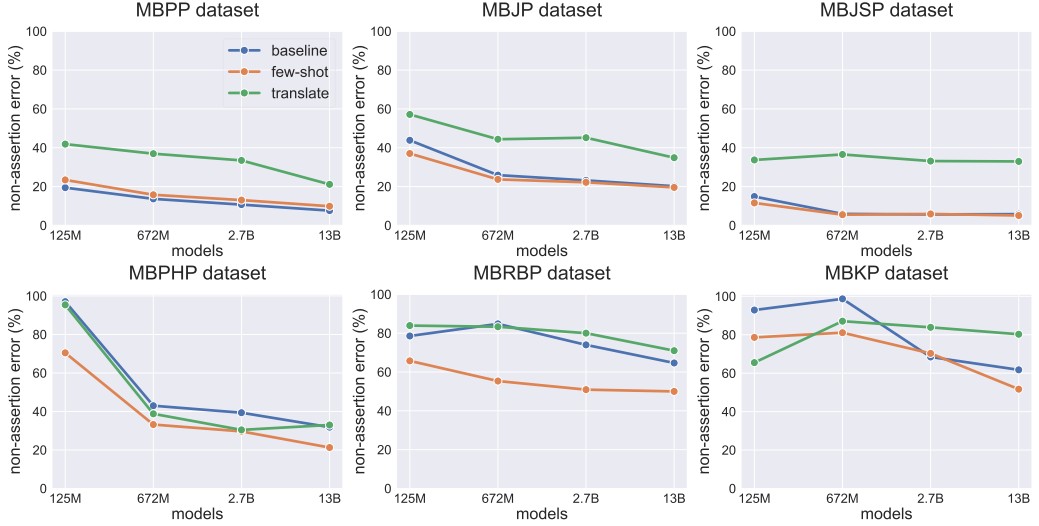

Figure 19: The percentage of test case non-assertion error out of all the error cases on different in-domain (upper) and out-of-domain (lower) datasets.

## I.2   SOLVE RATE PER PROBLEM DUE TO FEW-SHOT PROMPTING AND TRANSLATION

We perform sampling to generate 100 samples with temperature 0.8. For each task id, we calculate the fraction of number of code samples that pass the test over a total of 100 samples. We repeat this experiment for the function completion baseline, few-shot prompting, and translation settings. In Figure 21, we sort the task ids by the solve rate of the function completion baseline, which indicates the difficulty of various tasks in each dataset. We observe differences in how the solve rates for few-shot prompting or translation settings accumulate. For the few-shot case, the accumulation of the solve rates per task revolve near the baseline solve rate, indicating that the difficulty of the problem given the few-shot prompts do not deviate much from the difficulty in the baseline case. However, in the translation case, some tasks ids that correspond to low baseline solve rate have much higher solve rate in the translation case, sometimes with perfect rate 1.0.

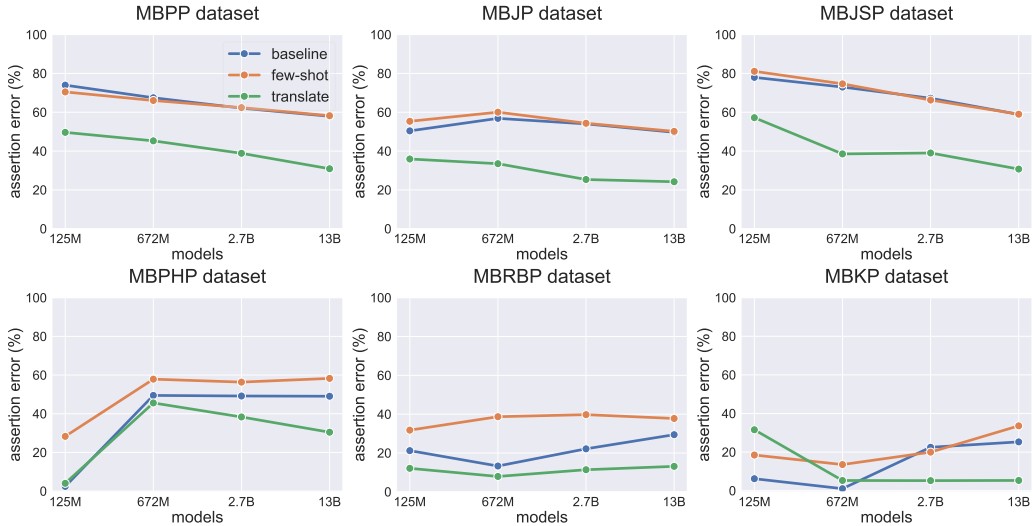

Figure 20: The percentage of test case assertion error out of all the error cases on different in-domain (upper) and out-of-domain (lower) datasets.

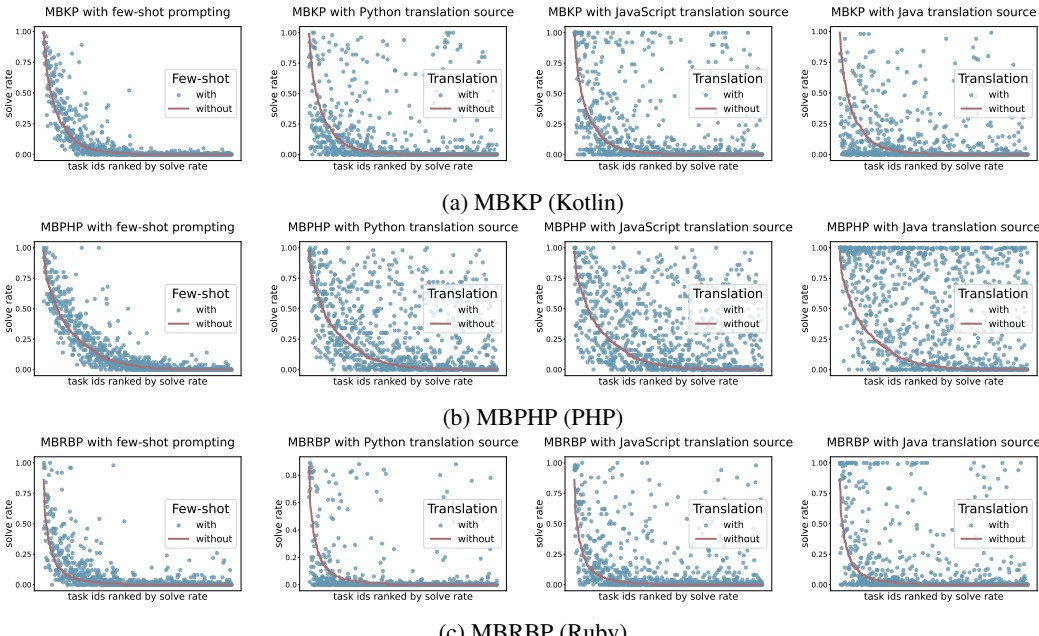

(a) MBKP (Kotlin)

(b) MBPHP (PHP)

(c) MBRBP (Ruby)

Figure 21: For each task, we show a fraction of generations that pass the tests over the total number of samples (solve rate), where the task indices are ranked to show increasing difficulty. In the translation setting, tasks that are previously difficult (low solve rate for the baseline) can become easily solvable, demonstrating that models can leverage reference solutions in the source language to solve hard tasks. In contrast, the solve rates with few-shot prompting do not deviate as much from the baseline solve rate. Translation from different language sources exhibit similar trends where the source solution can help solve hard tasks where we observe unequal effects from different source languages.

Figure 21 also demonstrates consistent effects of source languages. For a language such as Java, we observe that it can help solve many of the hard problems for MBPHP (PHP) evaluation, based on the high concentration of points around solve rate 1.0. This analysis also complements Section H which demonstrates unequal effects of source languages.

## J    ROBUSTNESS EVALUATION: R-MBXP

### J.1    DATASET PREPARATION AND EVALUATION SETUP

Robustness is an important indicator of the reliability of code generation models in practice. Here we provide a robustness benchmark for all the trained models across MBXP datasets. Specifically, we consider three natural data augmentations (1) Paraphrase by Back Translation (Li & Specia, 2019; Sugiyama & Yoshinaga, 2019) (e.g., "create a function" to "write one function") (2) Character Case Change ("Create A FunctioN") and (3) Synonym Substitutions (Miller, 1995) ("generate a function") as basic transformations to perturb the docstrings in prompts. We use the default settings and implementations of these three transformations from NL-Augmenter[§], a standard collection of data augmentations for robustness evaluation on text (Dhole et al., 2021). We select these three transformations since they can mostly maintain the naturalness for the tasks of code generations based on our observations. We then measure the average pass@1 with greedy decoding for all the models on datasets perturbed by each transformation. Here multi-lingual models are trained with multiple languages including Python, Java, Javascript (JS) while mono-lingual models are trained with each language individually. To simplify the comparisons, we call Ruby, PHP, and Kotlin as out-of-domain datasets for all the evaluated models while Python, Java, JS as in-domain datasets.

### J.2    EVALUATION RESULTS

We present the detailed results in Figure 22 and summarize several interesting observations below. (1) The percentages of pass@1 drops on perturbed datasets over regular ones are consistent across different sizes of the models. In specific, the average pass@1 over all the datasets drops from 2.26 to 2.07 for 125M models (8.56% drop), 6.40 to 5.87 for 672M models (8.25% drop), 9.20 to 8.33 for 2.7B models (9.40% drop), and 12.63 to 11.62 (8.02% drop). (2) For in-domain datasets, multi-lingual models have less percentage of performance drops compared to mono-lingual models under perturbations. On average, pass@1 of multi-lingual models drops from 21.36 to 19.73 (7.63% drop) while pass@1 of mono-lingual models drops from 16.90 to 15.24 (9.81% drop). (3) For out-of-domain datasets, multi-lingual models also have less percentage of performance drops compared to mono-lingual models under perturbations. On average, pass@1 of multi-lingual models drops from 6.78 to 6.24 (7.98% drop) while pass@1 of mono-lingual models drops from 2.72 to 2.47 (8.97% drop).

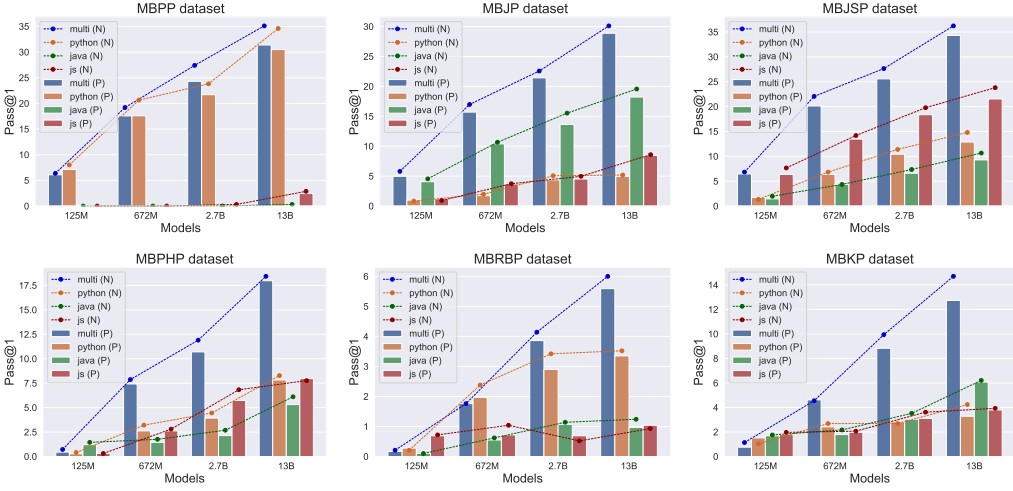

Figure 22: Pass@1 measured on nominal datasets (N) and perturbed datasets (P) across different model sizes for each dataset.

[§]https://github.com/GEM-benchmark/NL-Augmenter

### J.3 QUALITATIVE EXAMPLES

In this subsection, we provide qualitative examples to illustrate the three types of perturbations applied to the datasets. A successful and a failure MBPP sample completions before and after each type of perturbation are provided based on the code completion results by 672M_python model on MBPP dataset. We performed similar perturbations for all the other datasets for robustness evaluation (quantitative results shown in Section J.2).

**MBPP Example 1 for Back Translation Paraphrasing**

```
1  [672M_python completion on normal prompt] (passed: True)
2  def left_Rotate(n,d):
3      """
4      Write a python function to left rotate the bits of a given number.
5      >>> left_Rotate(16,2)
6      64
7      >>> left_Rotate(10,2)
8      40
9      >>> left_Rotate(99,3)
10     792
11     """
12 ### Begin of code completion ###
13     return n << d
14 ### End of code completion ###
15
16
17 [672M_python completion on perturbed prompt] (passed: False)
18 def left_Rotate(n,d):
19     """
20     Write a Python function to rotate the bits of a given number to the left.
21     >>> left_Rotate(16,2)
22     64
23     >>> left_Rotate(10,2)
24     40
25     >>> left_Rotate(99,3)
26     792
27     """
28 ### Begin of code completion ###
29     return n >> d | (n & (1 << d))
30 ### End of code completion ###
```

**MBPP Example 2 for Character Case Changing**

```
1  [672M_python completion on normal prompt] (passed: True)
2  def text_match_string(text):
3   """
4      Write a function that matches a word at the beginning of a string.
5      >>> text_match_string(" python")
6      ('Not matched!')
7      >>> text_match_string("python")
8      ('Found a match!')
9      >>> text_match_string(" lang")
10     ('Not matched!')
11     """
12 ### Begin of code completion ###
13     return "Not matched!" if text.startswith(" ") else "Found a match!"
14 ### End of code completion ###
15
16 [672M_python completion on perturbed prompt] (passed: False)
17 def text_match_string(text):
18     """
19     Write a function that matches a word at the BeginNing of a strinG.
20     >>> text_match_string(" python")
21     ('Not matched!')
22     >>> text_match_string("python")
23     ('Found a match!')
24     >>> text_match_string(" lang")
25     ('Not matched!')
26     """
27 ### Begin of code completion ###
28     return "Not matched!"
29 ### End of code completion ###
```

**MBPP Example 3 for Synonym Substitutions**

```
1  [672M_python completion on normal prompt] (passed: True)
2  def check_value(dict, n):
3      """
```

```
 4      Write a function to check if all values are same in a dictionary.
 5      >>> check_value({'Cierra Vega': 12, 'Alden Cantrell': 12, 'Kierra Gentry': 12, 'Pierre Cox
        ': 12},10)
 6      False
 7      >>> check_value({'Cierra Vega': 12, 'Alden Cantrell': 12, 'Kierra Gentry': 12, 'Pierre Cox
        ': 12},12)
 8      True
 9      >>> check_value({'Cierra Vega': 12, 'Alden Cantrell': 12, 'Kierra Gentry': 12, 'Pierre Cox
        ': 12},5)
10      False
11      """
12 ### Begin of code completion ###
13      for key in dict:
14          if dict[key] != n:
15              return False
16      return True
17 ### End of code completion ###
18
19 [672M_python completion on perturbed prompt] (passed: False)
20 def check_value(dict, n):
21      """
22      Write a function to check if all values are like in a dictionary.
23      >>> check_value({'Cierra Vega': 12, 'Alden Cantrell': 12, 'Kierra Gentry': 12, 'Pierre Cox
        ': 12},10)
24      False
25      >>> check_value({'Cierra Vega': 12, 'Alden Cantrell': 12, 'Kierra Gentry': 12, 'Pierre Cox
        ': 12},12)
26      True
27      >>> check_value({'Cierra Vega': 12, 'Alden Cantrell': 12, 'Kierra Gentry': 12, 'Pierre Cox
        ': 12},5)
28      False
29      """
30 ### Begin of code completion ###
31      for key in dict:
32          if type(dict[key]) != int:
33              return False
34      return True
35 ### End of code completion ###
```

# K CODE INSERTION: I-MBXP

## K.1 DATASET PREPARATION

Assume there are $n$ lines of code in canonical solution, we skip problems with $n < 2$ and randomly mask out $m = [1, 8]$ consecutive lines for remaining problems. For each problem, we run multiple times (say, 1 if $n < 5$, 2 if $n < 12$, otherwise 3) to generate variants and remove duplicate masks. We report data statistice in Table 3.

Table 3: Data statistics for i-MBxP. We report total number of problems and the number of problems as a function of the number of insertion lines.

|          | total | 0  | 1   | 2   | 3   | 4   | 5   | 6   | 7  |
|----------|-------|----|-----|-----|-----|-----|-----|-----|----|
| **i-MBPP**  | 1489  | 1  | 922 | 246 | 138 | 69  | 62  | 28  | 23 |
| **i-MBJP**  | 1813  | 19 | 805 | 328 | 217 | 172 | 119 | 114 | 39 |
| **i-MBJSP** | 1521  | 38 | 739 | 266 | 159 | 138 | 83  | 71  | 27 |

## K.2 EVALUATION SETUP

We evaluate InCoder-6.7B model (Fried et al., 2022) pretrained on 159 GB of code, 52 GB in Python and 107 GB in other 28 languages, and 57 GB of text content from StackOverflow. We do greedy search and report averaged execution accuracy over all problems. We feed a sequence `left <Mask:0> right <Mask:0>` to the model as prompt, where `left` denotes the right context and `<Mask:0>` denotes a sentinel token of Incoder model. We apply two stopping criteria: (1) `<EOM>` is generated and (2) any token from the predefined list `["\nclass", "\ndef", "\n#", "\nif"]` is generated for Python or braces that close the function scope for Java and Javascript. After that, we match generated tokens with right context and remove duplicated tokens. We compare it against left-right (L-R) baseline where we feed `left` to the model as prompt and follow the same stopping criteria.

## K.3 EVALUATION RESULTS

Results in Table 4 show that right context can significantly boost performance across all languages. We also studied the effect of the number of lines of right context and observed increasing accuracy as we add more lines of right context (in Table 5), which is intuitive since more context is beneficial. Furthermore, qualitative examples (K.4) show that models are able to leverage right context to fill in the blank. In example example 1, given index $j$ and $k$ in the right context, InCoder model can fill in two inner loops L15-L16. Otherwise, it fails to do so. In example 2, given $add$ operator in the right context, the InCoder model can mimic the behavior in L33-L38. Otherwise, model might generate irrelevant operator $remove$. In example 3, given $result[char] = 1$ in the right context, InCoder model generate $result[char]+ = 1$. Otherwise, the model will replace 1 with $countStr[char]$ which results in wrong outputs.

Table 4: Pass@1 accuracy on code insertion datasets: i-MbXP

| Model     | i-MBPP | i-MBJSP | i-MBJP |
|-----------|--------|---------|--------|
| **L-R**       | 30.1   | 48.65   | 41.7   |
| **Insertion** | 37.07  | 55.68   | 57.41  |

Table 5: Pass@1 vs the number of lines of right context.

| dataset  | 0    | 1    | 2    | 3    | ALL   |
|----------|------|------|------|------|-------|
| **i-MBPP** | 30.1 | 32.1 | 35.6 | 36.4 | 37.07 |

## K.4 QUALITATIVE EXAMPLES FOR I-MBXP

The example below shows that the model is able to use the right context information and generate appropriate insertion code that are consistent with the right context. In contrast, the left-to-right

approach does not make use of the right context which leads to a very different implementation that would not be consistent with the right context.

### Example 1: Python insertion mode

```python
def find_triplet_array(A, arr_size, sum):
        """
        Write a function to find if there is a triplet in the array whose sum is equal to a
    given value.
        >>> find_triplet_array([1, 4, 45, 6, 10, 8], 6, 22)
        (4, 10, 8)
        >>> find_triplet_array([12, 3, 5, 2, 6, 9], 6, 24)
        (12, 3, 9)
        >>> find_triplet_array([1, 2, 3, 4, 5], 5, 9)
        (1, 3, 5)
        """

        for i in range( 0, arr_size-2):
### begin of insertion ###
                for j in range( i+1, arr_size-1):
                        for k in range( j+1, arr_size):
### end of insertion ###
                                if A[i] + A[j] + A[k] == sum:
                                        return  A[i],A[j],A[k]
                                        return True
        return False
```

### Example 1: Python left-to-right mode

```python
### begin of left-right ###
                if A[i] + A[i+1] + A[i+2] == sum:
                        return (A[i], A[i+1], A[i+2])

        return None
### end of left-right ###
```

### Example 2: Java insertion mode

```java
import java.io.*;
import java.lang.*;
import java.util.*;

class CountVariable {

    /**
     * Write a function to iterate over elements repeating each as many times as its count.
     * > CountVariable.countVariable(4, 2, 0, -2)
     * ["p", "p", "p", "p", "q", "q"]
     * > CountVariable.countVariable(0, 1, 2, 3)
     * ["q", "r", "r", "s", "s", "s"]
     * > CountVariable.countVariable(11, 15, 12, 23)
     * ["p", "p", "p", "p", "p", "p", "p", "p", "p", "p", "p", "q", "q", "q", "q", "q", "q", "
    q", "q", "q", "q", "q", "q", "q", "q", "q", "r", "r
", "r", "r", "r", "r", "r", "r", "r", "r", "r", "r", "s", "s", "s", "s", "s", "s", "s", "s", "
    s", "s", "s", "s", "s", "s", "s", "s", "s", "s", "s
", "s", "s", "s", "s"]
     */
    public static List<String> countVariable(int a, int b, int c, int d) {
        // Write your code here
        List<String> finalResult = new ArrayList<>();
        List<String> countableList = new ArrayList<>();
        countableList.add("p");
        countableList.add("q");
        countableList.add("r");
        countableList.add("s");

        int countFor = a;
        for (int index = 0; index < countFor; index++) {
            finalResult.add(countableList.get(0));

### begin of insertion ###
        }

        countFor = b;
        for (int index = 0; index < countFor; index++) {
            finalResult.add(countableList.get(1));
        }
### end of insertion ###
        countFor = c;
        for (int index = 0; index < countFor; index++) {
```

```
41              finalResult.add(countableList.get(2));
42          }
43
44          countFor = d;
45          for (int index = 0; index < countFor; index++) {
46              finalResult.add(countableList.get(3));
47          }
48
49          return finalResult;
50      }
51 }
```

### Example 2: left-to-right mode

```
1  ### begin of left-right ###
2              countableList.remove(countableList.get(0));
3          }
4          countFor = b;
5          for (int index = 0; index < countFor; index++) {
6              finalResult.add(countableList.get(0));
7              countableList.remove(countableList.get(0));
8          }
9          countFor = c;
10         for (int index = 0; index < countFor; index++) {
11             finalResult.add(countableList.get(0));
12             countableList.remove(countableList.get(0));
13         }
14         countFor = d;
15         for (int index = 0; index < countFor; index++) {
16             finalResult.add(countableList.get(0));
17             countableList.remove(countableList.get(0));
18         }
19         return finalResult;
20     }
21 }
22 ### end of left-right ###
```

### Example 3: JavaScript insertion mode

```
1  /**
2   * Write a function to find all anagrams of a string in a given list of strings using lambda
         function.
3   * > anagramLambda(["bcda", "abce", "cbda", "cbea", "adcb"], "abcd")
4   * ["bcda", "cbda", "adcb"]
5   * > anagramLambda(["recitals", " python"], "articles")
6   * ["recitals"]
7   * > anagramLambda([" keep", " abcdef", " xyz"], " peek")
8   * [" keep"]
9   */
10 function anagramLambda(texts, str) {
11   const countFunc = (str) => {
12     let result = {};
13     for (let i = 0; i < str.length; i++) {
14       let char = str[i];
15       if (result[char]) {
16         result[char] += 1;
17       } else {
18         result[char] = 1;
19       }
20     }
21     return result;
22   };
23   const countStr = countFunc(str);
24   return texts.filter((word) => {
25     let result = {};
26     for (let i = 0; i < word.length; i++) {
27       let char = word[i];
28
29 ### begin of insertion ###
30       if (result[char]) {
31         result[char] += 1;
32 ### end of insertion ###
33       } else {
34         result[char] = 1;
35       }
36     }
37     return Object.keys(countStr).every((char) => {
38       return countStr[char] === result[char];
39     });
40   });
```

```
41  }
```

**Example 3: JavaScript left-to-right mode**

```
 1  ### begin of left-right ###
 2          if (result[char]) {
 3              result[char] += countStr[char];
 4          } else {
 5              result[char] = countStr[char];
 6          }
 7      }
 8      return Object.values(result).reduce((a, b) => a + b, 0) === Object.values(countStr).reduce
        ((a, b) => a + b, 0);
 9    });
10  }
11  ### end of left-right ###
```

## L  CODE SUMMARIZATION: S-MBXP

### L.1  DATASET PREPARATION AND EVALUATION SETUP

Here we re-purpose the original MBXP datasets for code summarization task. We remove the natural language description from the original prompt and use the function signature and the canonical solution as the model input for code summarization. To induce the model to generation natural language in comments, we design two types of prompt in zero-shot and few-shot setting, respectively.

**Zero-shot Evaluation.** In this setting, we append "The above code writes a " in the format of code comment after the original code prompt. For example, in Python, the appended sequence is "# The above code writes a ". See examples in different languages in Section L.3.

**Few-shot Evaluation.** In this setting, we select three code-summary pairs and prepend them before the original prompt. Examples are shown in Section L.3.

To evaluate the code summarization performance, we use smoothed BLEU score as the metrics following the setting in CodeXGLUE (Lu et al., 2021) that compare the generated outputs with the groundtruth docstrings. In MBXP datasets, the summarizations are short paragraphs with one or two sentences which makes smoothed BLEU score a suitable metrics (Feng et al., 2020).

### L.2  EVALUATION RESULTS

Experimental results are shown in Figure 23 covering Python, JavaScript and Java. Overall, we found that performances are improved along with the increasing of the model size. For example, the BLEU-4 scores on Python language in 13B, 672M and 125M models are 6.07, 5.59, 3.20 under zero-shot settings, and 34.10, 24.72, 20.75 under few-shot settings. We also noticed that multi-lingual models achieve better performances compared with monolingual models trained on individual languages. An interesting observation is that though the monolingual models are trained on a specific language, they can generalize to other languages well when few-shot examples are provided. From the table we can also notice that the improvements brought by few-shot settings are more significant on larger models. Comparing the multi-lingual models and monolingual models under few-shot settings, we found that the multi-lingual models are more robust to the few-shot examples while monolingual models in smaller sizes show unstable performances.

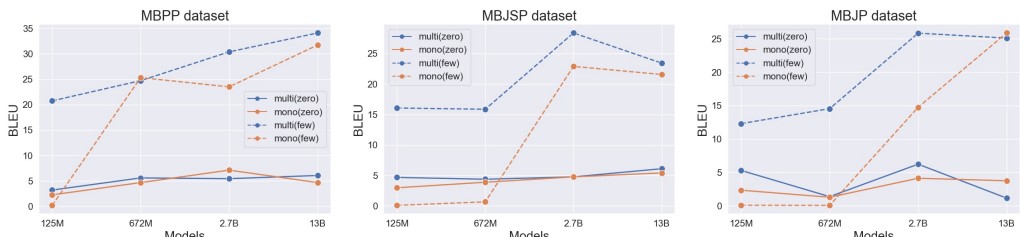

Figure 23: Code summarization evaluation in BLEU scores for all models.

### L.3  QUALITATIVE EXAMPLES

**Zero-shot Prompt Examples.**  Here we list some examples in Python, JavaScript and Java to show the zero-shot prompts, generation from models and their ground-truth.

```
1  Example Python-0
2  ------ begin of prompt ------
3  def discriminant_value(x,y,z):
4      discriminant = (y**2) - (4*x*z)
5      if discriminant > 0:
6          return ("Two solutions",discriminant)
7      elif discriminant == 0:
8          return ("one solution",discriminant)
9      elif discriminant < 0:
10         return ("no real solution",discriminant)
11 # The above code writes a
12 ------ end of prompt ------
```

```
13
14 # Groundtruth: Write a function to calculate the discriminant value.
15
16 # Generation: Write a "function" that returns the value of the discriminant of a quadratic
        equation.
17
18 Example Python-1
19 ------ begin of prompt ------
20 from collections import Counter
21 def add_dict(d1,d2):
22     add_dict = Counter(d1) + Counter(d2)
23     return add_dict
24 # The above code writes a
25 ------ end of prompt ------
26
27 # Groundtruth:  Write a function to combine two dictionaries by adding values for common keys.
28 # Generation:   Write a "add_dict" function that adds two dictionaries together.
```

```
1 Example Java-0
2 ------ begin of prompt ------
3 import java.io.*;
4 import java.lang.*;
5 import java.util.*;
6
7
8 class ChangeDateFormat {
9
10     public static String changeDateFormat(String dt) {
11         String[] tokens = dt.split("-");
12         return tokens[2] + "-" + tokens[1] + "-" + tokens[0];
13     }
14 }
15 // The above code writes a
16 ------ end of prompt ------
17
18 // Groundtruth: Write a function to convert a date of yyyy-mm-dd format to dd-mm-yyyy format.
19 // Generation: Write a .txt file with the date in the format "dd-mm-yyyy"
```

```
1 Example JavaScript-0
2 ------ begin of prompt ------
3 function swapList(newlist) {
4     let first = newlist[0];
5     let last = newlist[newlist.length - 1];
6     newlist[0] = last;
7     newlist[newlist.length - 1] = first;
8     return newlist;
9 }
10 // The above code writes a
11 ------ end of prompt ------
12
13 // Groundtruth: Write a javascript function to interchange the first and last elements in a
        list.
14 // Generation: Write a "swap" function that swaps the first and last elements of a list.
```

**Few-shot Prompt Examples.** The following examples show the few-shot prompts in different languages.

```
1 Example Python-0
2 ------ begin of prompt ------
3 # summarize the functionality of the code
4
5 # Code
6 import re
7 def find_char_long(text):
8     return (re.findall(r"\b\w{4,}\b", text))
9 # Summary: Write a function to find all words which are at least 4 characters long in a string
        by using regex.
10 ###
11 # Code
12 def find_Rotations(str):
13     tmp = str + str
14     n = len(str)
15     for i in range(1,n + 1):
16         substring = tmp[i: i+n]
17         if (str == substring):
18             return i
19     return n
20 # Summary: Write a python function to find the minimum number of rotations required to get the
        same string.
```

```
21 ###
22 # Code
23 def square_nums(nums):
24     square_nums = list(map(lambda x: x ** 2, nums))
25     return square_nums
26 # Summary: Write a function to find squares of individual elements in a list using lambda
        function.
27 ###
28 # Code
29 def check_tuples(test_tuple, K):
30     res = all(ele in K for ele in test_tuple)
31     return (res)
32 # Summary:
33 ------ end of prompt ------
34
35 # Groundtruth: Write a function to check if the given tuple contains only k elements.
36 # Generation: Write a function to check if a tuple is a subset of a list.
```

```
1  Example Java-0
2  ------ begin of prompt ------
3  // summarize the functionality of the code
4
5  // Code
6  import java.io.*;
7  import java.lang.*;
8  import java.util.*;
9
10
11 class FindCharLong {
12
13     public static List<String> findCharLong(String text) {
14       List<String> result = new ArrayList<>();
15
16       for (String str : text.split(" ")) {
17         if (str.length() >= 4) {
18           result.add(str);
19         }
20       }
21       return result;
22     }
23 }n// Summary: Write a function to find all words which are at least 4 characters long in a
      string by using rege
24 x.
25 ///
26 // Code
27 import java.io.*;
28 import java.lang.*;
29 import java.util.*;
30
31
32 class FindRotations {
33
34     public static int findRotations(String str) {
35
36         String tmp = str + str;
37         int n = str.length();
38         for (int i = 1; i <= n; i++) {
39             String substring = tmp.substring(i, i + n);
40             if (str.equals(substring)) {
41                 return i;
42             }
43         }
44         return n;
45     }
46 }
47 // Summary: Write a javascript function to find the minimum number of rotations required to
      get the same string
48 .
49 ///
50 // Code
51 import java.io.*;
52 import java.lang.*;
53 import java.util.*;
54
55
56 class SquareNums {
57
58     public static List<Integer> squareNums(List<Integer> nums) {
59
60
61         List<Integer> squareList = new ArrayList<Integer>();
```

```
62          for (Integer num : nums)
63              squareList.add(num*num);
64
65          return squareList;
66      }
67 }
68 // Summary: Write a function to find squares of individual elements in a list using lambda
       function.
69 ///
70 // Code
71
72 import java.io.*;
73 import java.lang.*;
74 import java.util.*;
75
76
77 class RecurGcd {
78
79     public static int recurGcd(int a, int b) {
80         int low = Math.min(a, b);
81         int high = Math.max(a, b);
82         if (low == 0) {
83             return high;
84         } else if (low == 1) {
85             return 1;
86         }
87         return recurGcd(low, high % low);
88     }
89 }
90 // Summary:
91 ------ end of prompt ------
92
93 // Groundtruth: Write a function to find the greatest common divisor (gcd) of two integers by
       using recursion.
94 // Generation: Write a function to find the greatest common divisor of two numbers.
```

```
1 Example JavaScript-0
2 ------ begin of prompt ------
3 // summarize the functionality of the code
4
5 // Code
6 function findCharLong(text) {
7   let regex = /\b\w{4,}\b/g;
8   return text.match(regex);
9 }
10 // Summary: Write a function to find all words which are at least 4 characters long in a
       string by using regex.
11 ///
12 // Code
13 function findRotations(str) {
14     let tmp = str + str
15     let n = str.length
16
17     for (let i = 1; i <= n; i++) {
18         let substring = tmp.substring(i, i + n)
19         if (str === substring) {
20             return i
21         }
22     }
23     return n
24 }
25 // Summary: Write a javascript function to find the minimum number of rotations required to
       get the same string.
26 ///
27 // Code
28 function squareNums(nums) {
29   // Write code here
30   let square_nums = nums.map(function (num) {
31     return num ** 2;
32   });
33   return square_nums;
34 }
35 // Summary: Write a function to find squares of individual elements in a list using lambda
       function.
36 ///
37 // Code
38 function findMin(arr, low, high) {
39     while (low < high) {
40         mid = low + (high - low) >> 1;
41         if (arr[mid] > arr[high]) {
42             low = mid + 1;
```

```
43          }
44          if (arr[mid] < arr[high]) {
45              high = mid;
46          }
47      }
48      return arr[high];
49  }
50  // Summary:
51  ------ end of prompt ------
52
53  // Groundtruth: Write a javascript function to find the minimum element in a sorted and
        rotated array.
54  // Generation: Write a function to find the minimum element in a list using lambda function.
```

## M    Evaluating Public Models

We used MBXP to evaluate several public models such as OPT (Zhang et al., 2022b), BLOOM (Mitchell et al., 2022), and CodeGen (Nijkamp et al., 2022). We use pass@1 to evaluate these models, where we generate the samples using greedy decoding. We generate 256 tokens per example and truncate the output to one function for evaluations. The trends we observe with public models are aligned with those observed with our models. In general, we observe a log-linear performance gain with model sizes, across all model families, and better execution accuracy in in-domain languages.

Among the general large-language models we observe that BLOOM models outperform OPT models (See Table 6). This can be attributed to the fact that $13.4\%$ of the pretraining data used for BLOOM models is code, while OPT does not train on code specifically. BLOOM's pretraining data includes code in PHP, Java, Python, Javascript, and Ruby among others, making them all in-domain languages. This explains relatively similar performance across all languages barring Kotlin and Ruby.

CodeGen models are trained in three stages, first is text pretraining, which is followed by code pretraining and python-only training. Here pretraining code data includes code in Python, Java, Javascript, C++, and Go. The CodeGen-multi refers to the models at the end of the code pretraining stage without the python-only training, while the CodeGen-mono is models at the end of all three training stages.

Experiments with CodeGen models show similar performance trends as our models listed in Section 4.2. Specifically, we observe that large models show better than log-linear performance on out-of-domain languages (Kotlin, Java, and PHP). Interestingly, when compared with CodeGen-multi, CodeGen-mono 16B models show $6\%$, and $8\%$ improvements on JavaScript, and PHP, respectively (See Table 6). We speculate the additional training with python data has improved model performance in other languages as well.

With few-shot prompting, we observe significant improvements in out-of-domain languages. Specifically, accuracy with Ruby (which is typically confused with python by the models) increased from $3.5\%$ to $16.46\%$ with few-shot prompting on CodeGen-multi models with few-shot learning (See Table 8). In translation mode, barring Ruby, we find significant improvements in all languages (See Table 10).

Table 6: Evaluating pass@1 execution accuracy of publicly available models on MBXP using greedy decoding

| Model Family | Model Size | Python | Java | JavaScript | Kotlin | Ruby | PHP |
|---|---|---|---|---|---|---|---|
| BLOOM | 350M | 1.54 | 3.21 | 3.21 | 0.83 | 0.00 | 2.80 |
| | 760M | 4.52 | 4.76 | 4.66 | 0.83 | 0.00 | 5.90 |
| | 1.3B | 5.34 | 4.66 | 5.49 | 2.07 | 0.00 | 6.94 |
| | 2.5B | 6.88 | 6.83 | 10.14 | 4.66 | 0.00 | 11.59 |
| | 6.3B | 6.98 | 7.76 | 7.35 | 6.21 | 0.00 | 6.94 |
| OPT | 1.3B | 0.10 | 0.00 | 0.83 | 0.52 | 0.00 | 0.41 |
| | 2.7B | 2.05 | 0.00 | 1.14 | 0.93 | 0.00 | 0.31 |
| | 6.7B | 2.05 | 1.97 | 1.35 | 1.55 | 0.00 | 1.66 |
| | 13B | 1.35 | 1.35 | 1.76 | 2.17 | 0.00 | 0.72 |
| | 30B | 1.64 | 1.45 | 2.69 | 1.45 | 0.00 | 1.55 |
| | 66B | 3.70 | 2.28 | 3.73 | 1.76 | 0.00 | 2.17 |
| CodeGen-multi | 350M | 7.90 | 8.17 | 7.45 | 1.14 | 1.04 | 0.8 |
| | 2B | 18.78 | 19.56 | 17.70 | 3.93 | 4.76 | 2.90 |
| | 6B | 22.48 | 21.74 | 22.87 | 4.55 | 4.24 | 5.90 |
| | 16B | 24.22 | 28.05 | 26.29 | 7.04 | 3.52 | 10.35 |
| CodeGen-mono | 350M | 18.37 | 1.86 | 6.00 | 1.04 | 1.55 | 1.35 |
| | 2B | 31.72 | 16.66 | 22.04 | 3.21 | 2.90 | 9.21 |
| | 6B | 37.16 | 19.77 | 27.74 | 3.83 | 1.66 | 10.14 |
| | 16B | 40.55 | 26.81 | 32.81 | 6.63 | 5.90 | 18.94 |
| Ours-multi | 125M | 6.37 | 5.59 | 6.94 | 1.04 | 0.21 | 0.52 |
| | 672M | 19.71 | 17.29 | 21.43 | 4.55 | 1.86 | 7.76 |
| | 2.7B | 27.00 | 22.46 | 27.23 | 9.73 | 3.93 | 11.28 |
| | 13B | 35.32 | 30.33 | 36.13 | 14.18 | 6.73 | 18.84 |
| Ours-mono | 125M | 8.11 | 4.35 | 7.45 | - | - | - |
| | 672M | 19.82 | 11.39 | 14.39 | - | - | - |
| | 2.7B | 24.13 | 15.32 | 19.67 | - | - | - |
| | 13B | 33.57 | 19.77 | 23.60 | - | - | - |

| Model Family | Model Size | go | c++ | c# | Typescript | Perl | swift | scala |
|---|---|---|---|---|---|---|---|---|
| OPT | 1.3B | 0.11 | 0.35 | 0.00 | 0 | 0.1 | 0.1 | 0.52 |
| | 2.7B | 0.43 | 0.47 | 0.00 | 0 | 0.1 | 0.83 | 0.21 |
| | 6.7B | 1.28 | 1.53 | 3.31 | 3.31 | 0.52 | 0.93 | 0.31 |
| | 13B | 1.7 | 1.06 | 3.31 | 3.31 | 0.31 | 1.97 | 0.31 |
| Bloom | 1.1B | 3.3 | 5.07 | 0.31 | 0.31 | 0.21 | 0.62 | 0.21 |
| | 1.7B | 4.15 | 6.01 | 0.31 | 0.31 | 0.1 | 2.07 | 0.41 |
| | 3B | 6.28 | 8.61 | 10.95 | 10.95 | 1.55 | 4.24 | 0.62 |
| | 7.1B | 7.77 | 15.09 | 13.84 | 13.84 | 3.31 | 5.07 | 0.21 |
| CodeGen-Mono | 350M | 1.38 | 5.19 | 7.13 | 7.13 | 0.1 | 1.14 | 0.1 |
| | 2B | 5.11 | 17.69 | 20.76 | 20.76 | 0.83 | 2.59 | 0.1 |
| | 6B | 3.83 | 17.33 | 19.21 | 19.21 | 1.24 | 4.14 | 0 |
| | 16B | 10.54 | 29.13 | 29.96 | 29.96 | 2.48 | 4.55 | 0.31 |
| CodeGen-Multi | 350M | 6.39 | 9.32 | 7.13 | 7.13 | 0.1 | 1.66 | 0 |
| | 2B | 12.03 | 18.04 | 17.25 | 17.25 | 2.07 | 2.17 | 0.62 |
| | 6B | 11.61 | 17.69 | 17.46 | 17.46 | 2.07 | 2.8 | 0.31 |
| | 16B | 15.23 | 26.06 | 21.49 | 21.49 | 7.14 | 3.62 | 0.41 |
| Ours | 125M | 0.64 | 1.53 | 0.62 | 6.61 | 0.1 | 0.41 | 0 |
| | 672M | 0 | 7.67 | 4.34 | 19.83 | 1.35 | 1.76 | 0.52 |
| | 2B | 0 | 15.68 | 8.16 | 26.14 | 0.93 | 3.11 | 0 |
| | 13B | 5.22 | 18.75 | 10.54 | 32.85 | 4.14 | 6.52 | 0.1 |

Table 7: Evaluating pass@1 execution accuracy of publicly available models on Humaneval using greedy decoding

| Model Family | Model Size | PY | Java | JS | Kotlin | Ruby | PHP | Perl | Swift | Scala |
|---|---|---|---|---|---|---|---|---|---|---|
| Bloom | 1.1B | 3.66 | 3.73 | 2.48 | 0.62 | 0.00 | 2.48 | 0.62 | 0.62 | 8.07 |
| | 1.7B | 3.66 | 1.86 | 4.97 | 0.62 | 0.00 | 4.35 | 0.00 | 0.62 | 24.22 |
| | 3B | 7.93 | 4.97 | 5.59 | 2.48 | 0.00 | 4.97 | 0.62 | 1.24 | 29.19 |
| | 7.1B | 7.93 | 8.07 | 6.21 | 0.62 | 0.00 | 3.11 | 0.62 | 2.48 | 34.16 |
| OPT | 1.3B | 0.00 | 0.00 | 0.62 | 0.62 | 0.00 | 0.00 | 0.00 | 0.62 | 0.00 |
| | 2.7B | 0.00 | 0.00 | 0.00 | 0.00 | 0.00 | 1.86 | 0.00 | 0.00 | 0.00 |
| | 6.7B | 0.61 | 0.62 | 0.62 | 0.62 | 0.00 | 1.24 | 0.00 | 0.62 | 9.32 |
| | 13B | 0.61 | 0.62 | 2.48 | 0.62 | 0.00 | 1.24 | 0.00 | 0.62 | 12.42 |
| CodeGen-Mono | 350M | 10.37 | 1.24 | 3.11 | 0.00 | 0.00 | 0.62 | 0.00 | 0.62 | 5.59 |
| | 2B | 20.73 | 4.97 | 10.56 | 1.24 | 0.00 | 3.73 | 1.24 | 0.62 | 8.07 |
| | 6B | 19.51 | 8.70 | 11.18 | 1.24 | 0.00 | 4.35 | 1.24 | 0.62 | 6.21 |
| | 16B | 22.56 | 17.39 | 12.42 | 0.62 | 0.00 | 11.80 | 2.48 | 0.62 | 16.15 |
| CodeGen-Multi | 350M | 7.32 | 4.97 | 4.35 | 0.62 | 0.00 | 0.62 | 0.00 | 0.62 | 1.86 |
| | 2B | 10.98 | 11.18 | 6.83 | 3.11 | 0.00 | 1.86 | 1.24 | 0.62 | 21.12 |
| | 6B | 15.24 | 10.56 | 11.80 | 3.11 | 0.62 | 3.73 | 1.24 | 0.62 | 10.56 |
| | 16B | 17.07 | 16.15 | 16.15 | 1.86 | 0.00 | 5.59 | 3.11 | 0.62 | 16.15 |
| Ours | 125M | 7.32 | 3.73 | 4.35 | 1.24 | 0.00 | 0.62 | 0.00 | 0.62 | 1.86 |
| | 672M | 17.07 | 9.32 | 13.04 | 1.86 | 0.00 | 1.86 | 1.24 | 0.62 | 1.24 |
| | 2B | 19.51 | 14.29 | 14.91 | 1.86 | 0.00 | 4.97 | 0.00 | 0.62 | 1.86 |
| | 13B | 22.56 | 22.36 | 20.50 | 8.07 | 0.00 | 11.80 | 3.11 | 0.62 | 4.35 |

Table 8: Evaluating pass@1 execution accuracy of publicly available models on MBXP with few-shot prompting using greedy decoding

| Model Family | Model Size | Python | Java | JavaScript | Kotlin | Ruby | PHP |
|---|---|---|---|---|---|---|---|
| Codegen-multi | 350M | 7.80 | 10.56 | 8.28 | 2.28 | 4.24 | 3.2 |
| | 2B | 20.02 | 22.15 | 21.43 | 7.76 | 12.73 | 8.8 |
| | 6B | 23.10 | 24.53 | 24.84 | 10.66 | 9.01 | 13.87 |
| | 16B | 26.69 | 29.92 | 28.26 | 11.59 | 16.46 | 17.60 |
| CodeGen-mono | 350M | 17.04 | 3.73 | 5.18 | 2.69 | 4.35 | 2.69 |
| | 2B | 28.95 | 15.42 | 15.53 | 5.69 | 8.07 | 11.70 |
| | 6B | 39.53 | 21.43 | 19.15 | 7.14 | 10.35 | 16.4 |
| | 16B | 46.41 | 27.64 | 26.50 | 11.08 | 15.11 | 20.2 |
| Ours-multi | 125M | 6.06 | 7.14 | 6.94 | 2.90 | 2.59 | 0.93 |
| | 672M | 19.40 | 16.56 | 19.46 | 5.90 | 5.90 | 8.90 |
| | 2.7B | 24.23 | 24.12 | 27.95 | 11.08 | 9.42 | 14.29 |
| | 13B | 31.93 | 30.75 | 37.37 | 15.11 | 12.53 | 20.19 |
| Ours-mono | 125M | 8.93 | 5.38 | 7.35 | - | - | - |
| | 672M | 18.89 | 10.56 | 16.87 | - | - | - |
| | 2.7B | 21.77 | 15.11 | 21.43 | - | - | - |
| | 13B | 31.31 | 21.22 | 25.16 | - | - | - |

| Model Family | Model Size | Go | C++ | C# | Typescript | Perl | Swift | Scala |
|---|---|---|---|---|---|---|---|---|
| CodeGen-Mono | 350M | 1.17 | 4.25 | 4.03 | 3.93 | 1.45 | 2.90 | 28.78 |
| | 2B | 5.86 | 19.34 | 8.99 | 16.94 | 4.97 | 4.45 | 26.29 |
| | 6B | 6.50 | 19.69 | 12.71 | 18.08 | 3.42 | 4.24 | 26.50 |
| | 16B | 13.84 | 32.31 | 16.63 | 28.20 | 8.18 | 5.49 | 28.67 |
| CodeGen-Multi | 350M | 6.39 | 9.91 | 5.68 | 9.19 | 1.66 | 3.62 | 30.43 |
| | 2B | 14.59 | 19.46 | 11.05 | 18.80 | 4.24 | 3.83 | 37.68 |
| | 6B | 12.78 | 21.58 | 13.43 | 19.83 | 6.00 | 4.55 | 28.26 |
| | 16B | 20.77 | 29.36 | 17.46 | 24.38 | 8.49 | 4.55 | 28.57 |

Table 9: Evaluating pass@1 execution accuracy of publicly available models on HumanEval with few-shot prompting using greedy decoding

| Model Family | Model Size | PY | Java | JS | Kotlin | Ruby | PHP | Perl | Swift | Scala |
|---|---|---|---|---|---|---|---|---|---|---|
| CodeGen-Mono | 350M | 12.80 | 3.11 | 2.48 | 1.86 | 2.48 | 1.24 | 0.62 | 0.62 | 19.25 |
| | 2B | 21.95 | 8.07 | 10.56 | 3.11 | 3.11 | 6.83 | 1.86 | 0.62 | 15.53 |
| | 6B | 23.17 | 8.07 | 8.70 | 3.11 | 2.48 | 7.45 | 1.24 | 0.62 | 13.66 |
| | 16B | 28.66 | 16.77 | 11.18 | 4.97 | 5.59 | 9.94 | 3.73 | 1.24 | 18.63 |
| CodeGen-Multi | 350M | 7.93 | 4.97 | 4.35 | 1.86 | 1.24 | 1.24 | 1.24 | 1.24 | 22.98 |
| | 2B | 13.41 | 10.56 | 12.42 | 4.35 | 7.45 | 4.35 | 4.35 | 0.62 | 24.22 |
| | 6B | 14.02 | 12.42 | 11.80 | 3.11 | 4.97 | 3.11 | 3.73 | 1.24 | 13.04 |
| | 16B | 20.12 | 17.39 | 13.66 | 4.35 | 8.07 | 9.94 | 5.59 | 0.62 | 16.15 |
| Ours | 125M | 7.32 | 4.35 | 5.59 | 1.24 | 0.62 | 1.86 | 0.00 | 0.62 | 0.62 |
| | 672M | 17.07 | 8.70 | 11.18 | 3.73 | 2.48 | 4.97 | 1.24 | 1.24 | 1.86 |
| | 2B | 19.51 | 14.29 | 14.91 | 4.97 | 3.11 | 8.07 | 2.48 | 1.24 | 3.11 |
| | 13B | 22.56 | 18.01 | 26.09 | 4.97 | 6.21 | 10.56 | 3.72 | 0.62 | 5.59 |

Table 10: Evaluating pass@1 execution accuracy of publicly available models on MBXP in translation mode (Python as a source language).

| Model Family | Model Size | Java | JavaScript | Kotlin | Ruby | PHP | Go |
|---|---|---|---|---|---|---|---|
| CodeGen-Mono | 350M | 3.83 | 10.24 | 3.11 | 4.66 | 2.07 | 3.94 |
| | 2B | 22.57 | 22.87 | 5.18 | 4.35 | 21.74 | 8.41 |
| | 6B | 30.95 | 35.92 | 8.07 | 6.63 | 30.23 | 11.93 |
| | 16B | 43.48 | 54.14 | 10.56 | 4.55 | 36.85 | 22.36 |
| CodeGen-Multi | 350M | 7.35 | 9.93 | 1.55 | 3.83 | 1.76 | 7.14 |
| | 2B | 23.19 | 36.12 | 3.42 | 4.76 | 22.46 | 16.83 |
| | 6B | 38.41 | 37.99 | 6.83 | 4.66 | 27.74 | 20.77 |
| | 16B | 44.82 | 50.10 | 8.28 | 5.18 | 40.17 | 26.41 |
| Ours | 125M | 7.04 | 9.52 | 3.21 | 3.73 | 0.41 | 2.77 |
| | 672M | 22.98 | 24.53 | 7.97 | 5.07 | 16.15 | 6.28 |
| | 2B | 30.75 | 28.47 | 10.56 | 6.52 | 33.02 | 6.92 |
| | 13B | 41.93 | 37.06 | 14.80 | 6.73 | 37.06 | 8.52 |

| Model Family | Model Size | C++ | C# | Typescript | Perl | Swift | Scala |
|---|---|---|---|---|---|---|---|
| CodeGen-Mono | 350M | 8.37 | 3.93 | 11.78 | 0.21 | 1.24 | 0.00 |
| | 2B | 30.19 | 16.01 | 31.51 | 4.14 | 6.83 | 0.00 |
| | 6B | 35.14 | 23.45 | 36.67 | 7.56 | 7.04 | 0.10 |
| | 16B | 49.65 | 36.67 | 0.41 | 7.04 | 10.97 | 0.10 |
| CodeGen-Multi | 350M | 10.85 | 1.86 | 5.89 | 0.00 | 0.83 | 0.00 |
| | 2B | 23.00 | 6.71 | 10.54 | 5.18 | 3.62 | 0.00 |
| | 6B | 40.45 | 22.52 | 26.96 | 8.39 | 6.21 | 0.10 |
| | 16B | 46.58 | 28.51 | 5.89 | 14.18 | 8.18 | 0.10 |
| Ours | 125M | 1.06 | 1.76 | 10.54 | 0.52 | 1.76 | 0.00 |
| | 672M | 16.51 | 10.43 | 19.11 | 4.66 | 3.93 | 0.10 |
| | 2B | 28.18 | 13.33 | 24.90 | 5.07 | 6.52 | 0.10 |
| | 13B | 33.14 | 25.52 | 44.52 | 10.56 | 8.49 | 0.10 |

# N TRAINING

## N.1 MODEL ARCHITECTURE AND TRAINING DETAILS

We train using 210B tokens for mono-lingual models, and 630B tokens for multi-lingual models with $210B$ tokens from each language. Across all models, we use max sequence length of 2048, and use larger batch size for larger models, while reducing max steps accordingly to train all models with same amount of per-language tokens. For example, for 13B, we use batch size of 1024 and max steps of 100,000 with 2048 sequence length, resulting in total 210B training tokens for each language. For multi-lingual models, there are three languages and we increase max steps by three times to 630B tokens. We use AdamW optimizer (Loshchilov & Hutter, 2018) with $\beta_1 = 0.9$, $\beta_2 = 0.95$, and $\epsilon = 10^{-8}$. We use warm up steps of 2000 steps with cosine annealing after peak learning rate, and the mininum learning rate being $10\%$ of corresponding peak learning rate, weight decay of 0.01, and gradient clipping of 1.0. We rescale the initialization weight standard deviation for larger models following (Shoeybi et al., 2019) for better training stability. Our training pipeline is based on PyTorch Lightning [¶] and we use bfloat16 (Kalamkar et al., 2019) and DeepSpeed (Rasley et al., 2020) for training optimization. We randomly split $0.1\%$ data as validation set. The validation loss curve for different sizes of multi-lingual and monolingual models are shown in Fig. 24.

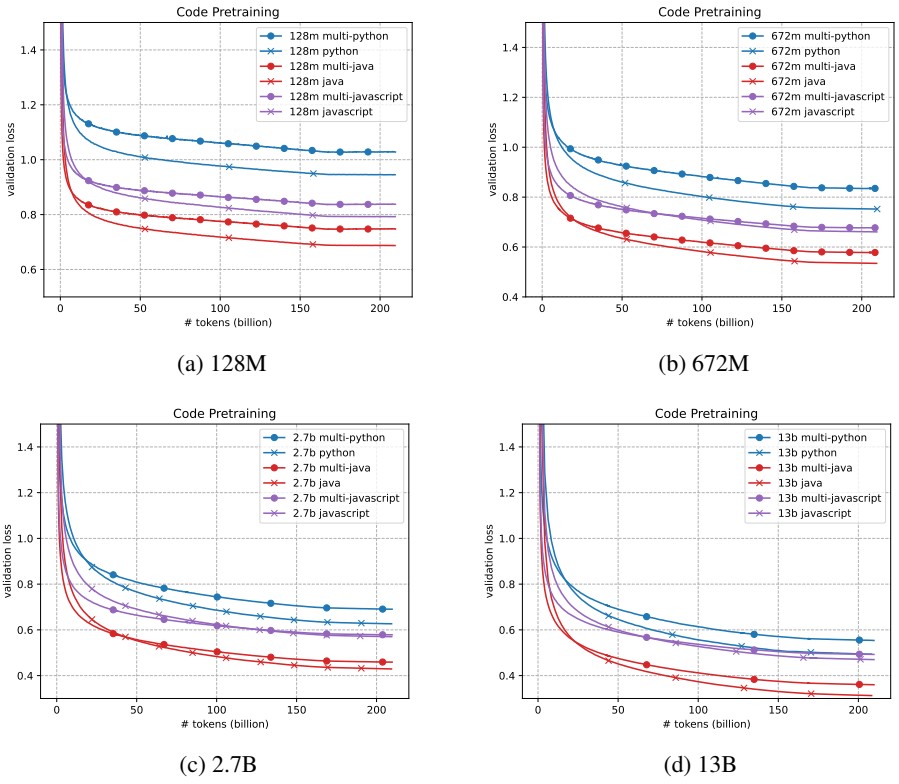

Figure 24: Validation loss curves for 128M, 672M, 2.7B and 13B multi-lingual and mono-lingual models.

## N.2 OBSERVATIONS ON VALIDATION LOSSES VERSUS PERFORMANCE

We plot the validation loss of multi/mono-lingual models on each programming languages in Figure 25. We can see that the trend of validation loss roughly follows log-linear relationship with respect to model sizes.

By comparing the validation loss curves between multi-lingual models and mono-lingual models, we can see that mono-lingual models consistently achieves lower loss than multi-lingual ones. This

---

[¶] https://www.pytorchlightning.ai/

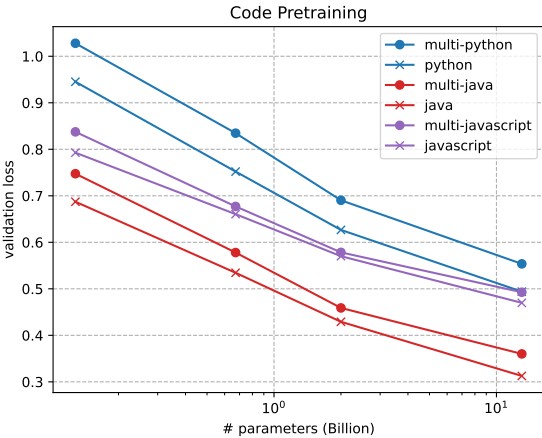

Figure 25: Validation loss vs number of parameters for 128M, 672M, 2.7B and 13B multi-lingual and mono-lingual models.

demonstrate that it is more difficult for the same size of model to fit multi-lingual datasets since with limited model capacity it needs to learn more diverse information while mono-lingual models can be more concentrated.

However, although the validation loss of mono-lingual models is generally lower, from Section 4.2, we observe that in terms of execution performance pass@k, multi-lingual models actually outperform mono-lingual ones especially when model sizes go beyond 672M. In fact, as model size increases, the improvement of multi-lingual models over mono-lingual models get more and more significant. The reason could be that although models get distracted to fit multiple languages, the knowledge sharing across different languages helps model to learn better in solving problems. For example, similar tasks might exist in different programming languages, hence models are easier to learn to transfer from one language to another. And larger models have better capability in knowledge sharing/transfer learning, with the evidence that the zero-shot learning performance of multi-lingual models on unseen programming languages get significantly better than mono-lingual ones as model sizes increases.

## O    DATASET CONVERSION FRAMEWORK

In this section, we describe a dataset conversion framework that transforms an execution-based evaluation in one programming language to another. In particular, we focus on a *function completion* format of execution-based evaluation as shown in Figure 2. Each problem in a function completion dataset consists of a *prompt*, a *test statement*, and a *canonical solution*. The prompt contains a function signature along with a docstring describing the desired functionality of the code. The canonical solution is an example of a function body that fulfills such functionality, usually written by human annotators. Given a candidate function body generated by the model, we can test whether the corresponding function is correct by executing the test statement against the candidate function.

To construct a evaluation dataset for function completion in a new language, we recognize that it is **sufficient** to convert only the *prompts* and the *test statements* (Section O.1). That is, we do not need to transform the canonical solutions, since they are simply examples and are not used to measure correctness in the test-based framework. This key feature of a test-based evaluation makes it possible to perform mapping of an evaluation set from one language to many others by static analyses, as outline below. For other code-related evaluations that require access to canonical solutions, we synthesize solutions by generating many code versions based on the converted prompt and use our converted test statement to filter for correctness (Section O.1.11).

### O.1    LANGUAGE CONVERSION OF PROMPTS AND TEST CASES

#### O.1.1    FORMAT CHOICE

The purpose of this work is to build datasets that allow us to measure multi-lingual code generation abilities. The function completion format helps steer the model to generate code in a specific language since the prompt consists of a partial function that has already been started, i.e. a function signature. This is in contrast to other formats such as that of the original MBPP where the prompt does not consist of a function signature, but contains more implicit information such as assert statements, example function calls, and a description such as "Write code in Python" (see Appendix R.1.1 for examples). Compared to other formats, the execution-based function completion aligns well with how an ML-driven model would perform code suggestion in a typical coding environment. Therefore, we process our converted datasets and the original datasets (MBPP, MathQA) to be of this format, except for the original HumanEval dataset whose format is already consistent.

#### O.1.2    INFERRING ARGUMENT AND RETURN TYPES

This step is applicable for statically typed target languages such as Java, C#, etc. The process starts from inferring the types of function arguments, which can be done by inspecting the argument values. We perform mapping of types from Python to types in a target language; for instance, to convert to Java, we map `list` → `ArrayList` or `dict` → `HashMap`. Values for different test cases can have different types, therefore we infer the common superclass of all observed types for each argument. Since there can also be many levels of types, due to containers such as list or sets, we recursively infer the types among each level to be consistent. For example, "list of list" and "list of object" has a common type of "list of object". The return type is inferred via expected return values in the test cases.

#### O.1.3    SUPPORTED TYPES OF OBJECT CONVERSION

Our conversion framework depends on the structure of basic programming problems which involve object types of the following:

- Integer or long version of integer
- Float or double
- Boolean
- String. We assume any string of single character is also of type string for the purpose of conversion.
- None. This depends whether the target language also supports None/null/nil types.

- List. Tuples in Python are also converted to list in all languages.

- Dictionary

- Set

For any container type, we recursively perform object conversion for all nested structure within the container.

### O.1.4 CONSTRUCTING REPRESENTATIONS OF CODE OBJECTS

We convert the argument and return values from Python to a target language by generating strings that represent the target language's objects. For example, the object `[1,2,3]` in Python is converted to `Arrays.asList(1, 2, 3)` in Java, as shown in Figure 2. We recursively construct container elements for any nested structures.

### O.1.5 CONVERTING TEST STATEMENT

We construct objects for function argument inputs, using above information regarding constructed objects and types, as well as expected output. We build the test statement in a target language using appropriate assertion to match the returned value with the expected output. We perform deep object comparison with an appropriate comparator for each language.

### O.1.6 PROMPT CONSTRUCTION

We ensure that the converted function, argument, and class names are stylistically appropriate for each language, e.g. camel case versus snake case, etc. We construct function call examples in the docstring to look representative without being too verbose, e.g., we use `[1,2,3]` in Java's docstring to represent a list, instead of an actual `ArrayList`. We avoid using language-reserve words for variable names such as `end` for Ruby or `char` for Java or C++ and escape certain substrings that are keywords such as `/*` or `//`.

We also deal with all formats in the prompt with great care. For instance, docstrings for Java and JavaScript are to be before the function signature, following the convention. For Java, this is crucial, otherwise it would be too out of distribution and the model would not generate anything, if docstring is below function signature.

### O.1.7 DOCSTRING AND NATURAL LANGUAGE CONVERSION

The natural language statements for datasets such as MBPP can contain python-specific statements that might not be applicable to Java or Javascript such as "Write a function in Python to ..." or "... if the object does not exist, return None". We substitute "Python" with the target language name, "None" as appropriate null values.

### O.1.8 VALIDATION

We validate that converted objects, test statements and function signatures parse and/or compile with respect to each language.

### O.1.9 QUALITY CHECK VIA REVIEWERS

To gauge the quality of our conversion, we also request annotators to manually review the converted programming problems in sample languages, namely, Java and JavaScript. We ask language experts to identify issues with converted examples consisting of natural language statement, test cases, and function signature and use this process to help iteratively improve our conversion algorithm. For the final review, annotators have not found issues specifically related to language conversion, but observed ambiguity in some cases attributed to the original dataset. In the future, any updates to the original datasets can propagated to all converted languages programmatically. We provide the detailed analysis of evaluation by annotators in Appendix Q.

### O.1.10 COMPARISON TO OFF-THE-SHELF ML TRANSLATOR

We find ML translators to be insufficient to perform the dataset conversion, due to limited support for language pairs, restriction on format, as well as transformation errors related to types and object construction. In contrast, our framework can convert data to many target languages and does not have non-deterministic errors related to type inference or object mapping. We provide further discussion and examples in Appendix O.2.

### O.1.11 SYNTHETIC CANONICAL SOLUTIONS

The availability of canonical solutions in each converted language can open up the possibilities to perform other types of evaluation. To generate such synthetic solutions for each language, we sample up to $10,000$ versions of code per problem and filter them for correctness with our converted tests. In order to generate at least one correct solution for as many problems as possible, we use both the function completion and zero-shot translation settings (see Section 4.3) where we prepend the Python solution, provided in the original datasets by human annotators, to the beginning of the function signature prompt. With high-temperature sampling, we are able to generate correct solutions for a large portion of all problems, with $96\%$ coverage for JavaScript, $93\%$ for Java, and even on an out-of-domain language such as PHP with $93\%$ coverage (more details in Appendix P).

## O.2 POTENTIAL USE OF TRANSCODER FOR DATASET CONSTRUCTION

We conduct preliminary experiments using a publicly available code translation model Transcoder (Lachaux et al., 2020b) to perform dataset conversion. Overall, there are two main limitations of this approach. First, these models typically support a limited number of language pairs, which means that we would not be able to perform conversion to 10+ languages like with our proposed framework. Second, we find that there are some common errors associated with type inference, for instance, when the return type should be `boolean`, the translation model can predict `int` as a return type. These types of error cause false negatives and can impact overall quality of the converted datasets. In contrast, we do not have these types of errors in our conversion framework due to the static analysis implementation.

In particular, Transcoder (Lachaux et al., 2020b) supports Python, Java, and C++. In this setup, we use a complete function in Python as an input prompt. The transcoder model then generates a complete function in Java and C++. Here, we are interested in whether the model is able to translate function signatures that capture necessary information.

**Example 1** While the model seems able to translate the function signature, the function name for Java and C++ appear to be in snake case, which is not the standard for these languages.

```
1  ==================== TASK_ID: 6  ===========================
2  ------------------- Input in Python ----------------------
3  def differ_At_One_Bit_Pos(a,b):
4      return is_Power_Of_Two(a ^ b)
5  ------------------- Translation in Java ------------------
6  public static boolean differAt_One_Bit_Pos ( int a , int b ) {
7      .
8      .
9  ------------------- Translation in C++ -------------------
10 bool Differ_At_One_Bit_Pos ( int a , int b ) {
11     .
12     .
```

**Example 2** The model seems to adapt the function name to be entirely different, i.e., `is_undulating` to `isAbundulating` or `isSkundulating`.

```
1  ==================== TASK_ID: 92  ==========================
2  ------------------- Input in Python ----------------------
3  def is_undulating(n):
4    if (len(n) <= 2):
5      return False
6    for i in range(2, len(n)):
7      if (n[i - 2] != n[i]):
8        return False
9    return True
10 ------------------- Translation in Java ------------------
```

```
11 public static boolean isAbundulating ( int [ ] n ) {
12    .
13    .
14 ------------------- Translation in C++ --------------------
15 bool isSkundulating ( string n ) {
16    .
17    .
```

**Example 3**  Incorrect type inference where a nested list `nestedList` is Python is mapped to a string in C++, or simply a flat list of strings in Java.

```
1 ==================== TASK_ID: 111  =========================
2 ------------------- Input in Python ----------------------
3 def common_in_nested_lists(nestedlist):
4     result = list(set.intersection(*map(set, nestedlist)))
5     return result
6 ------------------- Translation in Java ------------------
7 public static String commonInNestedLists ( String [ ] nestedlist ) {
8    .
9    .
10 ------------------- Translation in C++ --------------------
11 string commonInNestedLists ( string nestedlist ) {
12    .
13    .
```

**Example 4**  A list `test_list` is incorrectly inferred to have a type `string` in C++.

```
1 ==================== TASK_ID: 117 (fn: 1_of_1) ====================
2 ------------------- Input in Python ----------------------
3 def list_to_float(test_list):
4   res = []
5   for tup in test_list:
6     temp = []
7     for ele in tup:
8       if ele.isalpha():
9         temp.append(ele)
10       else:
11         temp.append(float(ele))
12     res.append((temp[0],temp[1]))
13   return (str(res))
14 ------------------- Translation in Java ------------------
15 public static String listToDouble ( ArrayList < String > testList ) {
16    .
17    .
18
19 ------------------- Translation in C++ --------------------
20 string listToDouble ( string testList ) {
21    .
22    .
```

## P  SYNTHETIC CANONICAL SOLUTIONS

### P.1  MULTI-STAGE DATA BOOTSTRAPPING

We combine solutions for (1) normal function completion or with few-shot prompting if the language is out-of-domain and (2) translation settings. This is because these different generation modes can synthetic correct solutions for different problems, according to our evaluation analyses in Section 4.2. We perform data generation in multiple stages. We first sample $n = 100$ samples for all cases, after which we sample for $n = 1000$ cases for the problems where we have not found at least one correct solution. The last step contains uses $n = 10000$ samples. We use temperature 0.8, 1.0 and 1.2 respectively.

### P.2  DISCUSSION: GROUND TRUTH ASSUMPTIONS OF TEST CASES

Our synthetic generation of canonical solutions make heavy use of test cases to filter whether each code is correct or not. The process implicitly assumes that the test cases act as a ground truth verifier that provides necessary and sufficient conditions for the correctness of each task's functionality. Such assumptions might not hold if the test cases are not thoroughly written in the original dataset.

In fact, false positives of execution-based datasets can typically occur as indicated in several previous work (Li et al., 2022a). Our framework do not aim to inject additional knowledge per each test case but merely acts to translate a task in one language to another, while carrying the information captured in test cases of the original dataset to the corresponding datasets in other languages. Any corrections to the benchmark shall be done upstream in the original dataset, and can easily propagate to the rest of the converted datasets, since the conversion process for prompts and test cases is purely programmatic. The usefulness of the conversion framework lies in the automated conversion into many languages which can be repeatedly done if the test cases in original datasets are updated, or new tasks are added. This helps reduce human effort to perform such manual translation of a dataset in one language to many others.

One step that can be done to improve thoroughness of the test cases is to use the provided canonical solutions in the original dataset to synthetically generate additional test cases, with the hope that additional test cases can provide test coverage for the functionality. However, this proposal also relies heavily on the canonical solution as the ground truth that captures the true functionality of the task, which might not necessarily be true since during the annotation process, annotators might write canonical solutions that are only partially correct but pass the specified test cases. Therefore, we leave this investigation as future work.

# Q QUALITY CHECK OF CONVERTED DATASETS

During the manual review process, we provide the converted programming problem in Java and Javascript to annotators and ask them to check if the problem is correct and clear. Below we categorize the issues identified by annotators, with examples. Almost all of these cases can be attributed to the source dataset we use for conversion. On the other hand, the conversion process itself does not introduce additional errors. For future work, we plan to thoroughly check for errors in the original MBPP. Any changes from there can be easily propagated to the converted datasets due to the automatic conversion.

- Natural language statement is ambiguous.

```java
1
2  import java.io.*;
3  import java.lang.*;
4  import java.util.*;
5
6
7  class CountCommon {
8
9      /**
10      * Write a function to count the most common words in a dictionary.
11      * > CountCommon.countCommon(["red", "green", "black", "pink", "black", "white", "
        black", "eyes", "white", "black", "orange", "pink", "pink", "red", "red", "white
        ", "orange", "white", "black", "pink", "green", "green", "pink", "green", "pink",
         "white", "orange", "orange", "red"])
12      * [["pink", 6], ["black", 5], ["white", 5], ["red", 4]]
13      * > CountCommon.countCommon(["one", "two", "three", "four", "five", "one", "two",
         "one", "three", "one"])
14      * [["one", 4], ["two", 2], ["three", 2], ["four", 1]]
15      * > CountCommon.countCommon(["Facebook", "Apple", "Amazon", "Netflix", "Google",
        "Apple", "Netflix", "Amazon"])
16      * [["Apple", 2], ["Amazon", 2], ["Netflix", 2], ["Facebook", 1]]
17      */
18      public static List<List<Object>> countCommon(List<String> words) {
19
20  // Comment: The problem should explicit state to count the 4 most common words.
```

- Spelling mistake.

```java
1
2  /**
3   * Write a javascript function to count numbers whose oth and nth bits are set.
4   * > countNum(2)
5   * 1
6   * > countNum(3)
7   * 2
8   * > countNum(1)
9   * 1
10   */
11  function countNum(n) {
12
13  // Comment: "oth" should be "0th"
```

- One or more test cases are wrong.

```java
1
2  import java.io.*;
3  import java.lang.*;
4  import java.util.*;
5
6
7  class FirstRepeatedChar {
8
9      /**
10      * Write a java function to find the first repeated character in a given string.
11      * > FirstRepeatedChar.firstRepeatedChar("Google")
12      * "o"
13      * > FirstRepeatedChar.firstRepeatedChar("data")
14      * "a"
15      * > FirstRepeatedChar.firstRepeatedChar("python")
16      * "\x00"
17      */
18      public static String firstRepeatedChar(String str) {
19
20  // Comment: The last test case is incorrect.
21
```

# R   DATASETS

## R.1   MBXP

Below, we show examples of the samples from the original dataset as well as the converted dataset.

Table 11: MBXP Datasets in 10+ Languages. The dataset names are loosely inspired by each language's file extension. For instance, the Scala or Perl dataset is MBSCP or MBPKP due to the file extension being `.sc` or `.pl`.

| Language | Dataset Name |
|---|---|
| Python | MBPP |
| Java | MBJP |
| JavaScript | MBJSP |
| TypeScript | MBTSP |
| Go | MBGP |
| C# | MBCSP |
| PHP | MBPHP |
| Ruby | MBRBP |
| Kotlin | MBKP |
| C++ | MBCPP |
| Perl | MBPLP |
| Scala | MBSCP |
| Swift | MBSWP |

## R.1.1   MBPP: PYTHON

Note that we convert the original MBPP dataset (Austin et al., 2021) which has a slightly different format into HumanEval format (Chen et al., 2021) with function signature and docstring, as shown below. The use of function signature and docstring in the formatted MBPP makes it consistent with the converted datasets in all other languages.

```
1  # ---------------------     MBPP/1:   PROMPT        ---------------------
2  def min_cost(cost, m, n):
3      """
4      Write a function to find the minimum cost path to reach (m, n) from (0, 0) for the given
        cost matrix cost[][] and a position (m, n) in cost[][].
5      >>> min_cost([[1, 2, 3], [4, 8, 2], [1, 5, 3]], 2, 2)
6      8
7      >>> min_cost([[2, 3, 4], [5, 9, 3], [2, 6, 4]], 2, 2)
8      12
9      >>> min_cost([[3, 4, 5], [6, 10, 4], [3, 7, 5]], 2, 2)
10     16
11     """
12 # ---------------------         CANONICAL SOLUTION      ---------------------
13     R = 3
14     C = 3
15
16     tc = [[0 for x in range(C)] for x in range(R)]
17     tc[0][0] = cost[0][0]
18     for i in range(1, m+1):
19         tc[i][0] = tc[i-1][0] + cost[i][0]
20     for j in range(1, n+1):
21         tc[0][j] = tc[0][j-1] + cost[0][j]
22     for i in range(1, m+1):
23         for j in range(1, n+1):
24             tc[i][j] = min(tc[i-1][j-1], tc[i-1][j], tc[i][j-1]) + cost[i][j]
25     return tc[m][n]
26
27 # ---------------------              TEST         ---------------------
28
29 def check(candidate):
30     assert candidate([[1, 2, 3], [4, 8, 2], [1, 5, 3]], 2, 2) == 8
31     assert candidate([[2, 3, 4], [5, 9, 3], [2, 6, 4]], 2, 2) == 12
32     assert candidate([[3, 4, 5], [6, 10, 4], [3, 7, 5]], 2, 2) == 16
```

## R.1.2   MBJP: JAVA

```
1  // --------------------     MBJP/1:   PROMPT        --------------------
2  import java.io.*;
3  import java.lang.*;
4  import java.util.*;
5  import java.math.*;
6
7
8  class MinCost {
9
10     /**
11      * Write a function to find the minimum cost path to reach (m, n) from (0, 0) for the
        given cost matrix cost[][] and a position (m, n) in cost[][].
12      * > MinCost.minCost([[1, 2, 3], [4, 8, 2], [1, 5, 3]], 2, 2)
13      * 8
14      * > MinCost.minCost([[2, 3, 4], [5, 9, 3], [2, 6, 4]], 2, 2)
15      * 12
16      * > MinCost.minCost([[3, 4, 5], [6, 10, 4], [3, 7, 5]], 2, 2)
17      * 16
18      */
19     public static int minCost(List<List<Integer>> cost, int m, int n) {
20  // --------------------- COMPLETION (model generated) ---------------------
21         int[][] dp = new int[m][n];
22         for (int i = 0; i < m; i++) {
23             for (int j = 0; j < n; j++) {
24                 if (i == 0 && j == 0) {
25                     dp[i][j] = cost.get(i).get(j);
26                 } else if (i == 0) {
27                     dp[i][j] = dp[i][j - 1] + cost.get(i).get(j);
28                 } else if (j == 0) {
29                     dp[i][j] = dp[i - 1][j] + cost.get(i).get(j);
30                 } else {
31                     dp[i][j] = Math.min(dp[i - 1][j], dp[i][j - 1]) + cost.get(i).get(j);
32                 }
33             }
34         }
35         return dp[m - 1][n - 1];
36     }
37  }
38  // --------------------            TEST            --------------------
39  class Main {
40      public static boolean compare(Object obj1, Object obj2) {
41          return obj1.equals(obj2);
42      }
43
44      public static void main(String[] args) throws Exception {
45          int x0 = MinCost.minCost(Arrays.asList(Arrays.asList(1, 2, 3), Arrays.asList(4, 8, 2),
         Arrays.asList(1, 5, 3)), 2, 2);
46          if (!(compare(x0, 8))) {
47              throw new java.lang.Exception("Exception -- test case 0 did not pass. x0 = " + x0)
         ;
48          }
49
50          int x1 = MinCost.minCost(Arrays.asList(Arrays.asList(2, 3, 4), Arrays.asList(5, 9, 3),
         Arrays.asList(2, 6, 4)), 2, 2);
51          if (!(compare(x1, 12))) {
52              throw new java.lang.Exception("Exception -- test case 1 did not pass. x1 = " + x1)
         ;
53          }
54
55          int x2 = MinCost.minCost(Arrays.asList(Arrays.asList(3, 4, 5), Arrays.asList(6, 10, 4)
         , Arrays.asList(3, 7, 5)), 2, 2);
56          if (!(compare(x2, 16))) {
57              throw new java.lang.Exception("Exception -- test case 2 did not pass. x2 = " + x2)
         ;
58          }
59
60      }
61  }
```

### R.1.3   MBJSP: JAVASCRIPT

```
1  // --------------------     MBJSP/1:   PROMPT        --------------------
2  /**
3   * Write a function to find the minimum cost path to reach (m, n) from (0, 0) for the given
       cost matrix cost[][] and a position (m, n) in cost[][].
4   * > minCost([[1, 2, 3], [4, 8, 2], [1, 5, 3]], 2, 2)
5   * 8
6   * > minCost([[2, 3, 4], [5, 9, 3], [2, 6, 4]], 2, 2)
7   * 12
```

```
8   * > minCost([[3, 4, 5], [6, 10, 4], [3, 7, 5]], 2, 2)
9   * 16
10  */
11 function minCost(cost, m, n) {
12 // --------------------- COMPLETION (model generated) ---------------------
13   const dp = new Array(m + 1).fill(0).map(() => new Array(n + 1).fill(0));
14   for (let i = 1; i <= m; i++) {
15     for (let j = 1; j <= n; j++) {
16       dp[i][j] = cost[i - 1][j - 1] + Math.min(dp[i - 1][j], dp[i][j - 1]);
17     }
18   }
19   return dp[m][n];
20 }
21
22 // ---------------------            TEST            ---------------------
23 const _ = require("lodash")
24
25 function compare(object1, object2){
26     return _.isEqual(object1, object2)
27 }
28
29 var x = minCost([[1, 2, 3], [4, 8, 2], [1, 5, 3]], 2, 2)
30 if(compare(x, 8)){} else { throw 'Error at 1th assert statement. Value = ' + JSON.stringify(x)
           }
31 var x = minCost([[2, 3, 4], [5, 9, 3], [2, 6, 4]], 2, 2)
32 if(compare(x, 12)){} else { throw 'Error at 2th assert statement. Value = ' + JSON.stringify(x
         )  }
33 var x = minCost([[3, 4, 5], [6, 10, 4], [3, 7, 5]], 2, 2)
34 if(compare(x, 16)){} else { throw 'Error at 3th assert statement. Value = ' + JSON.stringify(x
         )  }
```

### R.1.4   MBTSP: TYPESCRIPT

```
1  // ---------------------      MBTSP/1:   PROMPT      ---------------------
2  /**
3   * Write a function to find the minimum cost path to reach (m, n) from (0, 0) for the given
        cost matrix cost[][] and a position (m, n) in cost[][].
4   *
5   * Examples:
6   * >>> min_cost([[1, 2, 3], [4, 8, 2], [1, 5, 3]], 2, 2)
7   * >>> 8
8   * >>> min_cost([[2, 3, 4], [5, 9, 3], [2, 6, 4]], 2, 2)
9   * >>> 12
10  * >>> min_cost([[3, 4, 5], [6, 10, 4], [3, 7, 5]], 2, 2)
11  * >>> 16
12  */
13 const min_cost = function (cost: Array<Array<number>>, m: number, n: number) : number  {
14 // --------------------- COMPLETION (model generated) ---------------------
15     const dp = [];
16     for (let i = 0; i <= m; i++) {
17         dp[i] = [];
18         for (let j = 0; j <= n; j++) {
19             dp[i][j] = Number.MAX_SAFE_INTEGER;
20         }
21     }
22     dp[0][0] = cost[0][0];
23     for (let i = 1; i <= m; i++) {
24         dp[i][0] = cost[i][0] + dp[i - 1][0];
25     }
26     for (let j = 1; j <= n; j++) {
27         dp[0][j] = cost[0][j] + dp[0][j - 1];
28     }
29     for (let i = 1; i <= m; i++) {
30         for (let j = 1; j <= n; j++) {
31             dp[i][j
32 // ---------------------            TEST            ---------------------
33 import * as assert from 'assert'
34
35 let actual_1 = min_cost([[1, 2, 3], [4, 8, 2], [1, 5, 3]],2,2);
36 let expected_1 = 8;
37 assert.deepEqual(actual_1, expected_1);
38
39 let actual_2 = min_cost([[2, 3, 4], [5, 9, 3], [2, 6, 4]],2,2);
40 let expected_2 = 12;
41 assert.deepEqual(actual_2, expected_2);
42
43 let actual_3 = min_cost([[3, 4, 5], [6, 10, 4], [3, 7, 5]],2,2);
44 let expected_3 = 16;
45 assert.deepEqual(actual_3, expected_3);
```

### R.1.5 MBGP: GO

```go
1  // --------------------         MBGP/1:    PROMPT        ---------------------
2  package main
3
4  // Code should be written in Go/Golang
5  // Write a function to find the minimum cost path to reach (m, n) from (0, 0) for the given
        cost matrix cost[][] and a position (m, n) in cost[][].
6  // Examples:
7  // >>> min_cost([[1, 2, 3], [4, 8, 2], [1, 5, 3]], 2, 2)
8  // >>> 8
9  // >>> min_cost([[2, 3, 4], [5, 9, 3], [2, 6, 4]], 2, 2)
10 // >>> 12
11 // >>> min_cost([[3, 4, 5], [6, 10, 4], [3, 7, 5]], 2, 2)
12 // >>> 16
13 func min_cost (cost [][]int, m int, n int) int {
14 // --------------------- COMPLETION (model generated) ---------------------
15   R := 3
16   C := 3
17   tc := make([][]int, 0)
18   tc[0][0] := cost[0][0]
19   for i := 1, m {
20     tc[i][0] := tc[i-1][0] + cost[i][0]
21   }
22   for j := 1, n {
23     tc[0][j] := tc[0][j-1] + cost[0][j]
24   }
25   for i := 1, m {
26     for j := 1, n {
27       tc[i][j] := min(tc[i-1][j-1], tc[i-1][j], tc[i][j-1]) + cost[i][j]
28     }
29   }
30   return tc[m][n]
31 }
32 // ---------------------             TEST            ---------------------
33 func deepCompare(a interface{}, b interface{}) bool {
34   aJSON, err := json.Marshal(a)
35   if err != nil {
36     panic(err)
37   }
38   bJSON, err := json.Marshal(b)
39   if err != nil {
40     panic(err)
41   }
42   var a2 interface{}
43   var b2 interface{}
44   err = json.Unmarshal(aJSON, &a2)
45   if err != nil {
46     panic(err)
47   }
48   err = json.Unmarshal(bJSON, &b2)
49   if err != nil {
50     panic(err)
51   }
52   result := reflect.DeepEqual(a2, b2)
53   if result {
54     return true
55   } else {
56     panic(fmt.Sprintf("%v != %v", a, b))
57   }
58 }
59
60 func main() {
61   actual_1 := min_cost([][]int{[]int{1, 2, 3}, []int{4, 8, 2}, []int{1, 5, 3}},2,2)
62   expected_1 := 8
63   deepCompare(actual_1, expected_1)
64
65   actual_2 := min_cost([][]int{[]int{2, 3, 4}, []int{5, 9, 3}, []int{2, 6, 4}},2,2)
66   expected_2 := 12
67   deepCompare(actual_2, expected_2)
68
69   actual_3 := min_cost([][]int{[]int{3, 4, 5}, []int{6, 10, 4}, []int{3, 7, 5}},2,2)
70   expected_3 := 16
71   deepCompare(actual_3, expected_3)
72
73 }
```

### R.1.6 MBCSP: C#

```
1  // --------------------        MBCSP/1:   PROMPT        ----------------------
2  using System;
3  using System.Collections.Generic;
4  using System.Linq;
5  using System.Text.RegularExpressions;
6
7  namespace Solution
8  {
9      public class Program
10     {
11         /// <summary>
12         /// Write a function to find the minimum cost path to reach (m, n) from (0, 0) for the
           given cost matrix cost[][] and a position (m, n) in cost[][].
13         ///
14         /// Examples:
15         /// >>> MinCost([[1, 2, 3], [4, 8, 2], [1, 5, 3]], 2, 2)
16         /// >>> 8
17         /// >>> MinCost([[2, 3, 4], [5, 9, 3], [2, 6, 4]], 2, 2)
18         /// >>> 12
19         /// >>> MinCost([[3, 4, 5], [6, 10, 4], [3, 7, 5]], 2, 2)
20         /// >>> 16
21         /// </summary>
22         public static int MinCost (List<List<int>> cost, int m, int n)
23         {
24  // --------------------- COMPLETION (model generated) ----------------------
25             // TODO: Write your code here
26             return 0;
27         }
28  // ---------------------             TEST             ----------------------
29         public static void Main(string[] args)
30         {
31             CompareLogic compareLogic = new CompareLogic();
32             var actual1 = MinCost(new List<List<int>> {new List<int> {1,2,3},new List<int>
           {4,8,2},new List<int> {1,5,3}},2,2);
33             var expected1 = 8;
34             var result1 = compareLogic.Compare(actual1, expected1);
35             if (!result1.AreEqual) {throw new Exception("assertion failed");}
36
37             var actual2 = MinCost(new List<List<int>> {new List<int> {2,3,4},new List<int>
           {5,9,3},new List<int> {2,6,4}},2,2);
38             var expected2 = 12;
39             var result2 = compareLogic.Compare(actual2, expected2);
40             if (!result2.AreEqual) {throw new Exception("assertion failed");}
41
42             var actual3 = MinCost(new List<List<int>> {new List<int> {3,4,5},new List<int>
           {6,10,4},new List<int> {3,7,5}},2,2);
43             var expected3 = 16;
44             var result3 = compareLogic.Compare(actual3, expected3);
45             if (!result3.AreEqual) {throw new Exception("assertion failed");}
46
47         }
48     }
49 }
```

### R.1.7    MBPHP: PHP

```
1  // --------------------        MBPHP/1:   PROMPT        ----------------------
2  /**
3   * You are an expert PHP programmer, and here is your task.
4   * Write a function to find the minimum cost path to reach (m, n) from (0, 0) for the given
        cost matrix cost[][] and a position (m, n) in cost[][].
5   * php > minCost([[1, 2, 3], [4, 8, 2], [1, 5, 3]], 2, 2)
6   * 8
7   * php > minCost([[2, 3, 4], [5, 9, 3], [2, 6, 4]], 2, 2)
8   * 12
9   * php > minCost([[3, 4, 5], [6, 10, 4], [3, 7, 5]], 2, 2)
10  * 16
11  */
12 function minCost($cost, $m, $n) {
13 // --------------------- COMPLETION (model generated) ----------------------
14 $min = $cost[$m][$n];
15     for ($i = $m + 1; $i < $m + $n; $i++) {
16         for ($j = $i + 1; $j < $m + $n; $j++) {
17             if ($cost[$i][$j] < $min) {
18                 $min = $cost[$i][$j];
19             }
20         }
21     }
22     return $min;
```

```
23 }
24 // ---------------------          TEST            ----------------------
25 function compare($x, $y) {
26     return $x == $y;
27 }
28
29 $x = minCost([[1, 2, 3], [4, 8, 2], [1, 5, 3]], 2, 2);
30 if (!compare($x, 8)) {
31     throw new Exception("Error at 1th assert statement.");
32 }
33
34 $x = minCost([[2, 3, 4], [5, 9, 3], [2, 6, 4]], 2, 2);
35 if (!compare($x, 12)) {
36     throw new Exception("Error at 2th assert statement.");
37 }
38
39 $x = minCost([[3, 4, 5], [6, 10, 4], [3, 7, 5]], 2, 2);
40 if (!compare($x, 16)) {
41     throw new Exception("Error at 3th assert statement.");
42 }
```

## R.1.8   MBRBP: RUBY

```
1  # ---------------------       MBRBP/1:   PROMPT      ----------------------
2  ##
3  # You are an expert Ruby programmer, and here is your task.
4  # Write a function to find the minimum cost path to reach (m, n) from (0, 0) for the given
        cost matrix cost[][] and a position (m, n) in cost[][].
5  # irb> min_cost([[1, 2, 3], [4, 8, 2], [1, 5, 3]], 2, 2)
6  # => 8
7  # irb> min_cost([[2, 3, 4], [5, 9, 3], [2, 6, 4]], 2, 2)
8  # => 12
9  # irb> min_cost([[3, 4, 5], [6, 10, 4], [3, 7, 5]], 2, 2)
10 # => 16
11
12 def min_cost(cost, m, n)
13 # --------------------- COMPLETION (model generated) ----------------------
14     # Write your code here.
15     if cost[m][n] == 0:
16         return 0
17     if m == 0 and n == 0:
18         return cost[m][n]
19     if m == 0:
20         return cost[m][n] + min_cost(cost, m, n - 1)
21     if n == 0:
22         return cost[m][n] + min_cost(cost, m - 1, n)
23     return cost[m][n] + min(min_cost(cost, m - 1, n), min_cost(cost, m, n - 1))
24 end
25 # ---------------------          TEST            ----------------------
26 x = min_cost([[1, 2, 3], [4, 8, 2], [1, 5, 3]], 2, 2)
27 if x != 8
28     raise StandardError, "Error at test case 1"
29 end
30 x = min_cost([[2, 3, 4], [5, 9, 3], [2, 6, 4]], 2, 2)
31 if x != 12
32     raise StandardError, "Error at test case 2"
33 end
34 x = min_cost([[3, 4, 5], [6, 10, 4], [3, 7, 5]], 2, 2)
35 if x != 16
36     raise StandardError, "Error at test case 3"
37 end
```

## R.1.9   MBKP: KOTLIN

```
1  // ---------------------       MBKP/1:   PROMPT      ----------------------
2  /**
3   * You are an expert Kotlin programmer, and here is your task.
4   * Write a function to find the minimum cost path to reach (m, n) from (0, 0) for the given
        cost matrix cost[][] and a position (m, n) in cost[][].
5   * >>> minCost([[1, 2, 3], [4, 8, 2], [1, 5, 3]], 2, 2)
6   * 8
7   * >>> minCost([[2, 3, 4], [5, 9, 3], [2, 6, 4]], 2, 2)
8   * 12
9   * >>> minCost([[3, 4, 5], [6, 10, 4], [3, 7, 5]], 2, 2)
10  * 16
11  */
12 fun minCost(cost : List<List<Int>>, m : Int, n : Int) : Int {
```

```
13  // --------------------- COMPLETION (model generated) ---------------------
14      var dp = Array(m + 1).fill(0).map { Array(n + 1).fill(0) }
15      dp[0][0] = cost[0][0]
16      for (i in 0 until m) {
17          for (j in 0 until n) {
18              if (i == 0 || j == 0) {
19                  dp[i + 1][j + 1] = cost[i][j]
20              } else {
21                  dp[i + 1][j + 1] = cost[i][j] + min(dp[i][j], dp[i][j + 1], dp[i + 1][j])
22              }
23          }
24      }
25      return dp[m][n]
26  }
27  // ---------------------            TEST            ---------------------
28  fun main() {
29      var arg00 : List<List<Int>> = mutableListOf(mutableListOf(1, 2, 3), mutableListOf(4, 8, 2)
        , mutableListOf(1, 5, 3))
30      var arg01 : Int = 2
31      var arg02 : Int = 2
32      var x0 : Int = minCost(arg00, arg01, arg02)
33      if (x0 != 8) {
34          throw Exception("Exception -- test case 0 did not pass. x0 = " + x0)
35      }
36
37      var arg10 : List<List<Int>> = mutableListOf(mutableListOf(2, 3, 4), mutableListOf(5, 9, 3)
        , mutableListOf(2, 6, 4))
38      var arg11 : Int = 2
39      var arg12 : Int = 2
40      var x1 : Int = minCost(arg10, arg11, arg12)
41      if (x1 != 12) {
42          throw Exception("Exception -- test case 1 did not pass. x1 = " + x1)
43      }
44
45      var arg20 : List<List<Int>> = mutableListOf(mutableListOf(3, 4, 5), mutableListOf(6, 10,
        4), mutableListOf(3, 7, 5))
46      var arg21 : Int = 2
47      var arg22 : Int = 2
48      var x2 : Int = minCost(arg20, arg21, arg22)
49      if (x2 != 16) {
50          throw Exception("Exception -- test case 2 did not pass. x2 = " + x2)
51      }
52
53  }
```

### R.1.10    MBCPP: C++

```
1   // ---------------------        MBCPP/1:   PROMPT        ---------------------
2   #include <bits/stdc++.h>
3   using namespace std;
4
5
6   /**
7    * Write a function to find the minimum cost path to reach (m, n) from (0, 0) for the given
         cost matrix cost[][] and a position (m, n) in cost[][].
8    * > minCost(vector<vector<int>>{{1, 2, 3}, {4, 8, 2}, {1, 5, 3}}, 2, 2)
9    * 8
10   * > minCost(vector<vector<int>>{{2, 3, 4}, {5, 9, 3}, {2, 6, 4}}, 2, 2)
11   * 12
12   * > minCost(vector<vector<int>>{{3, 4, 5}, {6, 10, 4}, {3, 7, 5}}, 2, 2)
13   * 16
14   */
15  int minCost(vector<vector<int>> cost, int m, int n) {
16
17  // --------------------- COMPLETION (model generated) ---------------------
18      vector<int> dp(m + 1, n + 1);
19      dp[0][0] = 0;
20      for (int i = 1; i <= m; i++) {
21          dp[i][0] = dp[i - 1][0] + cost[i - 1][0];
22      }
23      for (int j = 1; j <= n; j++) {
24          dp[0][j] = dp[0][j - 1] + cost[0][j - 1];
25      }
26      for (int i = 1; i <= m; i++) {
27          for (int j = 1; j <= n; j++) {
28              dp[i][j] = cost[i - 1][j - 1] + Math.min(dp[i - 1][j], dp[i][j - 1]);
29          }
30      }
31      return dp[m][n];
```

```
32  }
33
34  // --------------------            TEST            --------------------
35
36  template <typename T> bool compare(T a, T b){
37      return a == b;
38  }
39
40  int main(int argc, char* argv[]) {
41      int x0 = minCost(vector<vector<int>>{{1, 2, 3}, {4, 8, 2}, {1, 5, 3}}, 2, 2);
42      if (!(compare(x0, 8))) {
43          throw runtime_error("Exception -- test case 0 did not pass.");
44      }
45
46      int x1 = minCost(vector<vector<int>>{{2, 3, 4}, {5, 9, 3}, {2, 6, 4}}, 2, 2);
47      if (!(compare(x1, 12))) {
48          throw runtime_error("Exception -- test case 1 did not pass.");
49      }
50
51      int x2 = minCost(vector<vector<int>>{{3, 4, 5}, {6, 10, 4}, {3, 7, 5}}, 2, 2);
52      if (!(compare(x2, 16))) {
53          throw runtime_error("Exception -- test case 2 did not pass.");
54      }
55
56      return 0;
57  }
```

### R.1.11   MBPLP: PERL

```
1   # --------------------      MBPLP/1:   PROMPT      --------------------
2   # You are an expert Perl programmer, and here is your task.
3   # Write a function to find the minimum cost path to reach (m, n) from (0, 0) for the given
        cost matrix cost[][] and a position (m, n) in cost[][].
4   # >>> min_cost([[1, 2, 3], [4, 8, 2], [1, 5, 3]], 2, 2)
5   # 8
6   # >>> min_cost([[2, 3, 4], [5, 9, 3], [2, 6, 4]], 2, 2)
7   # 12
8   # >>> min_cost([[3, 4, 5], [6, 10, 4], [3, 7, 5]], 2, 2)
9   # 16
10  sub min_cost
11  {
12      my ($cost, $m, $n) = @_;
13
14  # -------------------- COMPLETION (model generated) --------------------
15    my $min = 0;
16    my $path = 0;
17    my $i = 0;
18    my $j = 0;
19    my $min_path = 0;
20    my $min_i = 0;
21    my $min_j = 0;
22    while ($i < $m and $j < $n) {
23      if ($cost[$i][$j] < $min) {
24        $min = $cost[$i][$j];
25        $min_i = $i;
26        $min_j = $j;
27      }
28      if ($i < $m - 1) {
29        $i += 1;
30      } else {
31        $i = 0;
32        $j += 1;
33      }
34    }
35    $path = $min_i + 1;
36    $path
37  # -------------------- CANONICAL SOLUTION --------------------
38  use Data::Compare;
39  my $arg00 = [[1, 2, 3], [4, 8, 2], [1, 5, 3]];
40  my $arg01 = 2;
41  my $arg02 = 2;
42  my $x0 = min_cost($arg00, $arg01, $arg02);
43  my $v0 = 8;
44  unless(Compare($x0, $v0)) {
45      die "Exception -- test case 0 did not pass.";
46  }
47  my $arg10 = [[2, 3, 4], [5, 9, 3], [2, 6, 4]];
48  my $arg11 = 2;
49  my $arg12 = 2;
```

```perl
50  my $x1 = min_cost($arg10, $arg11, $arg12);
51  my $v1 = 12;
52  unless(Compare($x1, $v1)) {
53      die "Exception -- test case 1 did not pass.";
54  }
55  my $arg20 = [[3, 4, 5], [6, 10, 4], [3, 7, 5]];
56  my $arg21 = 2;
57  my $arg22 = 2;
58  my $x2 = min_cost($arg20, $arg21, $arg22);
59  my $v2 = 16;
60  unless(Compare($x2, $v2)) {
61      die "Exception -- test case 2 did not pass.";
62  }
```

### R.1.12   MBSCP: SCALA

```scala
1  // ---------------------         MBSCP/1:    PROMPT        ---------------------
2  object Main extends App {
3      /**
4       * You are an expert Scala programmer, and here is your task.
5       * Write a function to find the minimum cost path to reach (m, n) from (0, 0) for the
         given cost matrix cost[][] and a position (m, n) in cost[][].
6       * >>> minCost(List(List(1, 2, 3), List(4, 8, 2), List(1, 5, 3)), 2, 2)
7       * 8
8       * >>> minCost(List(List(2, 3, 4), List(5, 9, 3), List(2, 6, 4)), 2, 2)
9       * 12
10      * >>> minCost(List(List(3, 4, 5), List(6, 10, 4), List(3, 7, 5)), 2, 2)
11      * 16
12      */
13     def minCost(cost : List[List[Int]], m : Int, n : Int) : Int = {
14
15  // --------------------- COMPLETION (model generated) ---------------------
16         val dp = Array(m + 1, n + 1)
17         dp(0, 0) = 0
18         for (i <- 0 until m) {
19             for (j <- 0 until n) {
20                 if (i == 0 && j == 0) continue
21                 if (i == 0) dp(i, j) = cost(i, j)
22                 else if (j == 0) dp(i, j) = dp(i - 1, j) + cost(i, j)
23                 else dp(i, j) = min(dp(i - 1, j), dp(i, j - 1)) + cost(i, j)
24             }
25         }
26         println(dp(m, n))
27     }
28  // ---------------------              TEST         ---------------------
29
30  var arg00 : List[List[Int]] = List(List(1, 2, 3), List(4, 8, 2), List(1, 5, 3))
31  var arg01 : Int = 2
32  var arg02 : Int = 2
33  var x0 : Int = minCost(arg00, arg01, arg02)
34  var v0 : Int = 8
35  assert(x0 == v0, "Exception -- test case 0 did not pass. x0 = " + x0)
36
37  var arg10 : List[List[Int]] = List(List(2, 3, 4), List(5, 9, 3), List(2, 6, 4))
38  var arg11 : Int = 2
39  var arg12 : Int = 2
40  var x1 : Int = minCost(arg10, arg11, arg12)
41  var v1 : Int = 12
42  assert(x1 == v1, "Exception -- test case 1 did not pass. x1 = " + x1)
43
44  var arg20 : List[List[Int]] = List(List(3, 4, 5), List(6, 10, 4), List(3, 7, 5))
45  var arg21 : Int = 2
46  var arg22 : Int = 2
47  var x2 : Int = minCost(arg20, arg21, arg22)
48  var v2 : Int = 16
49  assert(x2 == v2, "Exception -- test case 2 did not pass. x2 = " + x2)
50
51  }
```

### R.1.13   MBSWP: SWIFT

```swift
1  // ---------------------         MBSWP/1:    PROMPT        ---------------------
2  /**
3   * You are an expert Swift programmer, and here is your task.
4   * Write a function to find the minimum cost path to reach (m, n) from (0, 0) for the given
       cost matrix cost[][] and a position (m, n) in cost[][].
5   * >>> minCost([[1, 2, 3], [4, 8, 2], [1, 5, 3]], 2, 2)
```

```
6   * 8
7   * >>> minCost([[2, 3, 4], [5, 9, 3], [2, 6, 4]], 2, 2)
8   * 12
9   * >>> minCost([[3, 4, 5], [6, 10, 4], [3, 7, 5]], 2, 2)
10  * 16
11  */
12  func minCost(cost : [[Int]], m : Int, n : Int) -> Int {
13  // --------------------- COMPLETION (model generated) ---------------------
14      if (m == 0 || n == 0) {
15          return 0;
16      }
17      if (m == 1 || n == 1) {
18          return cost[0][1];
19      }
20      if (cost[m][n] != -1) {
21          return cost[m][n];
22      }
23      cost[m][n] = min(minCost(cost, m - 1, n) + cost[m][n], minCost(cost, m, n - 1) + cost[m][n
        ]);
24      return cost[m][n];
25  }
26  // --------------------- CANONICAL SOLUTION ---------------------
27  var arg00 : [[Int]] = [[1, 2, 3], [4, 8, 2], [1, 5, 3]]
28  var arg01 : Int = 2
29  var arg02 : Int = 2
30  var x0 : Int = minCost(cost : arg00, m : arg01, n : arg02)
31  var v0 : Int = 8
32  assert(x0 == v0, "Exception -- test case 0 did not pass. x0 = ")
33
34  var arg10 : [[Int]] = [[2, 3, 4], [5, 9, 3], [2, 6, 4]]
35  var arg11 : Int = 2
36  var arg12 : Int = 2
37  var x1 : Int = minCost(cost : arg10, m : arg11, n : arg12)
38  var v1 : Int = 12
39  assert(x1 == v1, "Exception -- test case 1 did not pass. x1 = ")
40
41  var arg20 : [[Int]] = [[3, 4, 5], [6, 10, 4], [3, 7, 5]]
42  var arg21 : Int = 2
43  var arg22 : Int = 2
44  var x2 : Int = minCost(cost : arg20, m : arg21, n : arg22)
45  var v2 : Int = 16
46  assert(x2 == v2, "Exception -- test case 2 did not pass. x2 = ")
```

## R.2 MULTI-LINGUAL HUMANEVAL

HumanEval contains 164 cases, most of which are compatible with our conversion framework. For some cases where the tests are not explicit, such as using Python for loop to iterate over many test cases, we expand them out explicitly to make it compatible with the conversion framework. For instance, the test statement below

```
1  for x in range(2, 8):
2      assert candidate(x, x+1) == str(x)
```

is expanded to

```
1  assert candidate(2, 3) == "2"
2  assert candidate(3, 4) == "3"
3  assert candidate(4, 5) == "4"
4  assert candidate(5, 6) == "5"
5  assert candidate(6, 7) == "6"
6  assert candidate(7, 8) == "7"
```

There are some cases that we filtered out such as cases that involve a user defined function. In total, we keep 161 out of 164 cases. We format of multi-lingual HumanEval are similar to that of MBXP in each language; therefore, we skip the display of examples in this section for brevity.

## R.3 MULTI-LINGUAL MATHQA

By extending MathQA-python datasets Austin et al. (2021) for other programming languages, we obtained MathQA-Java and MathQA-JavaScript, for the purpose of evaluating the ability of the models to reason and synthesize code from more complex text, under multiple languages. The original MathQA-python problem contains a short text (which describes a mathematical question),

an answer (usually a real number) and a canonical solution in Python. Based on this, to build a version in a different language, we perform following two transformation steps:

- Convert MathQA-Python problem into our canonical MBXP format (Section 3). Specifically, we construct a unified function signature, ie. `def problem():`, followed by a docstring, which is equivalent to the short text of the original MathQA-Python. Additionally, a single test case can be generated based on the given answer, ie. `assert problem() == answer`.

- Obtain prompts and test cases in another language for execution-based evaluation using our proposed rule-based conversion framework.

- For the conversion framework outlined in Section 3, we emphasize that we handle floating point comparing numbers to be within $\epsilon = 1e - 8$ instead of exact comparison. This handling is suitable for floating points and helps avoid potential false negatives. It is also compatible with all conversions in other datasets since it is handled within the abstract `compare` function in each target language.

Below, we show converted examples after the first step (including Python prompts, the canonical solution and a single test case) and its counterparts for Java and JavaScript generated from the second step.

```
1
2  -------------------- MathQA-Python --------------------
3  def problem():
4      """
5      a shopkeeper sold an article offering a discount of 5 % and earned a profit of 31.1 % .
        what would have been the percentage of profit earned if no discount had been offered ? n0
        = 5.0 n1 = 31.1
6      """
7      n0 = 5.0
8      n1 = 31.1
9      t0 = n1 + 100.0
10     t1 = 100.0 - n0
11     t2 = t0 * 100.0
12     t3 = t2 / t1
13     answer = t3 - 100.0
14     return answer
15
16  import math
17  def compare(x, y):
18      return math.fabs(x-y)<1e-8
19  candidate = problem
20  assert compare(candidate(), 38.0)
21
22
23  -------------------- MathQA-Java --------------------
24  import java.io.*;
25  import java.lang.*;
26  import java.util.*;
27  import java.math.*;
28
29  class Problem {
30
31      /**
32       * a shopkeeper sold an article offering a discount of 5 % and earned a profit of 31.1 % .
         what would have been the percentage of profit earned if no discount had been offered ?
         n0 = 5.0 n1 = 31.1
33       */
34      public static double problem() {
35          ...... // the model output are inserted here.
36      }
37  }
38
39  class Main {
40      public static boolean compare(Object obj1, Object obj2) {
41          if (obj1 == null && obj2 == null){
42              return true;
43          } else if (obj1 == null || obj2 == null){
44              return false;
45          } else {
46              if ((obj1 instanceof Double || obj1 instanceof Float) &&
47                      (obj2 instanceof Double || obj2 instanceof Float)
48                  ){
49                  if (obj1 instanceof Float){
50                      obj1 = ((Float) obj1).doubleValue();
```

```
51                    }
52                    if (obj2 instanceof Float){
53                        obj2 = ((Float) obj2).doubleValue();
54                    }
55                    return Math.abs((double)obj1 - (double)obj2) < 1e-7;
56                }
57                else
58                    return obj1.equals(obj2);
59            }
60        }
61
62        //execution-based test case
63        public static void main(String[] args) throws Exception {
64            double x0 = Problem.problem();
65            if (!(compare(x0, 38.0))) {
66                throw new java.lang.Exception("Exception -- test case 0 did not pass. x0 = " + x0)
         ;
67            }
68
69        }
70 }
71
72
73 -------------------- MathQA-JavaScript --------------------
74
75 /**
76  * a shopkeeper sold an article offering a discount of 5 % and earned a profit of 31.1 % .
        what would have been the percentage of profit earned if no discount had been offered ? n0
        = 5.0 n1 = 31.1
77  */
78 function problem() {
79    ... // the model output are inserted here.
80 }
81
82 // execution-based test case
83 const _ = require("lodash")
84
85 function compare(object1, object2){
86    if(typeof object1 == "number" && typeof object2 == "number") {
87        return Math.abs(object1 - object2) < 1e-7
88    }
89    else{
90        return _.isEqual(object1, object2)
91    }
92 }
93
94 var x = problem()
95 if(compare(x, 38.0)){} else { throw 'Error at 1th assert statement. Value = ' + JSON.stringify
        (x)   }
```

