# OpenReview forum: "Multi-lingual Evaluation of Code Generation Models"
_ICLR.cc/2023/Conference — ICLR 2023 notable top 25%_

### Official Review · Reviewer_rPgV · 2022-10-19

**Confidence:** 4
**Correctness:** 4
**Technical Novelty And Significance:** 2
**Empirical Novelty And Significance:** 3
**Recommendation:** 8

**Clarity, Quality, Novelty And Reproducibility:**

Clarity:
The paper is well-written and easy to follow. One minor point is that there is some ambiguity in which model is being shown in Figure 3.

Novelty:
The idea to effectively “translate” the prompt and test cases, which may have simpler structures, is novel and interesting. It can also have broader applications for different natural languages, when extrinsic-evaluation is used. There is empirical novelty in terms of a new dataset that spans multiple programming languages, as well as with a lot of new empirical results for many different programming languages (e.g., in-domain vs out-of-domain training, monolingual vs multi-lingual training, model sizes, source/target pairs for translation). Additionally, the technical novelty seems to be somewhat limited, as the rule-based static analysis techniques used to build the framework seem to be derived from existing work/tools.

Quality:
The experiments are well-designed and findings are well-grounded.

Reproducibility:
If the datasets are publicly released (including the pretraining ones), the experiments should be reproducible. There is a detailed Appendix with training details provided in Appendix N.




**Strength And Weaknesses:**

Strengths:
- Deriving multilingual code data without having to translate code solutions (especially considering how error-prone code translation models are) is a neat idea and useful.
- Since it is challenging to get parallel data to study tasks like code translation, this framework which can give synthetic canonical solutions could be particularly useful for studying that tasks.
- There are extensive empirical results, including supplementary analyses (e.g., best combos of source/target pairs for zero-shot translation) and even minor observations (e.g., sampling efficiency, validation losses) that can be beneficial for guiding future work.

Weaknesses:
- One concern I have is regarding whether potential biases/assumptions in the original MBPP dataset can transfer over to the MBXP datasets. These could be possible ambiguities (as the authors have mentioned  in Section 2.1) but also more fundamental biases/assumptions, such as language-specific ones. Could the formulation of the prompt and the types of tests that are needed drastically change based on the specific language, depending on for example, whether it is a language that requires explicit memory management?
- While the synthetic canonical solutions may be useful for training during different tasks like translation, I find that using them for test to be slightly problematic because these are model-generated, and possible errors/biases in this output could propagate, and we don’t want to be evaluating using such references. However, these execution-based sets like MBPP are quite small, so it is not clear whether they offer enough training data to have a meaningful impact.
- It is not clear whether training the monolingual and multilingual models in Section 3.1 was necessary. As described in Appendix M, CodeGen entails monolingual and multilingual models trained on a subset of the languages studied in this work, including Python Java and Javascript, which were used in this work. Based on the results shown in  Appendix M, it seems that many of those models are better performing than the ones trained in this work. Most (if not all) of the analyses could have been done with these.



**Summary Of The Paper:**

This paper presents a framework for deriving multi-lingual data from an execution-based dataset (e.g., MBPP, HumanEval, MathQA), which entails applying rule-based transformations based on static analysis to the prompt and test cases. The key idea is that the canonical solution does not need to be translated since evaluation is done by running the test cases. Through this, they translate MBPP into 10+ languages. They train large decoder-only monolingual and multi-lingual models, of various sizes, using three languages, and they evaluate them in out-of-domain and in-domain settings. They highlight the benefits of the large multi-lingual models especially in the out-of-domain settings. They also consider few-shot learning, which results in fewer errors, and is most advantageous in out-of-domain evaluation. They further consider other evaluation tasks like zero-shot translation, code insertion, and summarization. Additionally, their framework also allows deriving synthetic canonical solutions for tasks which require it.

Contributions:
- A framework for acquiring multi-lingual data from a monolingual dataset, without directly requiring translation of code solutions.
- A large multilingual dataset which can be used for studying a variety of code-related end tasks, spanning multiple languages.
- Empirical results and observations, spanning many languages, model sizes, settings, and tasks.


**Summary Of The Review:**

Overall, the intuition and motivation behind this work are neat. The paper presents interesting empirical results and observations related to multilingual aspects of code generation, especially considering much of the focus in recent years has been on only a few languages like Python and Java. They also present a dataset spanning multiple languages, which could potentially be useful for the broader research community, although I have some concerns about whether their framework which effectively “translates” prompts and test cases in existing datasets might be propagating biases. Additionally, the technical novelty behind the framework seems to be somewhat limited. Nonetheless, I would say that the empirical novelty is still relatively significant.

[After Author Response]
I have read the response, and the authors have addressed most of my concerns. I have increased my score.

---

> ### Author Response · Authors · 2022-11-15
> **Addressing Reviewer rPgV's comments**
>
>
> * Thank you the insightful summary and for acknowledging the novelty and impact of our work. It is clear that the reviewer has read our paper carefully and understood the paper at a deep level and even gone through parts of supplementary materials to check for details.
>     * We thank the reviewer for recognizing the value of the parallel data which would be hard to construct otherwise — this was made possible due to our proposed transpiler for language conversion and the synthetic solutions which are in turn possible to due the existence of the converted tests.
>
> * We appreciate your concerns above — we have thought about these points as well. Below are our responses to help clarify.
>
> * [Mono vs. Multi] The question related to multi- vs mono-lingual and CodeGen is in fact a very important one. Our training is different in that we carefully control for the language sources we used to train, so that we can make observations regarding in-domain vs out-of-domain and fair comparison between multi- and mono-lingual models. We would not be able to make such conclusions based on existing models, even with CodeGen models which are labeled as mono-lingual versus multi-lingual.
>     * This is because CodeGen-mono is trained on many other sources besides Python, so the non-zero performance on other languages is very much expected does not reveal much about generalization behavior. In fact, CodeGen-mono is trained on BigPython, on top of BigQuery (6 languages) as well as the PILE (mostly text, but likely other programming languages data with unknown number of languages). See Appendix A.1 page 14 of CodeGen paper (https://arxiv.org/pdf/2203.13474.pdf)
>     * In addition, to compare mono- versus multi-lingual model performances, our work controls the compute and data for both settings to be equivalent. That is, both mono-lingual Python model and multi-lingual models gets 210B tokens of total compute corresponding to the same Python data (See Section 3.1 for more details). However, CodeGen uses different amount of Python data for mono and multi- settings. That is, BigPython as the mono-lingual Python data has 217GB of Python whereas BigQuery (multi-lingual data source) has 55.9GB out of 341.1GB of all data. Due to these differences, even if CodeGen-mono was only trained on BigPython without BigQuery, we would not be able to compare multi- and mono-lingual models meaningfully.
>         * Side note: From CodeGen, it seems as if mono-lingual models are better than multi-lingual when evaluated on Python, opposite to our observations, but we argue that it is likely due to the size of the BigPython which was excluded from multi-lingual training.
>     * Note: Our data collection of only permissively licensed code also allows us to release the synthetic solutions generated by our model. This adds to the careful design of our work.
>
> * [Translation evaluation] When we evaluate whether the model generates correct code in language Y or not, the reference solution in language X does influence how the model generate, since the model sees it as part of the context. However, whether the generated code in language Y is correct or not depends entirely on the tests in language Y, which is case is independent of the synthetic canonical solutions in language X, since these tests are converted beforehand. The canonical solution in language simply X serves as a reference that the model can refer to. Other evidence that can potentially reassure us of the synthetic solution quality, besides passing generated tests.
>     * The fact that using language X = Java, JS (synthetic solutions) as reference improve the function completion scores indicate that these solutions are of good quality at a high level. Manual inspection also reveal that the translated code makes sense at a high level.

---

> > ### Author Response · Authors · 2022-11-15
> > **Addressing Reviewer rPgV's comments - Part 2**
> >
> > * [Bias Carryover] The potential bias that might carry over from one language to another is a valid point. We observe that for basic programming problems that we cover, such language-specific biases are minimal.
> >     * We consider two main types of code-generation evaluations: (1) evaluating language/API specific capabilities (2) evaluating basic programming capabilities and natural language understanding.
> >     * Examples of (1) involve specific tests such as correct memory allocation in C, webpage construction with HTML5/JavaScript, server-client communication with Java, or more generally correct usage of specific APIs. Such evaluation requires specific types of tests, called integration tests, which are far more complex than rather than simply checking the returned values as in the basic programming cases. Testing these complex and language-specific capabilities likely requires either case-by-case manual annotation or large scale data scraping of integration tests and are likely not transferrable from one language to another.
> >     * The focus on this work is mainly on (2) where the problems focus on the generic understanding of problem description with solutions involving simple data structures (such as finding binomial coefficient, string manipulation, arithmetic problems, etc.) The correctness check also relies entirely on the input and the output — these correctness checks are not language specific, which allow for the conversion. While (2) is not impervious to bias, it is less likely to be language-specific, and allow us to scale the evaluation to measure of capabilities related to basic programming languages of code language models. That being said, during our review process with human annotators, we have not found problems that are unnatural in the new languages due to the conversion.
> >
> >     * For basic programming evaluation, the advantage of the automated conversion framework is that, if we observe tasks that require correcting (in terms of ambiguity, or increasing to test coverage), we can correct them in the original Python datasets, and propagate the changes to all other languages due to our automated conversion. This is a much more efficient way than using human annotator to correct all cases.
> >
> >
> > * We thank the reviewer for the detailed comments. Let us know if you have any other questions.

---

### Official Review · Reviewer_DMjf · 2022-10-19

**Confidence:** 4
**Correctness:** 3
**Technical Novelty And Significance:** 3
**Empirical Novelty And Significance:** 3
**Recommendation:** 6

**Clarity, Quality, Novelty And Reproducibility:**

- Clarity The paper is easy to follow
- Novelty: The technical novelty is limited, however this paper conducts extensive experiments to provide some empirical findings. Furthermore, the constructed benchmark is valuable.
- Quality: The experiments are extensive and some findings are interesting.
- Reproducibility: The experiments should be reproducible if all data and models are released.



**Strength And Weaknesses:**

Strength:

- The provided benchmark is valuable for AI4Code community. Currently, MBPP is a dataset designed for python, which is limited for evaluation, This paper proposes a new benchmark based on MBPP to enrich the evaluated programming languages in Python, Java, Go, etc. (13 in total) and this dataset is very valuable for the community.

- The extensive evaluation on this benchmark is very appreciated. Besides the code completion task including  multi-lingual version and  mono-lingual version, this paper also evaluates on zero-shot translation, prompt robustness, code insertion and the summarization task to show the effectiveness of the constructed benchmark.

- There are also some findings by the extensive evaluation and these findings can provide the readers with some insights about these large-scale pre-trained models.

Although I appreciated the authors for the extensive evaluation, I also find some problems and hope to fix these issues in the next version.

- The synthetic canonical solutions are generated by the models, so I do not know how diversity they have compared with the real semantic-equivalent data?

- In the abstract section, the authors mention that they will release part of the benchmark, but the main contribution for this paper is this benchmark, why not release the entire benchmark for the following-up researchers?

- I suggest authors provide more details about the quality check via reviewers part. How many reviewers do you have? How to evaluate the converted version? How to ensure the converted problem is correct? How long do you take to evaluate these converted problems?

- In Figure 3, I cannot know which line belongs to which model.  It should add some clarifications.



**Summary Of The Paper:**

This paper proposes a new benchmark MBXP for 10+ programming languages to facilitate the evaluation of execution-based code completion task. By the design of the conversion framework to translate prompts and test cases from the MBPP dataset, MBXP can evaluate code generation models in a multi-lingual manner. Furthermore, by the code generation model and some defined strategies, they can also provide the canonical samples in different programming languages and these data can also used for different code-related tasks such as source code summarization, code translation etc.

**Summary Of The Review:**

This paper propose a framework to construct a new benchmark across different programming languages for different code-related tasks. Based on this bechmark, an entensive experiment is conducted. Although the technique novelty is limited, I appreciate the efforts in this paper for some interesting findings.

---

> ### Author Response · Authors · 2022-11-15
> **Addressing Reviewer DMjf 's comments**
>
> * Thank you for for acknowledging the value of our datasets as well as the effort involving our experiments and findings. We address the questions below.
> * [Data release] We plan to release all parts of the benchmark — what we mean is that the benchmark encapsulates the additional tasks, and these additional tasks are released as part of the MBXP benchmark. We will clarify it in the updated draft.
>
> * [ Annotation review ] We have many forms of quality checks and will briefly summarize here:
>     * [automated check] We test that the prompt and test cases generated are syntactically correct, and are parsable/compilable. This helps prevent mistakes that results in incorrect test cases / invalid function signatures.
>     * [Quality check via annotators] The review spans 15 days with 8 total annotators. We show annotators the converted problem, each of which consists of (1) function signature, (2) natural language description, as part of the docstring (3) test cases. We ask annotators to determine whether each of these three components is correct or not, and to explain the reason if any of them is incorrect. After that, authors review the errors pointed out by annotators to determine whether they are related to the conversion framework or the original datasets (MBPP, HumanEval). After the final set of review, we do not find instances where our transpiler performs incorrect conversion — some ambiguity that the reviewers found are attributed to the original datasets.
>     * On the synthetic solutions, we do not have real semantic equivalent data to compare since they would require human annotators to write solutions. However, in terms of the quality of such solutions, we do that that, for instance, they help increase the scores in the translation setting to other languages (Section 3.3). Also using higher amount of right context from these synthetic solutions also uniformly increase prediction scores (Section 3.5 and appendix K).
>
>
>
> * In Figure 3, the results are from the 13B multi-lingual model (we will add it to the caption for clarity).
> * [Transpiler / language conversion framework] We emphasize that the transpiler design is a significant undertaking that requires careful consideration and effort. (However, once designed, it can be used to scale the conversion to many languages) For instance, we carefully design type deduction, allowing us to support the conversion to typed languages, even know the original Python datasets are not statically typed. We support object construction in heterogenous manner, including arbitrary nested structures. While it uses static analyses technique that existed, it connects many dots that unlocks the data conversion possibilities for execution-based evaluations.

---

### Official Review · Reviewer_3QeX · 2022-10-22

**Confidence:** 3
**Correctness:** 3
**Technical Novelty And Significance:** 3
**Empirical Novelty And Significance:** 3
**Recommendation:** 6

**Clarity, Quality, Novelty And Reproducibility:**

The method is clear and the paper well-written.

It looks novel (although I might have missed previous work since I'm not an expert of this area).

Based on the main paper, I don't think this is easily reproducible (of course I didn't read the 77 pages to check it...).

**Strength And Weaknesses:**

Strengths:
- the paper is very well written and illustrated. For someone like me who is not an expert of this area, I find this paper relatively easy to follow.
- the framework proposed is well-motivated and very simple
- this work can easily be applied to extend many benchmarks to many languages.


Weaknesses:
- The appendix is huge. It contain tons of information that would be more useful in the main paper. The limits for instance should definitely be included before the conclusion.
- most of  the details needed for reproducibility and fully understand the experiments are in the appendix. I couldn't check it thouroughly.


**Summary Of The Paper:**

This paper describes a new framework to convert prompt and test cases, for code generation models, to a new target programming language. The framework is used to generate an execution-based code completion benchmark for more than 10 programming languages.
This enables an easier evaluation of a "multilingual" model versus the standard monolingual model.

This work shows that multilingual models tend to be better than the monolingual model and show that code generation models have zero-shot capabilities.


**Summary Of The Review:**

The paper is very interesting and the method will be useful. My only concern is that the size of the PDF file submitted is huge and that I couldn't check all of it thouroughly. Some sections of the appendix must be move to the main paper before acceptance.
I would definitely recommend the authors to submit the entire paper to a journal.

---

> ### Author Response · Authors · 2022-11-15
> **Addressing Reviewer 3QeX 's comments**
>
> 1. We are glad that the reviewer finds the paper easy to follow. We thank the reviewer for acknowledging the significance and the key aspects of our framework — indeed, the data conversion framework can be used to automatically construct evaluation data for a new programming language that might not previously exist.
> 2. We agree that a journal-like version could also be appropriate for this work, given the scale of the experiments and findings (dataset construction in 10+ languages with many evaluation variants, evaluation on public and our own models trained in 4 model scales with thorough evaluation via large scale sampling) However, we believe that ICLR is a perfect venue to disseminate this research since the ICLR community’s interests and expertise align with the paper very well.
> 3. We briefly mentioned the limitation under discussion, but will gladly move more details to the main paper.
> 4. We acknowledge that the appendix is quite extensive. We see this as a strength since (1) In terms of reproducibility and providing useful references, we believe it is better to have extra details in the appendix. However, it is not necessary to check all details for full understanding. (2) we want the main text to contain sufficient amount of information for high level understanding without distracting the readers. (due to the scale of the experiments, there are a lot of details)
>
>
> We organize the paper such that reviewers do not need to go through the appendix to understand the overall picture of the work, but have the choice to dive deep into certain aspects (training, specific evaluation results, dataset examples, etc.) which are conveniently linked from the main text in appropriate sections. The fact that the paper is easy to follow is based on the careful consideration of succinctness of the main paper. :)

---

### Official Review · Reviewer_n1wH · 2022-11-02

**Confidence:** 4
**Correctness:** 4
**Technical Novelty And Significance:** 3
**Empirical Novelty And Significance:** 3
**Recommendation:** 8

**Clarity, Quality, Novelty And Reproducibility:**

The work is very clear and rigorous. It's also original, even though it might not be very creative.

**Strength And Weaknesses:**

Strengths:
- The paper is very clear and contains a coherent set of experiments.
- The paper presents a new dataset, which will be useful for future work.

Weaknesses:
- The proposed dataset creation and the experimental setup aren't very creative -- but the results are interesting, so this isn't a big weakness.

**Summary Of The Paper:**

This paper proposes a way to make an existing dataset (MBPP), an execution-based code completion benchmark, multilingual: while the original MBPP only contained python code, the new dataset (MBXP) contains a total of 10 programming languages. Problems in function completion datasets consists of *prompt*, *test statement*, and *canonical solution*. The proposed dataset creation is based on the insight that, for an execution-based benchmark, only translation of the prompt and test statement are necessary.

The paper further evaluates multiple monolingual and multilingual models on MBXP. This evaluation results in multiple interesting findings, for instance, models have a non-zero performance on languages they have not been trained on, what the authors attribute to code in multiple languages appearing in the same files. The authors further explore few-shot approaches and translation. Multilingual models mostly outperform monolingual ones.

**Summary Of The Review:**

As the paper provides interesting insights and contributes a useful dataset, I believe it should be accepted.

---

> ### Author Response · Authors · 2022-11-15
> **Addressing Reviewer n1wH 's comments**
>
> Thank you for acknowledging the significance of our work and our experiment effort, especially the recognition that our experiments provide insights into language model’s various code capabilities. This study is indeed only possible after the construction of meaningful evaluation datasets.
>
> On the creativity of our experiments, we argue that prior to the dataset construction, it was not obvious that some of the evaluation settings would yield any positive signals. For instance, it was not clear that models trained simply on code in the wild can be perform zero-shot translation, or can use few-shot prompts to reduce errors in out-of-domain languages, or have out-of-domain generalization at all. Without the imagination or “creativity” of what language language models can possibly do, it would be hard for one to have put together such probing experiments in a carefully planned manner. (See response to reviewer rPgV for more details)

---

### Author Response · Authors · 2022-11-19
**Summary**

We thank the reviewers for your thoughtful comments. In addition to the specific responses to each reviewer, we also summarize a few key points to complement the discussion.

* Understanding capabilities of large language models (or any ML models) requires meaningful evaluation that can probe the breadth and depth of models’ knowledge. High quality datasets have the potential to guide the development of research areas and help discover new abilities. We strongly believe that our datasets serve such a role, and will have a strong impact on the research of code generation in the foreseeable future.

    * One key feature of our evaluation framework is the execution-based aspect which allows us to captures code correctness more truthfully compared to the match-based approach (n-gram metric has little correlation on correctness for basic programming problems [1]). The benefits and interpretability of execution-based evaluation is evident in the case of Python, where the introduction of datasets such as HumanEval and MBPP quickly changed the field on how real progress in code generation is measured.
    * Our work expands the breadth of such evaluation to cover many more programming languages, where the availability of such multi-lingual datasets is unlocked by the language conversion transpiler. Building such a transpiler is a significant undertaking, involving careful design and considerable amount of attention to details. Once built, it is a power tool that allows to perform evaluation at scale.
    * The datasets equipped with synthetic solutions can also be composed and adapted to measure performance of other abilities. We demonstrated a few use cases (zero-shot code translation, code insertion, robustness evaluation, etc.). However, we are only scratching the surface on the possible use cases that the research community can build on top of the execution-based evaluation framework, which can now be done in 10+ languages.

* The training experiments were designed with great care.
    * We performed data filtering carefully via syntactic tree parsing with respect to each language, allowing us to observe generalization behavior due to natural occurrences of multi-lingual data. [rather than incorrect filtering or unknown data collection]
    * The comparison of multi- versus mono-lingual training is possible due to the careful control of training budget and data source. This is not the case for other publicly available “mono” or “multi”-lingual models. (See response to reviewer rPgV for more details)
    * Many of the interesting observations might seem to come at ease (generalization abilities to out-of-domain, or multi-lingual model outperforming mono-lingual once size is sufficiently large, etc). However, these trends are not necessarily obvious without hindsight. Only from careful experiment design and dedication, both training and large scale evaluation, that these patterns of results emerge.
* We thank all reviewers for recognizing the significance and impact of our work. The points made above emphasize the message we want to convey based on the reviewer feedback, and we plan to incorporate it in the future version of the paper. We thank ICLR/OpenReview for providing the venue for open discussion.




[1] Austin et al 2021. Program Synthesis with Large Language Models.

---

### Decision · Program_Chairs · 2023-01-20

**Decision:**

Accept: notable-top-25%

**Justification For Why Not Higher Score:**

The paper itself is not exceptionally novel or present a significant advance in machine learning to justify an oral presentation.

**Justification For Why Not Lower Score:**

Accepting the paper as a poster would be okay, but would reduce its visibility. Since the community seems to be interested in code generation, having good benchmarks widely known seems useful.

**Metareview: Summary, Strengths And Weaknesses:**

This work presents a new execution-based benchmark for code across multiple programming languages. Code generation benchmark datasets consists of a prompt, a test statement, and a canonical solution. The key idea is that the canonical solution does not need to be translated per language since evaluation can be done by running the test cases. Finally, the authors show how this benchmark performs in a series of tasks and LLMs.

The presented benchmark is novel and needed in the community and its clear that the authors have put a lot work into it. The conclusions drawn on the performance of LLMs on this benchmark are also useful. One weakness noted by the reviewers is that it is unclear what kind of biases and limitations the original (and translated benchmark) introduce to the measurement of a model's ability to generate code. For example, does performance in these datasets correlate with performance in code completion in library code?

**Note From Pc:**

if the above contains the word "oral" or "spotlight" please see: "oral" presentation means -> notable-top-5% and "spotlight" means -> notable-top-25%. As stated in our emails, we are disassociating presentation type from AC recommendations